# Construction of Zn-doped RuO$_2$ nanowires for efficient and stable water oxidation in acidic media

Dafeng Zhang [1,5], Mengnan Li[1,5], Xue Yong [2,3,5], Haoqiang Song [2], Geoffrey I. N. Waterhouse [4], Yunfei Yi[1], Bingjie Xue[1], Dongliang Zhang[1], Baozhong Liu[1] ✉ & Siyu Lu [2] ✉

Oxygen evolution reaction catalysts capable of working efficiently in acidic media are highly demanded for the commercialization of proton exchange membrane water electrolysis. Herein, we report a Zn-doped RuO$_2$ nanowire array electrocatalyst with outstanding catalytic performance for the oxygen evolution reaction under acidic conditions. Overpotentials as low as 173, 304, and 373 mV are achieved at 10, 500, and 1000 mA cm$^{-2}$, respectively, with robust stability reaching to 1000 h at 10 mA cm$^{-2}$. Experimental and theoretical investigations establish a clear synergistic effect of Zn dopants and oxygen vacancies on regulating the binding configurations of oxygenated adsorbates on the active centers, which then enables an alternative Ru−Zn dual-site oxide path of the reaction. Due to the change of reaction pathways, the energy barrier of rate-determining step is reduced, and the over-oxidation of Ru active sites is alleviated. As a result, the catalytic activity and stability are significantly enhanced.

Hydrogen (H$_2$) generation via electrochemical water splitting is a promising way to efficiently store intermittent renewable energy[1–3]. However, the sluggish oxygen evolution reaction (OER) on anode hinders the overall efficiency of water splitting and leads to large undesired energy consumption[4,5]. Therefore, the design of high performance OER catalysts is regarded as a matter of urgency for the industrial application of water-to-H$_2$ conversion[6–9]. To date, attractive candidates based on earth-abundant transition metals, especially the (oxy)hydroxides and layered double hydroxides of Ni−Fe[10–12], have been widely reported under basic conditions, offering a chance to build low-cost alkaline water electrolysis (AWE) assemblies without noble metals in application. However, the currently deployed AWE devices are still facing intrinsic challenges, including the low operating pressure, inevitable gas crossover, slow load response, and limited current density, mainly due to the utilization of a diaphragm and a liquid electrolyte[13].

Compared with AWE, water electrolysis using proton exchange membrane (PEM) electrolyzers can effectively address the above challenges with significantly improved performance[14–16]. But the highly corrosive conditions at high oxidation potentials under acidic environments make the development of efficient OER catalysts a great challenge. Most existing OER catalysts with excellent performance in basic condition generally show unsatisfied kinetics in acidic media, which, furthermore, suffer from severe degradation under the harsh conditions. So far, only the catalysts based on Ru and Ir noble metals can meet the requirements of PEM water electrolysis in practical deployment, though the scarcity of iridium and relatively low mass activity of Ir-based catalysts are serious obstacles to industrial scale H$_2$ production[8,17,18]. Ru-based catalysts, especially RuO$_2$, show promise as OER catalysts in acidic media, being much cheaper than their Ir-based counterparts. The moderate binding strength of OER intermediates

[1]State Collaborative Innovation Center of Coal Work Safety and Clean-efficiency Utilization, Henan Key Laboratory of Coal Green Conversion, College of Chemistry and Chemical Engineering, Henan Polytechnic University, Jiaozuo 454003, P. R. China. [2]Green Catalysis Center, and College of Chemistry, Zhengzhou University, Zhengzhou 450000, P. R. China. [3]Department of Chemistry, The University of Sheffield, Sheffield S3 7HF, UK. [4]School of Chemical Sciences, The University of Auckland, Auckland 1142, New Zealand. [5]These authors contributed equally: Dafeng Zhang, Mengnan Li, Xue Yong. ✉e-mail: bzliu@hpu.edu.cn; sylu2013@zzu.edu.cn

(O*, OH*, and OOH*) on Ru sites makes Ru-based catalysts very active for oxygen evolution, but the over-oxidation of Ru cations can create soluble species ($Ru^{n+}$, $n > 4$) under acidic OER conditions, leading to rapid catalyst degradation and large losses in performance[19–21]. Poor durability is the biggest obstacle hindering the practical application of Ru-based catalysts in PEM water electrolyzers[22,23].

Guest elements are usually introduced to improve the OER performance of $RuO_2$ by modulating the chemical environment of Ru centers[24]. As reported recently, via constructing guest single atomic (e.g., Ni, Pt)[25,26] and lattice doping (e.g., Mn, Cu, Na)[27–29] sites, the overpotential of acidic OER on $RuO_2$ can be reduced to ~180 mV@10 mA cm$^{-2}$ with a durability over 200 h[25]. It was found that the presence of charge transfer between guest atoms and Ru cations can change the electronic structures of the Ru active sites[30–33]. The introduction of electron-donating dopants into $RuO_2$ would reduce the oxidation state of Ru ($Ru^{n+}$, $n < 4$), thus protecting surface Ru cations from over oxidation to soluble species during OER[34,35]. However, lowering the Ru oxidation state can impair the catalytic activity for OER, since the strong binding of OER intermediates on low-valent Ru sites would hinder the deprotonation of the second water molecule to form *OOH species[36]. Thus, high-valent Ru species generally show faster kinetics with lower overpotentials during OER[26,37,38]. The introduction of guest metal ions further provides a chance to create structure defects (e.g., oxygen vacancies, $V_O$) to modulate the OER property of Ru centers[31]. Although the presence of $V_O$ defects would in principle reduce the oxidation state of Ru species and thus probably impair the OER activity[37,39], the possible synergy between $V_O$ defects and guest elements would efficiently regulate the OER activity of Ru centers, which is not yet fully understood[40–42].

From a practical perspective, in addition to the intrinsic activity, the number of active sites is also important for improving OER performance. The number of active sites can be enhanced by increasing the surface area-to-mass ratio of catalysts via morphology engineering[43]. $RuO_2$-based materials with high aspect ratio morphologies demonstrate excellent activity for OER in acidic media[32,44,45]. In order to improve the stability of $RuO_2$-based OER catalysts, direct construction of high aspect ratio $RuO_2$ nanoarrays on conductive/corrosion resistant substrates is a preferred strategy. Additionally, close contact between the catalyst and substrate can also reduce interfacial charge transport resistance and facilitate the electron transfer for more efficient OER[46].

Inspired by the structural advantages of dimensionally stable anodes (DSA)[47], we herein synthesized Zn-doped $RuO_2$ (py-$RuO_2$:Zn) nanowire arrays on Ti substrate using a simple pyrolysis method. The developed py-$RuO_2$:Zn catalyst offered outstanding catalytic activity and stability for OER in acidic media (0.5 M $H_2SO_4$). $RuO_2$ doping by $Zn^{2+}$ ions promoted the growth of nanowires (thereby increasing the availability of Ru active sites for OER), whilst also introducing $V_O$ defects and low-valent Ru sites. Theoretic investigations revealed that $V_O$ defects and Zn dopants caused a weakened binding of oxygen adsorbates at active Ru centers and, more interestingly, enable a moderate adsorption of *OH species on Zn sites. Consequently, a Ru–Zn dual-site oxide path of OER was favored and significantly enhanced the OER activity. In the meantime, the alternation of OER path avoided the over oxidation of the active metal centers, and the presence of Zn dopants and $V_O$ defects enabled a structure stabilization of $RuO_2$ matrix. As a result, the py-$RuO_2$:Zn nanowires exhibited low overpotentials for OER at current densities up to 1000 mA cm$^{-2}$, together with outstanding stability reaching 1000 h at 10 mA cm$^{-2}$, outperforming commercial $RuO_2$ and most recently reported $RuO_2$-based catalysts.

## Results and discussion
### Preparation and characterization of py-$RuO_2$:Zn
The py-$RuO_2$:Zn nanowire arrays were fabricated by a straightforward pyrolysis method directly on a Ti plate (Fig. 1a), similar to the DSA

production in industrial applications[47]. In brief, a certain volume of aqueous solution containing $RuCl_3$ and $Zn(NO_3)_2$ precursors (Zn/Ru atomic ratio = 0.5:1) was pipetted onto a freshly etched Ti plate over a confined rectangular area. After dried naturally at room temperature, the sample was then pyrolyzed at 350 °C in air to transform the precursors into metal oxides. Finally, the undesired ZnO component in the product was removed by an acid etching treatment. The derived py-$RuO_2$:Zn product appeared as a dark gray coating tightly adhered to the Ti substrate (Supplementary Fig. 1). Inductively coupled plasma mass spectrometry (ICP-MS) results reveal that about 10% of Ru and 90% of Zn were removed after the acid etching treatment, leading to a decrease in the Zn/Ru atomic ratio from 54.6% to 6.39% (Supplementary Table 1). This change was confirmed by an energy-dispersive X-ray spectroscopy (EDS) analysis, which shows a similar Zn/Ru decrease from 56.6% to 5.15%, with both elements uniformly dispersing in the etched coating (Supplementary Figs. 2–3). Mass loadings of Ru and Zn in the acid-etched py-$RuO_2$:Zn coating are calculated to be 520.0 and 21.5 μg cm$^{-2}$, respectively, according to the ICP-MS results.

Figure 1b shows a grazing incidence X-ray diffraction (GIXRD, incident angle 0.3°) pattern of the py-$RuO_2$:Zn catalyst, as well as a pattern of the py-$RuO_2$ catalyst that was prepared following the same pyrolysis method in the absence of the zinc precursor. The diffraction peaks of py-$RuO_2$:Zn were almost the same as those of py-$RuO_2$, well matching the database pattern of rutile $RuO_2$ (JCPDS no. 43-1027). No peaks for ZnO were found. According to previous reports, $Zn^{2+}$ prefers to substitutionally dope $RuO_2$ at $Ru^{4+}$ sites[30,48], which is readily understood by the similar ionic radius of $Ru^{4+}$ (0.62 Å) and $Zn^{2+}$ (0.60 Å)[49]. Zn-doped $RuO_2$ retains the rutile structure of pristine $RuO_2$[48], with negligible shift in the XRD peaks seen for py-$RuO_2$:Zn due to the low Zn content (<5 at.%). However, the diffraction peaks for py-$RuO_2$:Zn were much broader than those of py-$RuO_2$ (Fig. 1b), suggesting a decrease in the grain size resulting from Zn doping. The morphology of py-$RuO_2$:Zn was next characterized by the scanning electron microscopy (SEM), with the analysis revealing a thin adherent coating with abundant microscale cracks from the precursor drying and pyrolysis steps (Fig. 1c and Supplementary Fig. 4). The coating contains dense arrays of high-quality nanowires with an average length of about 100 nm and a square cross-section along the growth direction (Fig. 1c (inset) and d–f). Control experiments revealed that the morphology of the nanowire arrays depended greatly on the composition and dosage of precursor solution and the pyrolysis temperature (Supplementary Figs. 5–7). In addition, the py-$RuO_2$:Zn nanowire arrays could readily be fabricated on other substrates, such as carbon fiber paper (CFP) and fluorine-doped tin oxide (FTO) glass (Supplementary Fig. 8), highlighting the versatility of one-step pyrolysis catalyst fabrication strategy developed herein[50–54]. But both CFP and FTO supported py-$RuO_2$:Zn catalyst shows a relatively lower OER activity for acidic OER (Supplementary Fig. 8). Therefore, the Ti plate was selected as the support for py-$RuO_2$:Zn catalyst in this work, which is a widely used DSA material in chlorine evolution process[47].

Transmission electron microscopy (TEM) analysis revealed that the nanowires had a length around 100 nm and an average diameter of 9.7 nm, corresponding to an aspect ratio (length/diameter) of ~10 (Fig. 1g). In the high-resolution TEM (HRTEM) images, random step and kink defects were found at the edges, possibly caused by the acid etching treatment (Fig. 1h). Lattice fringes with distinct interplanar distances of 3.20 Å and 1.53 Å were seen in the HRTEM images, well matching the (110) and (002) planes of rutile $RuO_2$, respectively. This indicated that the nanowires were enclosed by {110} facets (Fig. 1i–j). The corresponding fast Fourier-transformation (FFT) electron diffraction pattern was in good agreement with the simulated one viewed along [$\bar{1}$10] zone axis, suggesting single phase character and a [001] growth direction (c-axis) in the nanowire (Fig. 1i (inset) and j). The relative spatial distribution of Ru, Zn, and O elements in a single nanowire was studied by EDS under high-angle annular dark-field

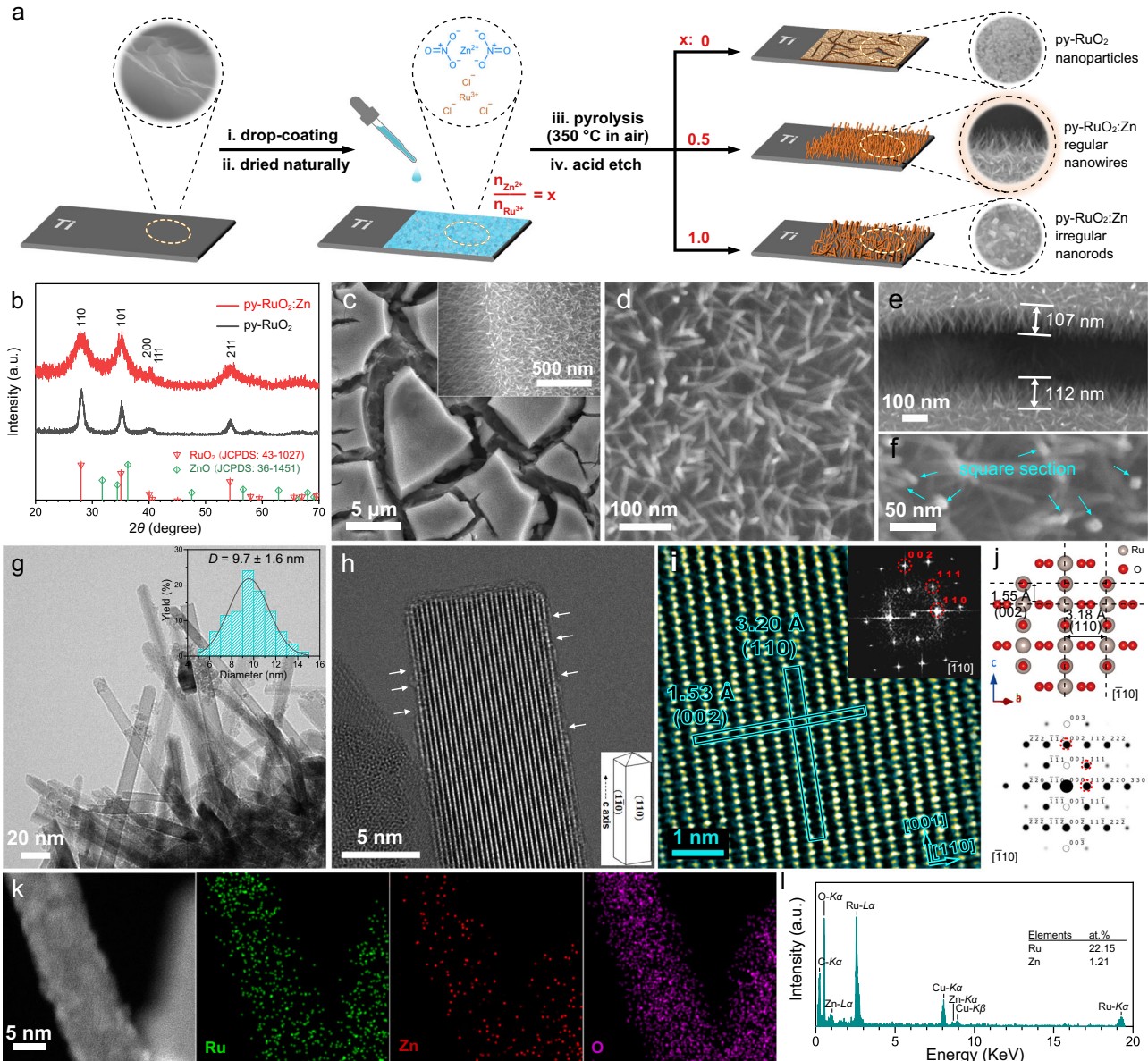

**Fig. 1 | Preparation scheme and physical characterizations of py-RuO₂:Zn.**
**a** Schematic illustration of catalyst fabrication method. **b** GIXRD patterns of py-RuO₂:Zn and py-RuO₂ catalysts. **c**, **d** SEM images of py-RuO₂:Zn catalyst. **e** length and **f** section morphology analysis of nanowires. **g** TEM image and diameter distribution analysis (inset), and **h** HRTEM image (inset: illustration of the exposed planes and growth direction) of nanowires. **i** HRTEM image and corresponding FFT pattern (inset), and **j** crystal structure simulation of the py-RuO₂:Zn catalyst. **k** HAADF-STEM images and **l** EDS analysis of Ru, Zn, and O elements in the py-RuO₂:Zn catalyst.

scanning TEM (HAADF-STEM) mode. As shown in Fig. 1k and l, each element was uniformly dispersed throughout the nanowire with a Zn/Ru atom ratio of 5.46%. This value is very close to those obtained from the ICP-MS and SEM-EDS studies (Supplementary Table 1). Interstitial Zn dopants were not seen in the atomic-resolution TEM image of a nanowire (Fig. 1i), confirming substitutional Zn doping and consistent with the XRD results.

The surface chemical information of py-RuO₂:Zn and two pure RuO₂ catalysts were next investigated by X-ray photoelectron spectroscopy (XPS). The Survey XPS spectrum confirmed the presence of Zn in py-RuO₂:Zn (Supplementary Fig. 9), while the core-level Zn 2$p$ spectrum showing peaks at 1021.4 and 1044.3 eV in a 2:1 area ratio which could readily be assigned to the Zn 2$p_{3/2}$ and Zn 2$p_{1/2}$ signals, respectively, of Zn²⁺ species (Supplementary Fig. 10)[55–57]. The Ru 3$d$ XPS spectrum for py-RuO₂:Zn showed intense peaks at ~281.0 and 285.0 eV (3:2 area ratio), which could readily be assigned to the 3$d_{5/2}$ and 3$d_{3/2}$ orbitals, respectively, of Ru⁴⁺ in RuO₂ (Supplementary Fig. 11)[32,58]. Corresponding Ru⁴⁺ shake-up satellites were seen at ~283.1 and 287.2 eV, with the C 1$s$ peak of adventitious hydrocarbons being buried under the Ru 3$d$ signal. The Ru 3$d$ peaks for py-RuO₂:Zn and py-RuO₂ were positively shifted by about 0.3−0.4 eV compared with data for the commercial RuO₂ powder catalyst (c-RuO₂), indicating a variation of the local chemical environment at Ru sites (possibly originating from particle size effects)[58,59]. Peaks in less intensity were further observed at ~282.0 and 286.2 eV, assigned to the 3$d_{5/2}$ and 3$d_{3/2}$ orbitals, respectively, of Ru³⁺ in RuO₂[58]. The abnormally positive shifts in binding energies are caused by the coordination with hydroxyl adsorbates[58,60]. In fact, non-stoichiometric Ru³⁺ species generally exist in RuO₂ films prepared by thermal decomposition of RuCl₃ precursors[61,62]. The Ru 3$p_{3/2}$ spectra for the different catalysts showed a main peak at 462.9 eV confirmed the predominance of Ru⁴⁺ species in all samples (Fig. 2a and Supplementary Fig. 12)[45,58,59]. A weak peak at

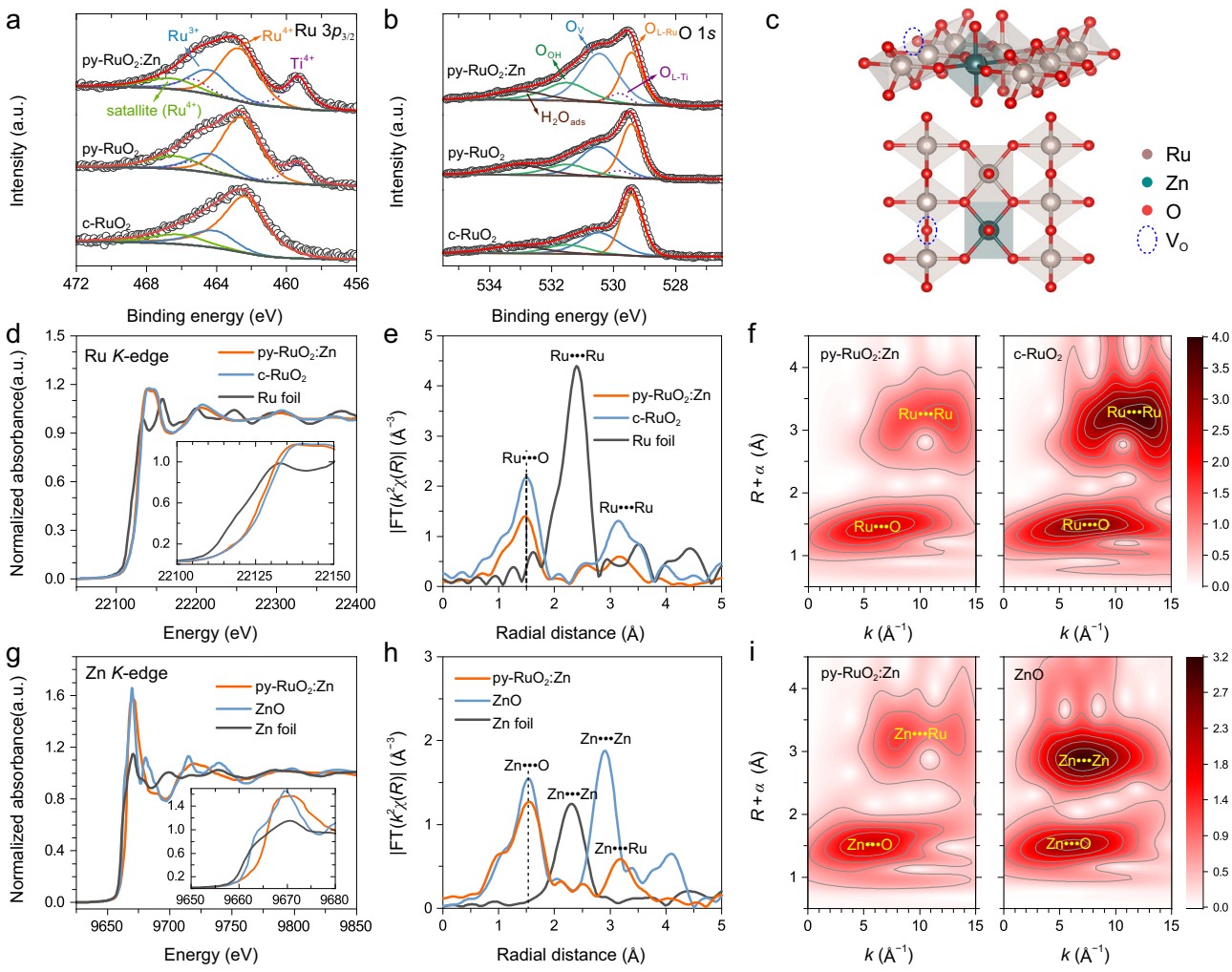

**Fig. 2 | XPS and XAS characterizations of py-RuO₂:Zn. a** Ru $3p_{3/2}$ and **b** O $1s$ core level XPS spectra for the py-RuO₂:Zn, py-RuO₂, and c-RuO₂ catalysts. **c** Simulated structure of the py-RuO₂:Zn catalyst. **d** Normalized Ru K-edge XANES and **e** Ru K-edge FT-EXAFS spectra of py-RuO₂:Zn, c-RuO₂, and Ru foil. **f** Ru K-edge WT-EXAFS spectra of py-RuO₂:Zn and c-RuO₂. **g** Normalized Zn K-edge XANES and **h** Zn K-edge FT-EXAFS spectra of py-RuO₂:Zn, ZnO, and Zn foil. **i** Zn K-edge WT-EXAFS spectra of py-RuO₂:Zn and ZnO.

464.7 eV showed the presence of some Ru³⁺ species[58,60]. The integrated area of the Ru³⁺/Ru⁴⁺ signals was then calculated to examine the relative abundance of Ru³⁺ in the different catalysts. As shown in Supplementary Table 2, the Ru³⁺/Ru⁴⁺ ratio was similar for c-RuO₂ (0.25) and py-RuO₂ (0.31), but increased considerably on going to py-RuO₂:Zn (0.55), suggesting that Zn doping increased the concentration of Ru³⁺ species. The higher content of low-valent Ru species on the surface of py-RuO₂:Zn catalyst remained under the OER conditions, as confirmed by the Raman measurements (Supplementary Fig. 13). Since the TEM data in Fig. 1i confirmed the presence of the rutile phase in py-RuO₂:Zn, a higher Ru³⁺ concentration suggested abundant oxygen vacancy (V$_O$) defects in the catalyst. The O $1s$ XPS spectra validated this hypothesis. As shown in Fig. 2b, the O $1s$ spectra showed peaks below 530.0 eV due to lattice oxygen of RuO₂ (O$_{L–Ru}$) and TiO₂ (O$_{L–Ti}$, from the thin oxide layer on the Ti substrate in cracked areas of the film), and the peaks at 530.5, 531.5, and 533.0 eV due to O atoms in the vicinity of V$_O$ defect (O$_V$), chemisorbed hydroxyl groups (O$_{OH}$), and surface-adsorbed H₂O (H₂O$_{ads}$), respectively[63–65]. Clearly, the intensity of O$_V$ peak increased on going from pure c-RuO₂ to py-RuO₂:Zn. The O$_V$/O$_{L–Ru}$ ratio on py-RuO₂:Zn (1.63) was more than twice that of c-RuO₂ (0.67) (Supplementary Table 2). Good linear relationships were further recognized between the concentrations of O$_V$ and O$_{OH}$ and the abundance of Ru³⁺ species (Supplementary Fig. 14), again proving the change in the

coordination of Ru sites in the catalyst. In summary, the XPS results reveal that Zn doping caused more low-valent Ru³⁺ species and V$_O$ defects in the RuO₂ structure, both of which would impact the activity and durability of RuO₂-based catalysts for acidic OER[33,42].

XPS probes the top few nanometers in materials. To gain more comprehensive insights about the bulk electronic structure of the py-RuO₂:Zn catalysts, we carried out X-ray absorption spectroscopy (XAS) measurements at the Ru K-edge and Zn K-edge. Figure 2d shows Ru K-edge X-ray absorption near-edge structure (XANES) spectra for py-RuO₂:Zn, Ru metal foil, and pure RuO₂ powder. The absorption edge positions for py-RuO₂:Zn and RuO₂ were at higher energy compared to that of the Ru foils, reflecting the higher oxidation state of Ru in the oxide materials. The absorption edge of py-RuO₂:Zn was at slightly lower energy than for the c-RuO₂, suggesting a slightly lower Ru valence state in py-RuO₂:Zn. Calculations on basis of adsorption edge energy revealed that an average oxidation state of Ru species in the catalyst was approximately +3.4 (Supplementary Fig. 15), which was considered as the combination of pristine Ru⁴⁺ and Ru³⁺ cations. The Zn K-edge XANES spectra for py-RuO₂:Zn, ZnO powder, and Zn foil are shown in Fig. 2g. The spectrum for py-RuO₂:Zn was quite distinct to those of the references samples, revealing a Zn²⁺ oxidation state but without the fine structure associated with ZnO. The "white line" feature of py-RuO₂:Zn was considerably broader than that of the bulk ZnO

reference and did not show the characteristic ZnO shoulder at -9663 eV[66,67]. The results indicate that the coordination of $Zn^{2+}$ atoms in py-$RuO_2$:Zn was different to the tetrahedral Zn−O coordination found in wurtzite ZnO, with the obvious explanation being the adoption of an octahedral structure through substitutional doping of Zn at Ru sites in the $RuO_2$ lattice[66,67].

We note that when the Zn dopants took an octahedral coordination structure through substitutionally doping at Ru sites in the $RuO_2$ lattice, a fraction of the Ru will, in principle, be oxidized above +4 to accommodate the divalent metal, associated with a generation of stoichiometric oxide[30]. However, when oxygen vacancies ($V_O$) present, the oxidation state of $Ru^{n+}$ ($n > 4$) would be reduced. Recently, Liu and colleagues reported a Na-doped amorphous/crystalline $RuO_2$ catalyst containing more low-valent $Ru^{n+}$ ($n < 4$) species with the presence of high abundant $V_O$ defects[40]. To further understand the role of Zn doping on the generation of $V_O$ defects, the relationship between Zn content and $V_O$ concentration was analyzed on basis of XPS results. A linear dependence of $O_V/O_{L-Ru}$ on the Zn content was found (Supplementary Fig. 16), indicating that the doping of Zn element can induce the generation of $V_O$ defects. In the meantime, the presence of $Ru^{3+}$ and $V_O$ defects was also found in the undoped py-$RuO_2$ catalyst, seemly caused by the catalyst synthesis method used here[61,62]. Thus, it can conclude that the Zn doping, in addition to the catalysis synthesis method, has induced the generation of $V_O$ defects and the low-valent Ru sites.

Figure 2e shows Fourier transformed (FT) $k^2$-weighted Ru $K$-edge extended X-ray absorption fine structure (EXAFS) spectra for py-$RuO_2$:Zn and relevant reference samples. The main peaks at 1.50 and 3.14 Å for pure $RuO_2$ correspond to the first Ru−O and Ru−Ru coordination shells of Ru cations[31,68], respectively. For py-$RuO_2$:Zn, these sample features were observed at 1.47 and 3.17 Å, respectively, indicating a slight change in Ru cation coordination environment with Zn doping. Compared with $RuO_2$, py-$RuO_2$:Zn showed reduced intensities for both Ru−O and Ru−Ru peaks, suggesting that the coordination number of Ru sites was decreased[69], consistent with the presence of $V_O$ defects. Further, the substitutionally doping of Zn would make the second peak a mixture of Ru−Ru and Ru−Zn scattering. In addition, the Zn $K$-edge EXAFS spectrum of py-$RuO_2$:Zn in Fig. 2h closely resembled the Ru $K$-edge spectrum, suggesting an octahedral-like Zn coordination (Supplementary Fig. 17). The first peak observed at 1.55 Å in the Zn $K$-edge $R$-space plot for py-$RuO_2$:Zn, assigned to the first Zn−O coordination shell, was longer than the 1.47 Å for Ru−O bonds (as expected since $Zn^{2+}$ has a lower charge than $Ru^{4+}$). The second strong peak at 3.18 Å was longer than the Zn−Zn shell distance in ZnO (2.91 Å), being more comparable to the Ru−Ru distance (3.17 Å) in py-$RuO_2$:Zn or c-$RuO_2$ (3.14 Å). Previously, Petrykin and colleagues reported similar Zn $K$-edge EXAFS spectra for $Ru_{1−x}Zn_xO_2$ ($x ≤ 0.2$) materials and assigned the peak located at ~3.1 Å to Zn−Ru backscattering at Zn sites based on a structural fitting analysis[48]. We believe, the peak at 3.18 Å in the Zn $K$-edge spectrum of py-$RuO_2$:Zn has the same origin, arising from substitution of Ru ions by Zn ions in the py-$RuO_2$:Zn catalyst. The wavelet transform (WT) EXAFS measurements provided further confirmation for this assignment (Fig. 2f and i, and Supplementary Fig. 18). The maximum-intensity Ru $K$-edge values for py-$RuO_2$:Zn were observed at $k ≈ 6.5$ and 12.5 Å$^{−1}$, attributed to Ru−O and Ru−Ru/Zn scattering paths (Fig. 2f), respectively. These features were weaker than those of the reference $RuO_2$ sample, which may have been due to the nanosize of the py-$RuO_2$:Zn nanowires and also the mixed Ru−Ru/Zn coordination shell. The Zn $K$-edge WT plot of py-$RuO_2$:Zn in Fig. 2i showed a similar contour profile to the Ru $K$-edge plot in Fig. 2f. On basis of the observations, the model crystal structure for py-$RuO_2$:Zn could be proposed (Fig. 2c) based on the rutile structure of pure $RuO_2$ with partial substitution of Ru atoms by Zn atoms. The introduction of $Zn^{2+}$ ions promotes the formation of oxygen vacancies in the near vicinity. The loss of O at the vertex of the $RuO_6$ octahedra would lower

the average Ru valence, which has particular relevance to the OER performance[33,68].

## Electrocatalytic performance of py-$RuO_2$:Zn toward acidic OER

The OER activity of the as-prepared py-$RuO_2$:Zn catalyst was examined in an $O_2$-saturated 0.5 M $H_2SO_4$ electrolyte using a conventional three-electrode set-up. The Ag/AgCl reference electrode was first calibrated against the reversible hydrogen electrode (RHE) (Supplementary Fig. 19). For comparison, the activities of py-$RuO_2$ and c-$RuO_2$ were also measured under identical conditions. In order to minimize the background capacitive current, the linear sweep voltammetry (LSV) curves reported were obtained by taking the average results of the positive/negative-going scans of a cyclic voltammetry curve (CV) (Supplementary Fig. 20a). The capacitance-corrected LSV curve was then performed an 85% $iR$-compensation correction (Supplementary Fig. 20b)[70]. The CV curves shows no obvious degradation during the first 30 cycles on py-$RuO_2$:Zn (Supplementary Fig. 21), implying that the pristine surface offered high OER activity without the need for pre-activation treatment[71]. Figure 3a displays the LSV results of OER on the different catalysts. The sharply rising anodic current related to the OER process appeared at more negative potentials on py-$RuO_2$:Zn compared to pure c-$RuO_2$. The associated OER onset potential was -1.33 V (vs RHE), corresponding to an overpotential ($\eta$) of -100 mV, much lower than those of py-$RuO_2$ (-1.38 V, $\eta ≈ 150$ mV) and c-$RuO_2$ (-1.42 V, $\eta ≈ 190$ mV) (Supplementary Fig. 22)[72]. Accordingly, the OER process is a more easily triggered on py-$RuO_2$:Zn. The superior OER activity of py-$RuO_2$:Zn was retained on increasing the current density. To achieve a current density of 10 mA cm$^{−2}$, py-$RuO_2$:Zn required a low potential of 1.403 V ($\eta = 173$ mV), outperforming py-$RuO_2$ (1.458 V, $\eta = 228$ mV) and the commercial c-$RuO_2$ reference catalyst (1.521 V, $\eta = 291$ mV)[40,73]. At a higher overpotential of $\eta = 300$ mV, py-$RuO_2$:Zn achieves a current density of 476 mA cm$^{−2}$, which was 4.4 and 36.1 times as larger than values for py-$RuO_2$ and c-$RuO_2$, respectively (Fig. 3c). Moreover, the OER process can be polarized to an industrial current density of 1.0 A cm$^{−2}$ on py-$RuO_2$:Zn catalyst, operating at a very competitive potential of 1.603 V ($\eta = 373$ mV) (Fig. 3b)[33,73]. Such a large current density can be reached more than five continuous CV cycles, but accompanied by a gradually degradation in the OER activity (Supplementary Fig. 23). The faradaic efficiency (FE) of OER on py-$RuO_2$:Zn catalyst was measured by the water displacement method under the chronopotentiometric condition at current densities of 25 and 40 mA cm$^{−2}$. As shown in Supplementary Fig. 24, the measured oxygen amount fits well with the theoretical values calculated from Faraday's law of electrolysis, approaching -99% and -100% FE at 25 and 40 mA cm$^{−2}$, respectively. Notably, the OER activity of py-$RuO_2$:Zn was greatly affected by the conditions of catalyst preparation (Supplementary Figs. 5-7), with the optimal OER performance being achieved with a regular nanowire morphology on Ti plate.

To evaluate the intrinsic OER performance of the py-$RuO_2$:Zn catalyst, we further calculated the mass activity according to the total loading of Ru metal determined by ICP-MS (Supplementary Table 1). As shown in Fig. 3a, the OER mass activity of the py-$RuO_2$:Zn catalyst greatly surpassed those of py-$RuO_2$ and c-$RuO_2$. A current density of 100 mA mg$_{Ru}^{−1}$ can be achieved at a low potential of 1.442 V ($\eta = 212$ mV) on py-$RuO_2$:Zn, whereas 1.508 V ($\eta = 278$ mV) and 1.607 V ($\eta = 377$ mV) were required on py-$RuO_2$ and c-$RuO_2$, respectively. At $\eta = 300$ mV, py-$RuO_2$:Zn delivered a current density up to 881 mA mg$_{Ru}^{−1}$ (Fig. 3c). In contrast, just 181 mA mg$_{Ru}^{−1}$ was realized on py-$RuO_2$ and 22.0 mA mg$_{Ru}^{−1}$ on c-$RuO_2$[73]. The remarkable OER activity of the py-$RuO_2$:Zn catalyst well retained at high current densities, evidenced by values of 1.538 V@1.0 A mg$_{Ru}^{−1}$ and 1.611 V@2.0 A mg$_{Ru}^{−1}$ (Fig. 3b). It is also worth noting that the OER activity of py-$RuO_2$:Zn greatly surpassed those of reported Zn-doped $RuO_2$ catalysts[30,48,74]. A rutile-type $Zn_{0.19}Ru_{0.81}O_2$ was previously studied by Burnett and colleagues[30]. Although the $Zn_{0.19}Ru_{0.81}O_2$ catalyst reported in that work displayed an OER activity

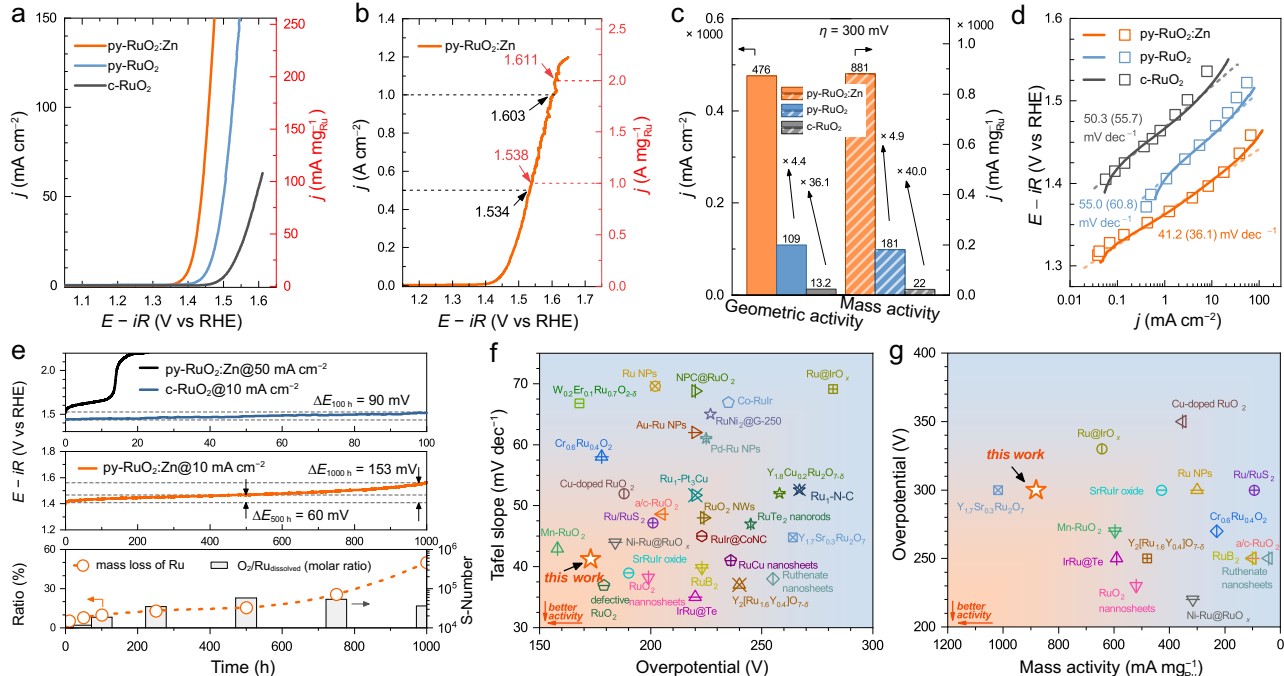

**Fig. 3 | OER performance of py-RuO₂:Zn in acidic media. a** Geometric area and Ru mass normalized LSV curves with 85% *iR*-correction of py-RuO₂:Zn, py-RuO₂, and c-RuO₂ for OER in 0.5 M H₂SO₄ solution (pH = 0.30 ± 0.01) with O₂ saturation. Solution resistances for *iR*-correction are 2.8, 2.6, and 4.5 Ω for py-RuO₂:Zn, py-RuO₂, and c-RuO₂, respectively. Mass loadings of Ru metal are 0.52, 0.60, and 0.60 mg cm⁻² for py-RuO₂:Zn, py-RuO₂, and c-RuO₂, respectively. **b** Geometric area and Ru mass normalized LSV curve of py-RuO₂:Zn for OER under high current density. **c** Comparisons of OER geometric and mass activities at an overpotential of 300 mV on py-RuO₂:Zn, py-RuO₂, and c-RuO₂. **d** Tafel plots derived from the LSV curves (solid line) and the steady-state polarization curves (scatters). Values in parentheses were derived from steady-state polarization curves. **e** Chronopotentiometric stability tests of py-RuO₂:Zn and c-RuO₂ (upper plot: 100 h at 50 mA cm⁻²; middle plot: 1000 h at 10 mA cm⁻²) and mass loss analysis of Ru and corresponding stability number (S-Number) on py-RuO₂:Zn during the stability test determined by ICP-MS (lower plot). Comparison of overpotentials and **f** Tafel slopes, and **g** mass activities for py-RuO₂:Zn and other recently reported high performance RuO₂-based OER catalysts.

better than commercial RuO₂, its performance was vastly inferior to the py-RuO₂:Zn catalyst in the current study, with OER activity at η = 300 mV only reached at 60 mA mg⁻¹ᴿᵘ. Actually, Zn₀.₁₉Ru₀.₈₁O₂ was reported to possess a defect-free stoichiometric oxide. The fully occupied oxygen sites were proposed to require a higher average Ru oxidation state (above +4) to balance charge, which is obviously different to the structure of the defective py-RuO₂:Zn catalyst. The difference in crystal structure of the Zn-doped RuO₂ catalysts explains the variation in OER performance between our work and that of Burnett and colleagues[30]. Recently, a surface evolution of Zn-doped RuO₂ under the reaction was found to enable a construction of surface defects (e.g., V_O defects) and active Ru sites[75], consistent with the theoretically predicted results on RuO₂ catalyst[39]. A low overpotential of 190 mV and a good stability up to 60 h were observed at the current density of 10 mA cm⁻² on this surface etched catalyst. Next, the electrochemical active surface area (ECSA) was calculated for the different catalysts and used to normalize the OER current, in order to eliminate the effect of catalyst morphology. As shown in Supplementary Fig. 25, py-RuO₂:Zn possessed a much larger ECSA and a higher specific OER activity compared to the pure RuO₂ catalysts studied in this work, largely due to the significant difference in the morphology of them (Supplementary Fig. 5). Figure 3d shows the Tafel slope analyses for the different catalysts. The plots were derived from the *iR*-corrected LSV curves and the steady-state polarization curves (Supplementary Fig. 26)[76]. Clearly, py-RuO₂:Zn offered the lowest Tafel slope of 41.2 (36.1) mV dec⁻¹, suggesting faster OER kinetics compared to the py-RuO₂ and c-RuO₂ catalysts[25,73]. A Tafel slope around 40 mV dec⁻¹ implies a better kinetics of the OH_ads deprotonation to form O_ads and the O−O bond formation[8,77]. Moreover, electrochemical impedance spectroscopy (EIS) results (Supplementary Figs. 27−28, and

Supplementary Table 3) showed that the charge transfer resistance (R_ct) was significantly smaller on py-RuO₂:Zn than it on pure py-RuO₂, for instance, 9.0 Ω and 114.7 Ω at 1.40 V, respectively, further proving a much faster charge transfer rate of OER and thereby an improved reaction kinetics on py-RuO₂:Zn. In summary, the py-RuO₂:Zn catalyst demonstrated excellent OER activity compared to the pure RuO₂ reference catalysts and state-of-the-art performance compared to RuO₂-based acidic OER catalysts recently reported (Fig. 3f, g, and Supplementary Table 4).

Next, catalytic stability of py-RuO₂:Zn during OER was investigated using a chronopotentiometric (CP) method at a constant current density. As shown in Fig. 3e, py-RuO₂:Zn displayed far better stability than the c-RuO₂ catalyst. At a typical current density of 10 mA cm⁻², the OER potential on py-RuO₂:Zn increased by only 60 mV during the initial 500 h of testing and by only 153 mV over 1000 h of testing. In contrast, the c-RuO₂ catalyst dramatically lost activity over 15 h under identical conditions[41,73]. At a higher current density of 50 mA cm⁻², py-RuO₂:Zn showed excellent stability over 100 h with an overpotential increase of only 90 mV (Fig. 3e), while potential increase was 70 mV after a test at the current density of 100 mA cm⁻² for 24 h (Supplementary Fig. 29). The good stability of py-RuO₂:Zn was further investigated under the CV cycling condition. The potential at 100 mA cm⁻² was increased by about 28 mV after a 2000-cycles test (Supplementary Fig. 30). Compared with recently reported RuO₂-based catalysts, the stability of py-RuO₂:Zn was also more distinguished (Supplementary Table 4). For example, the degradation of the OER overpotential (ΔE) at 10 mA cm⁻² for py-RuO₂:Zn was much smaller than that reported for the best Ru/α-MnO₂ (ΔE = 169 mV@200 h) and Li_xRuO₂ (ΔE = 120 mV@70 h) catalysts under similar testing conditions[35,41]. During the stability test, the dissolution of Ru from py-RuO₂:Zn catalyst was

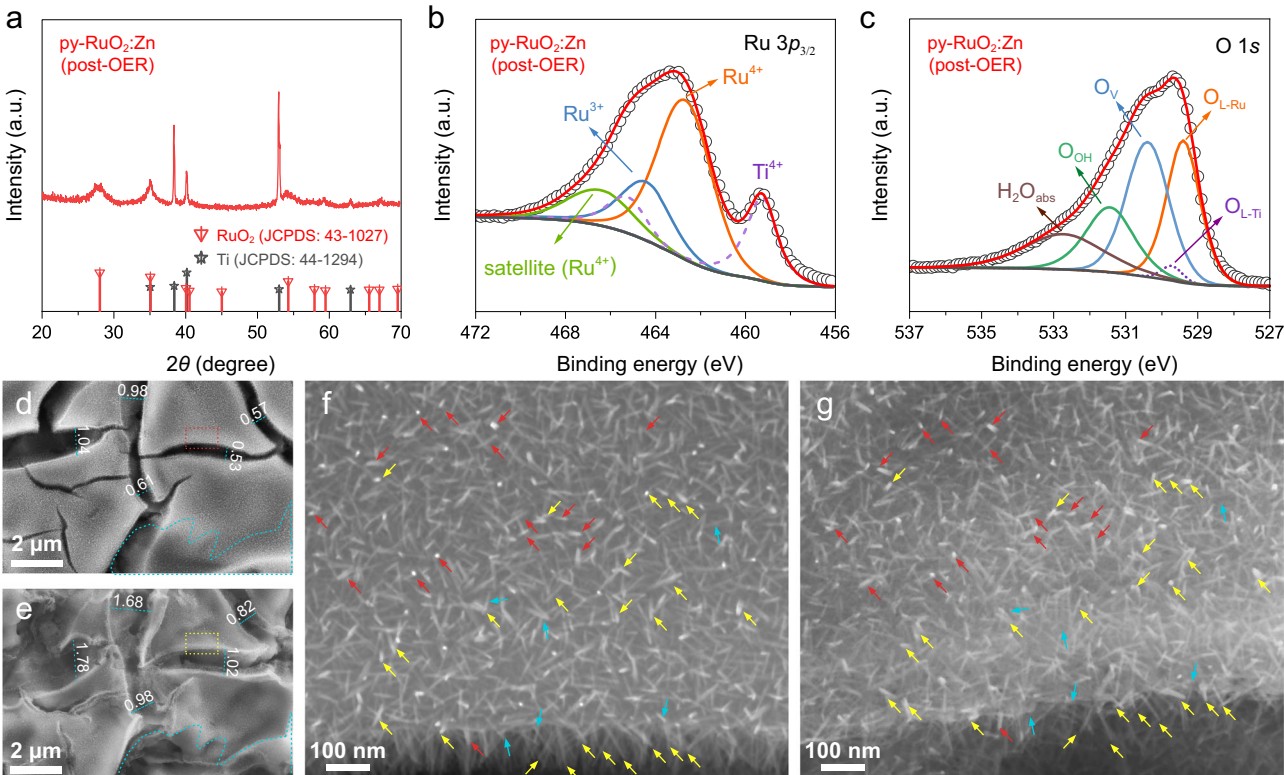

**Fig. 4 | Stability analysis of py-RuO$_2$:Zn for acidic OER. a** XRD pattern, **b** Ru 3$p_{3/2}$ and **c** O 1$s$ XPS spectra of py-RuO$_2$:Zn catalyst post OER test. Quasi-in situ SEM images of the py-RuO$_2$:Zn catalyst **d**, **f** before and **e**, **g** after a continuous OER test at 10 mA cm$^{-2}$ for 350 h. The red and yellow arrows indicate nanowires that retained or slightly lost their morphologies during OER, respectively, whilst the cyan arrows indicate nanowires that completely disappeared after the test.

determined by ICP-MS. Figure 3e (lower plot) shows the mass loss of Ru normalized against the initial Ru loading in freshly prepared py-RuO$_2$:Zn. Three distinct stages were seen in the Ru dissolution profile. During the initial 100 h, ~10% Ru loss occurred with the loss increasing slowly to ~15% after 500 h. In the final 1000 h, the Ru loss increased to ~50%. This trend is generally consistent with the performance degradation behavior seen for the py-RuO$_2$:Zn catalyst in the CP stability test (Fig. 3e, middle plot). The OER potential shows an increase of 28 mV, from 1.408 V to 1.436 V, in the first 100 h test, followed by a further 32 mV increase from 1.436 V to 1.468 V between 100 h and 500 h. Finally, a larger 93 mV increase, from 1.468 V to 1.561 V, was found between 500 h and 1000 h. Consequently, although the OER potential degradation of py-RuO$_2$:Zn was very modest over 1000 h (compared to previously reported RuO$_2$ catalysts in acidic media), the ~50% Ru mass loss at the end of the tests indicated serious corrosion in the latter stages, which was then confirmed by post-stability test SEM imaging and optical photographs (Supplementary Fig. 31). In addition, stability number (S-number), a recommended metric to quantify the catalyst stability during the reaction[78], was calculated by normalizing the moles of O$_2$ evolved ($n_{O_2 \text{evolved}}$) with the moles of Ru dissolved ($n_{Ru \text{ dissolved}}$), i.e., S-number = $n_{O_2 \text{evolved}}/n_{Ru \text{ dissolved}}$[38]. As shown in Fig. 3e lower plot, the S-number exhibited an increase in the initial 500 h and then a decrease in the following 500 h. A top value of ~6 × 10$^4$ was obtained, which is comparable to those observed on Ru-based pyrochlores[38]. We also note that the mass loss of Ru up to ~15% within 500 h from the py-RuO$_2$:Zn catalyst seems to be serious for industrial applications. However, the corresponding dissolution rate of Ru, 0.156 µg cm$_{geo}^{-2}$ h$^{-1}$, is much lower than that of the commercial RuO$_2$ (~40 µg cm$_{geo}^{-2}$ h$^{-1}$). Further compared with the high active RuO$_2$-based acidic OER catalysts recently reported (Supplementary Table 5), the py-RuO$_2$:Zn catalyst also ranks the top-level of stability in terms of the Ru dissolution rate. When normalized by the ECSA, a value of

37.6 pg cm$_{ECSA}^{-2}$ h$^{-1}$ for Ru dissolution rate was obtained on the py-RuO$_2$:Zn catalyst, significantly lower than the ~1.05 µg cm$_{ECSA}^{-2}$ h$^{-1}$ on commercial RuO$_2$, indicating an intrinsically improved stability of the catalyst.

In order to gain deeper insights about py-RuO$_2$:Zn catalyst degradation during acidic OER, we performed a further durability test at 10 mA cm$^{-2}$ for 350 h (Supplementary Fig. 32). The 350 h was selected on basis of the apparent inflection point in the Ru dissolution curve (Fig. 3e, lower plot). XRD revealed that the py-RuO$_2$:Zn catalyst retained a rutile structure after 350 h (Fig. 4a). The core level Ru 3$d$, Ru 3$p_{3/2}$, and O 1$s$ XPS spectra showed a slight decrease in the concentrations of Ru$^{3+}$ species and V$_O$ defects on the surface (Fig. 4b, c, Supplementary Fig. 33, and Supplementary Table 2). No obvious change was found in the Zn 2$p$ XPS spectra (Supplementary Fig. 34). Quasi-in situ SEM measurements taken at some pre-marked locations before and after the OER stability test were used to study Ru dissolution from the py-RuO$_2$:Zn catalyst. Some corrosion was observed in the catalyst coating on the Ti plate after the test (Supplementary Fig. 35), accompanied by an expansion of the original coating cracks (Fig. 4d–e). The corrosion appears to begin preferentially at the edges of the cracks and then gradually expand into the plateau domains. A close comparison (marked by arrows) revealed that the majority of py-RuO$_2$:Zn nanowires retained their original locations and morphologies (marked by red arrows), especially those far from the cracks. Nanowires near the cracks showed more obvious changes in their spatial directions and morphologies (marked by yellow arrows). A few nanowires disappeared completely after the 350 h of testing (marked by cyan arrows). Optical images (Supplementary Fig. 31c) revealed that the py-RuO$_2$:Zn coating remained in a good condition after the 350 h test, showing good adhesion and a uniform dispersion of elements (Supplementary Fig. 36), which is consistent with a relatively slow mass loss of Ru during the first 500 h of OER testing (Fig. 3e, lower plot).

Results suggest that there is likely a threshold potential that determines the dissolution rate of Ru in the py-RuO$_2$:Zn catalyst during OER, above which dissolution proceeds very rapidly. Based on the CP results, this threshold potential appears to be -1.46 V (Supplementary Fig. 37). Anodic polarization higher than 1.46 V will result in accelerated corrosion of the py-RuO$_2$:Zn catalyst. The accelerated degradation of py-RuO$_2$:Zn at potentials above 1.46 V was further observed under a CP test at 100 mA cm$^{-2}$, which exhibited a faster increase of overpotential by 70 mV within 24 h (Supplementary Fig. 29). The result agrees with the previous reports on the stability window of RuO$_2$-based catalysts[6,79,80]. Although the stability of py-RuO$_2$:Zn did not obviously break the reported potential limit, the onset overpotential of OER was significantly reduced, providing a widened stability window to the application of py-RuO$_2$:Zn. Furthermore, we find that the potential of 1.46 V is close to the inflection region in the Tafel plot (Supplementary Fig. 38), indicating a change in the rate-determining step of OER with the change in Ru dissolution rate[8,77].

### Insights into OER process and relevant mechanism

On the LSV curve for OER (Fig. 3a and Supplementary Fig. 22), low onset potential (-1.33 V) and overpotential (173 mV at 10 mA cm$^{-2}$) were observed and have been assigned to an anodic OER process on the py-RuO$_2$:Zn catalyst. Such low threshold potentials are impressive because they well exceeded the theoretical limit of OER onset overpotential (-250 mV) on the optimal catalyst, based on the adsorbate evolution mechanism (AEM) involving single active metal site and the linear scaling relationships between the adsorption energies of *O, *OH, and *OOH intermediates ($\Delta E_{OOH} = \Delta E_{OH} + 3.2$ eV $\pm$ 0.2 eV)[36,81]. We then performed experiments using a rotating ring-disk electrode (RRDE) setup and confirmed the explicit contribution of OER process to the observed anodic current at potentials around 1.40 V (Supplementary Fig. 39). Thus, the low threshold potentials of OER suggested that there may be other paths of OER on the py-RuO$_2$:Zn catalyst in addition to the AEM, especially at low overpotentials. Recently, Scott and colleagues performed a trace detection of O$_2$ and found an electrochemical generation of O$_2$ from OER on the RuO$_x$ catalyst at the potential as low as 1.30 V[82]. Further by comparing the trends in Ru dissolution and oxygen evolution, they suggested a negligible contribution of lattice oxygen evolution to the overall OER activity for RuO$_x$ in acidic media[22]. A comprehensive theoretical study on the recently reported mechanisms of OER revealed that the presence of nonelectrochemical steps (e.g., *OO dimer formation/desorption) tends to increase rather than to reduce the thermodynamic overpotential of OER, while the presence of surface defects (e.g., V$_O$ defects) probably alters the configuration of adsorbed intermediates to improve the OER activity[83].

In this work, a high concentration of V$_O$ defects and low-valent Ru species existed in the py-RuO$_2$:Zn catalyst, which may play important roles in improving the OER property[31,41], in addition to the catalyst electrical conductivity (Supplementary Fig. 28 and Supplementary Table 3)[84]. When plotting specific current densities against the V$_O$ concentrations, a good linear relationship was established, revealing a clear impact of V$_O$ defects on the OER activity (Supplementary Fig. 40). However, the lower oxidation state of Ru sites and higher concentration of V$_O$ defects were expected to result in much stronger *OH adsorption and be detrimental to the OER activity of RuO$_2$-based catalysts, based on the linear scaling relationships between the adsorbates binding energies following conventional AEM path[36,81]. Accordingly, enhancement on OER activity was achieved when there were high-valent Ru sites and less V$_O$ defects[37,38]. This seems conflict with our result that an enhanced OER activity was obtained on V$_O$ defects containing Zn-doped RuO$_2$ catalyst. We speculated that the positive effect of V$_O$ defects on OER activity was realized with the assistance of the Zn dopants. V$_O$ defect and Zn dopants can synergistically regulate the coordinative environment and electronic structure

of vicinal Ru centers and thus optimize the binding configurations of OER intermediates[40,41,85]. Consequently, the OER activity may be improved.

To understand the Zn doping and oxygen vacancies effect on the OER activity, density functional theory (DFT) calculations were performed. The Zn doped RuO$_2$ (RuO$_2$:Zn) and that with O vacancies (RuO$_2$:Zn_V$_O$) were built on the optimized RuO$_2$ (110) surfaces (Supplementary Fig. 41). Zn was found to be more stably doped at the coordinatively unsaturated Ru (Ru$_{cus}$) position than the fully coordinated bridge Ru (Ru$_{bri}$) site, while the bridge row O could form stable vacancy site. Then, different OER paths were investigated to determine the preferred reaction pathways, including the AEM and lattice oxygen mechanism (LOM), as well as the recently highlighted dual-site oxide path mechanism (OPM) (Supplementary Fig. 42)[35,83]. The adsorption energies of reaction intermediates were summarized in the Supplementary Table 6. For clean RuO$_2$, stronger binding of OH adsorbates ($\Delta G_{OH} = 0.82$ eV) resulted in the OER proceeding favorably via a AEM path, following the four-proton-coupled electron transfer steps as H$_2$O $\rightarrow$ *OH $\rightarrow$ *O $\rightarrow$ *OOH $\rightarrow$ O$_2$[36]. The formation of *OOH is the rate-determining step (RDS) with a large free energies barrier of 2.10 eV. By comparison, the LOM and dual site OPM paths are suppressed with much higher energy barriers of RDS ($\Delta G_{max}$ for LOM 3.79 eV and OPM 2.48 eV, where $\Delta G_{max}$ is the maximum free energy differences among the primary proton-coupled electron transfer steps) (Supplementary Fig. 43). For RuO$_2$_V$_O$, the presence of bridged O vacancies caused accumulated charge density at both the vicinal Ru$_{bri}$ and Ru$_{cus}$ sites (Supplementary Fig. 44), which then enhanced the binding of *OH at Ru$_{cus}$ centers ($\Delta G_{OH} = 0.70$ eV) and induced a larger free energies barrier of 2.28 eV for *OOH formation (Supplementary Fig. 45). Therefore, the presence of V$_O$ defects is harmful to the OER proceeding on RuO$_2$[37,38]. In contrast, on the surface of stoichiometric RuO$_2$:Zn oxide, the doping of Zn at Ru$_{cus}$ sites induced a reduction of the charge density at Ru centers, which agreed with the knowledge that a fraction of the Ru will be oxidized above +4 to accommodate the divalent Zn metal[30]. As a result, the *OH binding is weakened ($\Delta G_{OH} = 1.01$ eV) and the OER activity is improved. More interestingly, a Ru−Zn dual-site OPM appeared to be more favorable with a lower $\Delta G_{max}$ of 1.91 eV for the third proton-coupled electron transfer step (*O$_{Ru}$ $\rightarrow$ *O$_{Ru}$...*OH$_{Zn}$), caused by the different binding strength of intermediates on the two sites (Supplementary Fig. 46). The density of sates (DOS) and charge density difference suggested that Zn donated some electron to the O and Zn had a lower $d$-band center than Ru (Fig. 5c−e). Therefore, Zn showed weaker absorption of *O, *OH, and *OOH. For example, Zn sites had a $\Delta G_{OH}$ of 1.77 eV, while Ru site had a $\Delta G_{OH}$ of 1.01 eV. This would ease the formation of second *O. In addition, the charge difference between Zn and Ru also played an important role in promoting the OER, which resulted in a ~ 0.1 $e$ charge difference for the two absorbed *O on Zn and Ru and thus promoted the formation of O−O coupling, and eventually the formation of O$_2$ (Fig. 5d). With the presence of V$_O$ defects, the charge density at both the Ru$_{cus}$ and Zn$_{cus}$ sites on RuO$_2$:Zn_V$_O$ surface is slightly increased (Supplementary Fig. 44), associated with a shift of Ru $d$-band center away from Fermi, which further optimized the absorption of intermediates (Fig. 5e). Consequently, the $\Delta G_{max}$ (*O$_{Ru}$ $\rightarrow$ *O$_{Ru}$...*OH$_{Zn}$) of OPM is further decreased to 1.84 eV for RuO$_2$:Zn with V$_O$ defects (Fig. 5b). Therefore, we believed that the down shift of Fermi by O vacancy, the weaker absorption of *OH on Zn and the charge difference of Zn and Ru synergistically lowered the OER overpotential ($\eta = \triangle G_{max} - 1.23$) from 0.87 V for RuO$_2$ to 0.61 V for the O vacancy-containing Zn doped RuO$_2$, by converting the OER path from the single-site AEM to the dual-site OPM (Fig. 5a, b).

In terms of the stability enhancement, the present dual-site OPM path of OER avoids the step of *O $\rightarrow$ *OOH, which generally proceeds above 1.3 V on single Ru site[86]. Thus, it was possible to stabilize the OER active sites against the excessive oxidation under the OPM path.

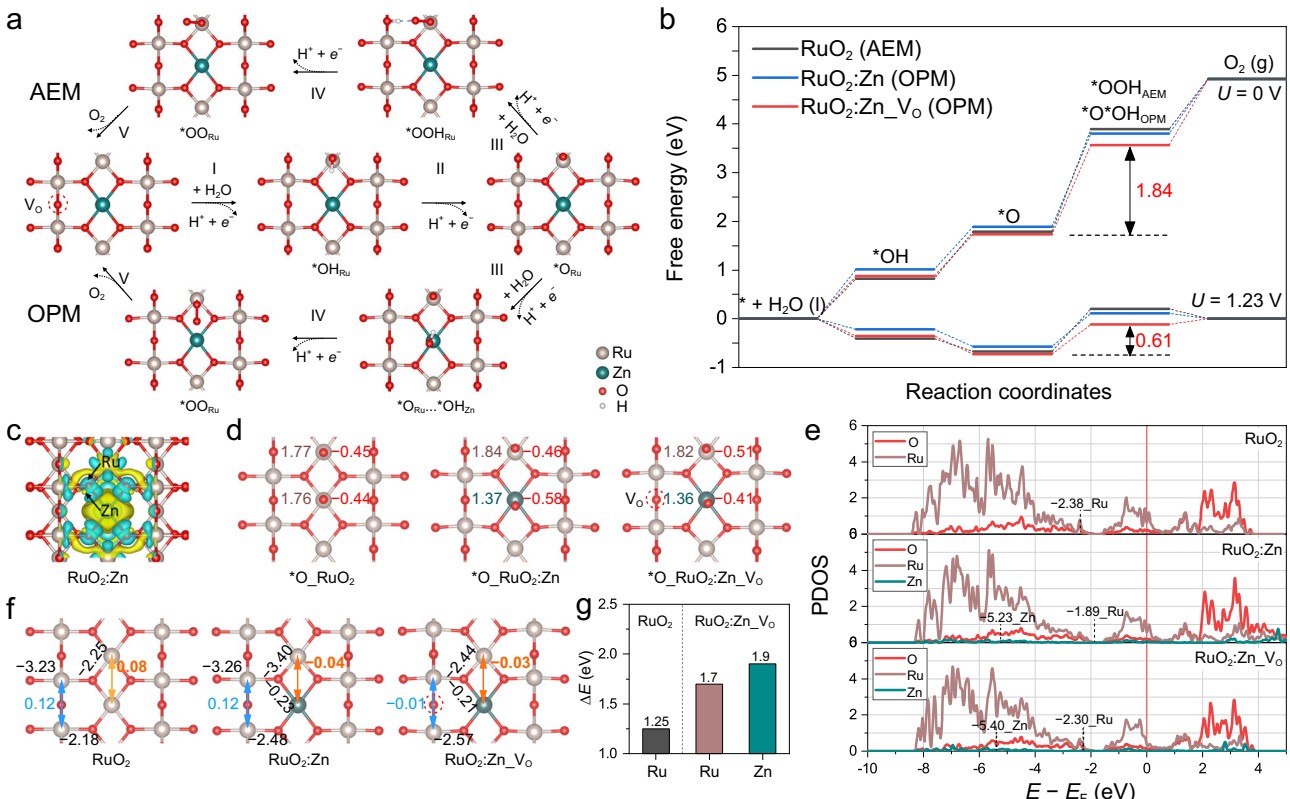

**Fig. 5 | OER mechanism analysis. a** AEM and OPM paths of OER on the RuO$_2$:Zn_V$_O$ surface. **b** The free energy diagrams for preferred OER paths on the surfaces of RuO$_2$, RuO$_2$:Zn, and RuO$_2$:Zn_V$_O$. **c** Differential charge density analysis of RuO$_2$:Zn. The blue and yellow shaded area mean the electron density accumulation and donation. **d** Bader charge analysis for Ru (brown), Zn (dark cyan), and O (red) sites on the double *O adsorbed surfaces of RuO$_2$, RuO$_2$:Zn, and RuO$_2$:Zn_V$_O$. **e** PDOS of Ru 4$d$, O 2$p$, and Zn 3$d$-bands for RuO$_2$, RuO$_2$:Zn, and RuO$_2$:Zn_V$_O$; corresponding $d$-band centers are denoted by dashed lines. **f** ICOHP analysis of Ru–O, Ru···Ru, Ru···Zn, and Zn–O on the surfaces of RuO$_2$, RuO$_2$:Zn, and RuO$_2$:Zn_V$_O$. **g** De-metallization energies of Ru from RuO$_2$, and Ru and Zn from RuO$_2$:Zn_V$_O$.

We then studied the electrochemical redox features of the Ru species on py-RuO$_2$:Zn, py-RuO$_2$, and c-RuO$_2$ catalysts in potential regions preceding OER process (Supplementary Fig. 47). Compared with those on py-RuO$_2$ and c-RuO$_2$, the redox peaks of Ru$^{4+}$/ Ru$^{n+}$ ($n > 4$) above 1.2 V were significantly suppressed on py-RuO$_2$:Zn, indicating an efficient protection on Ru cations from over oxidation to soluble species[21,87,88]. Consequently, the catalytic stability of py-RuO$_2$:Zn for OER would be enhanced. To gain more insights into the effect of Zn doping and V$_O$ defects on the structure stabilization of RuO$_2$, the crystal orbital Hamilton population (COHP) of Ru–O and Zn–O bonds, as well as Ru···Ru and Ru···Zn metal couplings, were analyzed on the optimized RuO$_2$, RuO$_2$:Zn, and RuO$_2$:Zn_V$_O$ surfaces. As shown in Fig. 5f, the integrated COHP (ICOHP) values of Ru$_{cus}$–O for RuO$_2$:Zn, and RuO$_2$:Zn_V$_O$ are −3.40 eV and −2.44 eV, respectively, which have been negatively shifted from that for pristine RuO$_2$ (−2.25 eV), thereby revealing a strengthened Ru$_{cus}$–O bond on those Zn-doped samples. In addition, small negative ICOHP values of Ru$_{cus}$···Zn were found on both the RuO$_2$:Zn (−0.04 eV) and RuO$_2$:Zn_V$_O$ (−0.03 eV) with the Zn doping, indicating a weak long range orbital coupling between Zn dopants and the vicinal Ru$_{cus}$ sites. In contrast, there is no clear interaction of Ru$_{cus}$···Ru$_{cus}$ (0.08 eV for ICOHP) on the pristine RuO$_2$. Accordingly, the Ru$_{cus}$ sites would be further stabilized by the Zn dopants. When bridged V$_O$ defects present, the ICOHP of Ru$_{bri}$···Ru$_{bri}$ for RuO$_2$:Zn_V$_O$ also acquired a small negative value of −0.01 eV, while it was a positive value of 0.12 eV on both the RuO$_2$ and RuO$_2$:Zn. This indicated an enhanced interaction between two adjacent Ru$_{bri}$ sites in the vicinity of V$_O$ defect. The enhanced stability of Zn doped RuO$_2$ with Vo is also demonstrated by the de-metallization energies of Ru and Zn (Fig. 5g). The doping of Zn induced an increased de-metallization energy of Ru

by around 0.5 eV and thus stabilized the RuO$_2$. The Zn dopants themselves possessed relatively higher de-metallization energies by around 0.2 eV than the Ru in RuO$_2$:Zn_V$_O$. The overall results suggested that the RuO$_2$ structure become more stable after the introduction of Zn dopants and V$_O$ defects.

In summary, Zn-doped RuO$_2$ nanowire arrays with outstanding performance of acidic OER were successfully synthesized by a simple pyrolysis method. The substitutionally doping of Zn both regulated catalyst morphology and created an abundance of V$_O$ defects and low-valent Ru sites. The self-supporting py-RuO$_2$:Zn nanowires (on Ti) exhibited impressive activity and durability for OER in 0.5 M H$_2$SO$_4$, evidenced by low overpotentials of 173, 304, and 373 mV at 10, 500, and 1000 mA cm$^{-2}$, respectively, and very modest degradations during continuous tests at 10 mA cm$^{-2}$ for 1000 h and 50 mA cm$^{-2}$ for 100 h. Theoretical studies showed that the V$_O$ defects and Zn dopants caused an weakened binding of oxygen adsorbates at active Ru centers and, more interestingly, enabled a moderate adsorption of *OH species on Zn sites. As a result, the OER path was altered from the conventional AEM to a Ru–Zn dual-site OPM, thereby significantly enhancing the OER activity. In the meantime, the OPM path avoided the over oxidation of the OER metal sites and thus protected the active centers, and the presence of Zn dopants and V$_O$ defects enabled a structure stabilization of RuO$_2$ matrix. Consequently, an excellent OER stability was obtained on the V$_O$-containing Zn-doped RuO$_2$ oxide.

## Methods
### Preparation of py-RuO$_2$:Zn on metallic Ti plate
Ti plate was first etched in 10 wt.% oxalic acid solution at 95 °C for 2 h to remove the surface oxide, then rinsed with copious deionized water

and dried in air. Amount of aqueous solution containing $RuCl_3$ and $Zn(NO_3)_3$ with controlled mole ratio of Zn/Ru and dosage of $Ru^{3+}$ cation was pipetted onto the freshly cleaned Ti plate with a confined area of $0.5 \times 1.0$ cm$^2$. The obtained precursor coating was dried naturally in air and then pyrolyzed in a muffle furnace at 350 °C for 4 h in air (ramping rate: 5 °C min$^{-1}$) to transform the precursors to metal oxide. After naturally cooled to room temperature, the sample was then etched in 1.0 M HCl aqueous solution to remove the unwanted ZnO species. The resulted sample was rinsed with copious water and dried in air. To optimize the morphology and OER performance of py-RuO$_2$:Zn catalyst, the conditions of preparation were screened, including the Zn/Ru mole ratio (0.2, 0.5, 1.0, 5.0) and $Ru^{3+}$ dosage (1.0, 3.0, 6.0 μmol cm$^{-2}$) in the precursor solution, and the reaction temperature (300, 350, 400, 450, 500 °C) of pyrolysis. The results revealed that py-RuO$_2$:Zn catalyst with regular nanowire array appearance and the best OER property can be controllable constructed under the conditions: 0.5, 6.0 μmol cm$^{-2}$, and 350 °C for the Zn/Ru mole ratio, $Ru^{3+}$ dosage, and reaction temperature, respectively.

For comparison, pure RuO$_2$, referred as py-RuO$_2$, was also prepared by the pyrolysis method under the optimal conditions without the addition of Zn precursor. The commercial RuO$_2$, referred as c-RuO$_2$, purchased from Sigma Aldrich was also used as a control sample for comparison. In addition, other materials, such as carbon fiber paper (CFP) and fluorine-doped tin oxide glass (FTO), were then used to replace the Ti plate in the fabrication of py-RuO$_2$:Zn nanowire arrays coating under the identical conditions, in order to examine the practicability of this method on different substrates.

## Physical characterizations

Scanning electron microscopic (SEM) images and energy-dispersive X-ray spectroscopy (EDS) analysis were obtained with Merlin Compact (Carl Zeiss NTS GmbH) at 15 kV. Grazing incidence XRD (GIXRD) patterns were performed on a Phillips PANalytical X'Pert Pro diffractometer operating at 40 mA and 40 kV using a curved graphite diffracted-beam monochromator with Cu $K\alpha$ radiation (incident angle = 0.3° for GIXRD, $\lambda = 1.541$ Å). XRD patterns were recorded by the SmartLab (Rigaku) diffractometer with Cu-$K\alpha$ radiation. High-resolution transmission electron microscopic (HRTEM) studies and high-angle annular dark-field scanning TEM (HADDF-STEM) analyses were performed on JEOL 2100 F at 200 kV. X-ray photoelectron spectroscopy (XPS) studies were carried out on Thermo ESCALAB 250XI using an Al $K\alpha$ monochromated source (150 W, $h\nu = 1486.6$ eV). The X-ray absorption fine structure spectra (XAFS) were collected at BL14W beamline in Shanghai Synchrotron Radiation Facility (SSRF). The mass loadings of Ru and Zn in py-RuO$_2$:Zn catalysts before and after the acid etching treatment were separately measured by ICP-MS method (Supplementary Table 1). To prepare the analytical solution, 5 mg of the py-RuO$_2$:Zn powder scraped off the Ti substrate was dispersed in 20 mL solution containing HNO$_3$, HCl, and HClO$_4$ with the ratio of 4:12:3, then transferred into a hydrothermal 50 mL Teflon-lined stainless-steel autoclave. Finally, the sample was sealed and treated at 180 °C for 72 h to fully digest all solid parts. The dissolution of Ru element was studied by inductively coupled plasma mass spectrometry (ICP-MS, Aglient 7800). Degradation of py-RuO$_2$:Zn in duration test was monitored by taking a 1 mL sample of the electrolyte solution at different time after the test (1, 10, 50, 100, 250, 750, 1000 h) for ICP-MS analysis. The 1 mL sample was diluted with 0.5 M H$_2$SO$_4$ and 0.1 M HCl.

## Electrochemical measurements

All electrochemical measurements were tested using a CHI 660E electrochemical analyzer (CH Instruments, Inc., Shanghai) in O$_2$-saturated 0.5 M H$_2$SO$_4$ electrolyte. The pH of the electrolyte, $0.30 \pm 0.01$, was measured with a microprocessor-based pH-meter (Leici PHSJ-3F) and further calibrated by a reversible hydrogen electrode (RHE).

A H-type three-electrode cell was used with a proton exchange membrane to separate each chamber. Saturated Ag/AgCl immersed in a double salt bridge and Pt plate served as the reference and counter electrodes, respectively. The Ag/AgCl reference electrode was first calibrated by a reversible hydrogen electrode, and the potential was reported on RHE scale with 85% $iR$-correction unless otherwise specified. Solution resistance ($R = 3.2 \pm 0.4$, $2.9 \pm 0.4$, and $4.0 \pm 0.6$ Ω for py-RuO$_2$:Zn, py-RuO$_2$, and c-RuO$_2$, respectively) was measured by electrochemical impedance spectroscopy (EIS) at frequencies ranging from 10 Hz to 100 kHz. The current densities were calculated with respect to the geometrical area of the electrodes ($0.5 \times 1.0$ cm$^2$). Linear sweep voltammetry (LSV) and cyclic voltammetry (CV) techniques were performed to examine the electrocatalytic performances of the as-prepared catalysts toward oxygen evolution reaction (OER) in acidic environments. A potential scan rate of 10 mV s$^{-1}$ is used. Chronopotentiometric (CP) technique was employed for the long-term stability test of OER.

## Density functional theory (DFT) calculations

We have employed the Vienna Ab Initio Package (VASP)[89,90] to perform all the density functional theory (DFT) calculations within the generalized gradient approximation (GGA) using the RPBE[91] formulation. We have chosen the projected augmented wave (PAW) potentials[92,93] to describe the ionic cores and take valence electrons into account using a plane wave basis set with a kinetic energy cutoff of 450 eV. Partial occupancies of the Kohn−Sham orbitals were allowed using the Gaussian smearing method and a width of 0.05 eV. The electronic energy was considered self-consistent when the energy change was smaller than $10^{-6}$ eV. A $2 \times 2$ unit cell with 4-layers thickness was employed with 15 Å vacuum in the $z$ axis to avoid image interactions. The bottom two layers were keep fixed while the top two layers were relaxed during geometry optimization. A geometry optimization was considered convergent when the force change was smaller than 0.05 eV/Å. Grimme's DFT-D3 methodology[94] was used to describe the dispersion interactions. The Brillouin zone integral uses the surfaces structures of $3 \times 3 \times 1$ monk horst pack K point sampling. The demetallization energies were computed as $\Delta E = E_{surface} - E_{atom} - E_{surface-vac}$, where $E_{surface}$ and $E_{surface-vac}$ are the total energies of the surface and surfaces with one metal removed, $E_{atom}$ is the single atom energies in the hexagonal Ru and Zn. The computational hydrogen electrode (CHE) approach was used which assumes that the chemical potential of a proton-electron pair is equal to that of gas-phase H$_2$, at $U_{elec} = 0$ V vs. RHE. The reaction free energy of each proton-electron transfer step were obtained by $\Delta G = \Delta E + \Delta ZPE - T\Delta S + \Delta G_U + \Delta G_{pH} + \Delta G_{filed}$, where $\Delta E$ is the change in the total ground-state energy obtained from DFT calculations, $\Delta ZPE$ is the change in zero-point energies, $T$ is 298 K and $\Delta S$ is the change in entropy. $\Delta G_U = eU$, where $U$ is the electrode potential. $\Delta G_{pH} = 0.0592 \times pH$ and the pH = 0 was used. $\Delta G_{filed}$ is neglected in the calculations.

## Data availability

The data that support the findings of this study are available within the article and its Supplementary Information, where the source data of Figs. 1−5 are listed in the Source Data file (https://doi.org/10.6084/m9.figshare.22621852)[95]. Extra data are available from the corresponding authors upon reasonable request. Source data are provided with this paper.

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

## Acknowledgements

The authors would like to acknowledge financial supports from the Natural Science Foundation of Henan Province (182300410196), National Natural Science Foundation of China (U22A20120, 52071135, 51871090, U1804135, and 51671080), Plan for Scientific Innovation Talent of Henan Province (194200510019), and Key Project of Educational Commission of Henan Province (19A150025).

## Author contributions

D.Z., B.L., and S.L. conceived the study. M.L., Y.Y., B.X., and D.Z. designed the experiment and performed the initial tests. X.Y conducted the theoretical calculations. H.S., and G.I.N.W. assisted in the data analysis. D.Z., M.L., and X.Y. co-wrote the manuscript. All authors discussed the results and commented on the manuscript.

## Competing interests

The authors declare no competing interests.
