## [Peer Review File · Nature Communications]

REVIEWER COMMENTS

Reviewer #1 (Remarks to the Author):

The authors report a Zn-doped RuO₂ nanowire array electrocatalyst with OER activity and stability under an acidic environment. Their theoretical calculations indicate that the abundant VO defects assisted the dissociation of water molecules to generate OH_{ads} adsorbates at vicinal low-valent Ru sites. The py-RuO₂:Zn nanoarrays (on Ti) catalyst exhibits high intrinsic OER activity, including ultralow OER overpotentials of 173, 280, and 330 mV to reach current densities of 10, 500, and 1000 mA cm⁻², respectively, in 0.5 M H₂SO₄. The work is interesting. However, it requires a revision for publication, as commented below.

This manuscript reports Zn-doping in the RuO₂ lattice (catalyst name py-RuO₂:Zn) to enhance the electrocatalytic activity and stability toward OER in acidic conditions. By introducing VO defect sites and low-valent Ru cations in the material, the catalyst showed a low overpotential of 173 mV at 10 mA cm⁻² in 0.5 M H₂SO₄ solution. The VO defect sites and low-valent Ru cations were considered from the XPS and XAFS. The free energy profiles were calculated by density functional theory (DFT). Overall, the Ru based electrocatalyst showed good activity in acidic condition. It is an interesting work. The authors claimed that the obtained py-RuO₂: Zn nanoarrays were rich in oxygen vacancies (VO defects) and low-valent Ru³⁺ sites, and therefore had outstanding OER performance. However, the manuscript lacks high novelty and the clear experimental evidences which support the oxygen vacancies, low-valent Ru³⁺ sites, doping of Zn in the RuO₂ lattice or changes in the coordination environment of Ru-Ru /Ru-O-Ru with Zn doping. It does not provide insights into the py-RuO₂:Zn structure and the role of Zn doping in the structure of RuO₂ is unclear. Furthermore, though the material uses an expensive novel metal of Ru, this OER performance is not better than those of high-performing non-expensive transition metal materials.

Comments:

1. The authors report Zn-doped RuO₂ lattice showing a low overpotential of 173 mV at 10 mA cm⁻² in 0.5 M H₂SO₄ solution. The authors stress the high OER performance in acidic solution. Previously, the RuO₂ based catalyst (a compressed metallic Ru-core and oxidized Ru-shell with Ni single atoms) which showed a low overpotential of 184 mV at 10 mA cm⁻² in 0.5 M H₂SO₄ solution was already reported (Adv. Ener. Mater. 2021, 11, 2003448). This needs to be addressed in text.

2. It seems that the authors chose different facet for DFT calculation of OER reaction on RuO₂ with O vacancies from the facet RuO₂/Zn-doped RuO₂. It is unclear why the authors chose different facet for RuO₂ with O vacancies.
3. In addition, the facet of RuO₂ and Zn-doped RuO₂ seems as (001) not (110). The RuO₂ (110) was found to be stable and widely studied for OER. As the rate determining step was found to be OOH* formation rather than OOH* deprotonation in case of Ru (001), the improvement in performance may come from different aspects. Besides, a large (110) surface exists in the Zn-doped RuO₂. The activity on RuO₂ (110) and Zn-doped RuO₂ (110) surfaces need to be considered too.
4. It seems that the intermediate adsorbed surfaces are not considered to construct Pourbaix diagram. The activity of each model could be affected by different stable surface state at a given condition. It is recommended to construct Pourbaix diagram by considering *O, *OH, *OOH, and H₂O adsorbed surfaces.
5. The optimized py-RuO₂: Zn nanoarrays showed OER overpotentials of 173 mV at 10 mA/cm² in 0.5 M H₂SO₄ solution. In Introduction, a current status of high-performing OER catalysts needs to be addressed in both acid and base conditions. For example, note that the catalysts of Ir/NFS and non-expensive 3D-a-NiFeOOH/N-CFP show a low overpotential of 170 mV at 10 mA/cm² in 1 M KOH (Nat. Commun. 2022, 13, 24; J. Mater. Chem. A 2021, 9, 14043), while the material in this manuscript uses an expensive novel metal of Ru. Nevertheless, this OER performance is not better than those of high-performing non-expensive transition metal materials in basic condition.
6. To provide an in-depth understanding of the OER mechanism on the py-RuO₂:Zn catalyst, the authors should pay more attention to clearly figuring out the oxygen vacancies, location of Zn²⁺ ions in the RuO₂ lattice, changes in the coordination environment of Ru-Ru/Ru-O-Ru after the Zn doping and catalyst structure after the stability test. The possible OER mechanism needs to be discussed among many mechanisms suggested in the literature.
7. The catalyst can be stable for 500 h with ~15% Ru loss. However, this loss seems to be serious for practical applications because Ru is highly expensive.
8. The catalytic stability of py-RuO₂:Zn for OER needs to be checked at a high current density.
9. The Tafel plots should be derived from the steady-state polarization curves (ACS Energy Lett. 2021, 6, 1607).

10. Faradaic efficiency analysis should be performed.

11. The authors need to perform all the OER experiments in oxygen-saturated environments.

12. The authors in the current manuscript show the fabrication of py-RuO₂: Zn nanowire arrays on carbon fiber paper and fluorine-doped tin oxide glass. The authors also need to show the catalytic performance of py-RuO₂: Zn nanowire arrays on these substrates also. What are the advantages of using a Ti plate over the other substrate?

13. Schematic illustration of the catalyst fabrication method (Figure 1. (a)) is not up to the standard level for publication.

Reviewer #2 (Remarks to the Author):

I recommend the paper "Construction of Zn-doped RuO₂ nanowires for exceptional efficient and stable water oxidation in acidic media" to be published in Nature Communications after Major Revision. The catalytic performance of the catalyst studied for the OER is very good and it shows acceptable durability for a Ru catalyst. The characterization is also well discussed. However, the authors would need to perform some more electrochemical measurements and correct some details.

1. Line 101, page 5; Results and Discussion Section: Which are the "unstable species formed" during the synthesis process?

2. Line 99-106, page 5; Results and Discussion Section: The mass loading of Ru and Zn in the plates was calculated by comparing the initial RuCl₃ and Zn(NO₃)₂ pipetted on the Ti plate and then subtracted the part (calculated from ICP) lost after the acid leaching, right? There was no loss of the initial precursors-dissolution when it was pipetted onto the freshly etched Ti plate? How did the authors control it?

3. Line 168, page 9; I do not understand why the electrochemical characterization proves the existence of Ru³⁺ in the catalyst. The redox peaks on the cyclic voltammetry (CV) curves between 0.5 ~ 0.8 V

associated to Ru³⁺/Ru⁴⁺ couple (Figure S12) appear below the open circuit potential, so it only proves that there is Ru⁴⁺ in the initial catalyst.

4. The XAS study is interesting, with the observation of Zn-Ru distances similar to Ru-Ru on the RuO₂, and therefore verifying the incorporation of Zn²⁺ in the structure. Could the authors try to explain why even with the introduction of Zn²⁺ in the RuO₂ structure (which is already more reduced than Ru⁴⁺ and therefore has to produce oxygen vacancies) do they think that Ru⁴⁺ is in part also reduced into Ru³⁺? I would expect the opposite effect on Ru⁴⁺ to balance the introduction of Zn²⁺ and keep the electroneutrality of RuO₂ in a more stable structure, even with some oxygen vacancies.

5. In relation with the previous question, in Figure S14. Why did the authors not included the effect of the introduction of Zn²⁺? Zn²⁺ should introduce more oxygen defects than the presence of Ru³⁺, right?

6. Figure 3a inset should be plotted bigger in the main text, maybe instead of Figure 3c. That the catalyst can achieve such large current densities at low potentials is a relevant result, so it should be plotted independently and larger and without the capacitance correction, so the readers can see the hysteresis between the anodic and cathodic curves. Also, is it possible to measure (and plot) more than one cycle up to such current densities?

7. Are Figure 3a, Figure 3b and Figure S19 already iR corrected? Please, check it and if they are iR corrected change the axis to E-iR. The results change a lot if the graphs are corrected or not.

8. Figure S19: Could the authors measure more than 30 OER cycles (at least up to 150 mA cm⁻²) to determine the durability of the catalyst over cycling?

9. In Figure S5f, which is the meaning of x? Is it the value of Zn/Ru? The values seems to be too high compared to the value of Zn/Ru in py-RuO₂:Zn, right?

10. Figure S21a: Is the morphology of py-RuO₂ also nanowires. How do the authors explain the huge difference between the ECSA of py-RuO₂:Zn and py-RuO₂?

11. The authors did not find differences in the Raman bands at 430 and 588 cm⁻¹ between py-RuO₂:Zn and c-RuO₂ catalysts. In principle, they claim that those peaks are associated with the vibration of Ru⁴⁺-O bonds and Ru³⁺-O bonds. So, that means that both catalysts have the same number of reduced Ru³⁺? Then the large number of defects can be related to Zn²⁺ or not?

Reviewer #3 (Remarks to the Author):

This is my review of the MS titled "Construction of Zn-doped RuO₂ nanowires for exceptionally efficient and stable water oxidation in acidic media"

by Baozhong Liu, Siyu Lu, and collaborators. This work describes a Zn-Doped RuO₂ catalyst working in an acidic electrolyte with an ultra-low overpotential of 173 meV and stable for 1000h. The works argue that the origin of this performance comes from Ru⁺³ and oxygen vacancies (V_O) and that the OOHads to the O₂ step/barrier is lowered. The formation of Zn-O-Ru is argued to prevent Ru dissolution.

Reading this manuscript, which on the surface is very promising leaves me with many unanswered questions and lots of potential issues.

The noteworthy results are the synthesis of the nice RuO₂ nanowires and low overpotentials obtained.

The significance of this work to the field and related fields is that low onset overpotential is observed.

How does it compare to the established literature? If the work is not original, please provide relevant references.

Many theoretical references and stability studies are omitted. The originality of the work is somehow limited as previous Zn@RuO₂ has been made.

Does the work support the conclusions and claims, or is additional evidence needed?

The characterization is somewhat supporting the claim of Ru⁺³, but again the XPS signal is quite small and XAS Ru⁺³ signal doesn't exist. The whole DFT part and claims there are simply unsupported in the data. The stronger OH* adsorption will lead to worse, not better OER as concluded by the authors, as binding OH* too strongly will prohibit the OOH⁻ → O₂ step, which is in direct conflict with the author's claims. Where is the detailed view of why the OOH⁻ → O₂ step is suddenly lowered, while OH* and therefore OOH* bind stronger?

The lower the oxidation state of Ru, or when binding at the vacancy sites will lead to much stronger OH* binding and higher overpotentials. See 10.1021/acs.jpcc.7b03481. In fact, higher oxidation state of Ru leads to better activity. (10.1021/acs.jpcc.7b03481, 10.1021/acscatal.0c02252). This is in stark contrast to what the authors find. Also, authors need to provide detailed dGs for each OER step in a table format. Why the calculated OER overpotentials are so high? They should be in the 0.5 to 0.3 V range as in all these other studies. Lastly, authors need to show how/where the Zn dopant is more stable with vacancy

present as opposed to without vacancy. If authors cannot calculate lower overpotentials and show calculated dGs w. structures, or to show a more stable Zn in presence of V₂O₅, I request to remove the whole DFT part.

Are there any flaws in the data analysis, interpretation and conclusions? Do these prohibit the publication or require revision?

The stability window is limited to 1.46 eV above which the catalyst dissolves! This is well known for all RuO₂-containing compounds. So far none of the works was able to fix this problem. (pls cite these works such as <https://doi.org/10.1021/ja510442p> or)

Is the methodology sound? Does the work meet the expected standards in your field?

The whole DFT part and claims there are simply unsupported in the data.

Is there enough detail provided in the methods for the work to be reproduced?

To a degree.

In summary, the low overpotential is likely due to oxidation of water (not O₂ evolution) or dual site OER mechanism, but the authors failed to prove convincingly that is caused by Ru³⁺.

Response to the reviewers' comments

We thank all the reviewers for their valuable comments and questions that help us significantly improve the revised manuscript. The point-by-point responses to the comments are attached below and all the corresponding revisions newly made are marked in *red font* highlighted in the revised manuscript.

Reviewers' comments:

Reviewer #1 (Remarks to the Author):

Comments: The authors report a Zn-doped RuO₂ nanowire array electrocatalyst with OER activity and stability under an acidic environment. Their theoretical calculations indicate that the abundant V_O defects assisted the dissociation of water molecules to generate OH_{ads} adsorbates at vicinal low-valent Ru sites. The py-RuO₂:Zn nanoarrays (on Ti) catalyst exhibits high intrinsic OER activity, including ultralow OER overpotentials of 173, 280, and 330 mV to reach current densities of 10, 500, and 1000 mA cm⁻², respectively, in 0.5 M H₂SO₄. The work is interesting. However, it requires a revision for publication, as commented below.

Response: We appreciate the reviewer for the positive comments on this work. The reviewer's suggestions and criticisms help us to substantially improve the quality of manuscript. We have addressed the comments point-by-point as follows.

Comments: This manuscript reports Zn-doping in the RuO₂ lattice (catalyst name py-RuO₂:Zn) to enhance the electrocatalytic activity and stability toward OER in acidic conditions. By introducing V_O defect sites and low-valent Ru cations in the material, the catalyst showed a low overpotential of 173 mV at 10 mA cm⁻² in 0.5 M H₂SO₄ solution. The V_O defect sites and low-valent Ru cations were considered from the XPS and XAFS. The free energy profiles were calculated by density functional theory (DFT). Overall, the Ru based electrocatalyst showed good activity in acidic condition. It is an interesting work. The authors claimed that the obtained py-RuO₂: Zn nanoarrays were rich in oxygen vacancies (V_O defects) and low-valent Ru³⁺ sites, and therefore had outstanding OER performance. However, the manuscript lacks high novelty and the clear experimental evidences which support the oxygen vacancies, low-valent Ru³⁺ sites, doping of Zn in the RuO₂ lattice or changes in the coordination environment of Ru-Ru /Ru-O-Ru with Zn doping. It does not provide insights into the py-RuO₂:Zn structure and the role of Zn doping in the structure of RuO₂ is unclear. Furthermore, though the material uses an expensive novel metal of Ru, this OER performance is not better than those of high-performing non-expensive transition metal materials.

Response: Thanks for the thoughtful comments.

Our response to the reviewer's concern on the novelty of this work:

Our contribution to fields mainly includes in two aspects: 1) a highly active RuO₂-based catalyst for OER in acidic media; 2) a facial new way of preparing high quality RuO₂ nanowires.

RuO₂ is a versatile material widely used in the fields of catalysis, supercapacitor-based energy storage, and electronics (Chem. Rev. 2016, 116, 2982–3028; Chem. Rev. 2012, 112, 3356–3426; Nat Rev Mater 2020, 5, 5–19; J. Electrochem. Soc. 166 D3219–D3225). However, up to now, wire-like RuO₂-based nanoarrays were mainly prepared by complex precipitation-recrystallization (ACS Appl. Nano Mater. 2020, 3, 3847–3858; Cryst. Growth. Des. 2010, 10, 2585–2590.) and reaction-recrystallization (Energy Environ. Mater., 2019, 2, 201–208; Adv. Mater. 2007, 19, 143–149) methods. The present work reports a new skillful

one-step way to construct high quality RuO₂-based nanowire arrays on different substrates, including metallic Ti foil, carbon fiber, and F-doped tin oxide coated glass, making it more practical in applications.

Moreover, the RuO₂-based nanowires prepared in this work showed excellent catalytic properties for OER process in acidic media. An overpotential as low as 173 mV at 10 mA cm⁻² and a durability up to 1000 h were observed on the material. The performance ranks among the top level of recently reported RuO₂-based catalysts in the acidic OER field (please see the Table 3 in the Supplementary Information). On the basis of theoretic calculations, we found that the oxygen vacancies and Zn dopants can synergistically regulate the OER activity of Ru centers and, more interestingly, the doped Zn ions further exhibited as active sites to bind *OH adsorbates, evoking a new Ru-Zn dual-site oxide path mechanism of OER to improve the OER activity. This work provides a guideline to rationally design active OER electrocatalysts in acidic media.

We also noticed that the design of high performance OER catalysts has made great progress in the past decade, promoting a large development of water electrolysis to produce eco-friendly H₂ gas. At present, both traditional alkaline water electrolyzer and advanced proton exchange membrane (PEM) based electrolyzer have been deployed in practice, while the former is on a much larger scale. However, in terms of operating safety and energy efficiency, the latter has more advantages (Adv. Energy Mater. 2017, 7, 1601275; Nat. Energy 2019, 4, 430–433; Adv. Energy Mater. 2022, 12, 2103670). But the related highly corrosive conditions at high oxidation potentials under acidic environments make the development of efficient OER catalysts a great challenge. Recent years, attractive OER catalysts for alkaline water electrolysis (AWE) have been widely reported, especially those based on the earth-abundant transition metals. The most representatives are NiFe-based (oxy)hydroxides and layered double hydroxides (J. Mater. Chem. A 2021, 9, 14043–14051; Angew. Chem. Int. Ed. 2021, 60, 9699-9705; Nat. Commun. 2021, 13, 2191.), on which OER overpotential can be significantly reduced to ca. 170 mV at 10 mA cm⁻². This offers a chance to build efficient AWE assemblies without noble metals in application. However, most of those OER catalysts show unsatisfied kinetics in acidic media, which, furthermore, suffer from severe degradation under the harshly corrosive conditions in acidic media. So far, only the catalysts based on Ru and Ir noble metals can meet the requirements of PEM water electrolysis assemblies in practice (Adv. Mater. 2021, 33, 2006328; Adv. Energy Mater. 2022, 12, 2103670). Our work had thus been conducted with aim to develop high performance relatively low-cost RuO₂-based acidic OER catalyst.

To further clarify the significance of this work, we have upgraded the Introduction section in the revised manuscript. The revision is as follows:

On page 3: “Hydrogen (H₂) generation via electrochemical water splitting is a promising way to efficiently store intermittent renewable energy.¹⁻³ However, the sluggish oxygen evolution reaction (OER) on anode hinders the overall efficiency of water splitting and leads to large undesired energy consumption.^{4,5} Therefore, the design of high performance OER catalysts is regarded as a matter of urgency for the industrial application of water-to-H₂ conversion.⁶⁻⁹ To date, attractive candidates based on earth-abundant transition metals, especially the (oxy)hydroxides and layered double hydroxides of Ni–Fe,¹⁰⁻¹² have been widely reported under basic conditions, which offers a chance to build low-cost alkaline water electrolysis (AWE) assemblies without noble metals in application. However, the currently deployed AWE devices are still facing intrinsic challenges, including the low operating pressure, inevitable gas crossover, slow load response, and limited current density, mainly due to the utilization of a diaphragm and a liquid

electrolyte.¹³

Compared with AWE, water electrolysis using proton exchange membrane (PEM) electrolyzers can effectively address the above challenges with significantly improved performance.¹⁴⁻¹⁶ But the highly corrosive conditions at high oxidation potentials under acidic environments make the development of efficient OER catalysts a great challenge. Most existing OER catalysts with excellent performance in basic condition generally show unsatisfied kinetics in acidic media, which, furthermore, suffer from severe degradation under the harsh conditions. So far, only the catalysts based on Ru and Ir noble metals can meet the requirements of PEM water electrolysis in practical deployment, though the scarcity of iridium and relatively low mass activity of Ir-based catalysts are serious obstacles to industrial scale H₂ production.^{8,17,18}

References cited in this section are listed as bellow.

“1. Gür, T. M. Review of electrical energy storage technologies, materials and systems: Challenges and prospects for large-scale grid storage. *Energy Environ. Sci.* 11, 2696–2767 (2018).

2. Li, W. et al. Exploiting Ru-induced lattice strain in CoRu nanoalloys for robust bifunctional hydrogen production. *Angew. Chem. Int. Ed.* 60, 3290–3298 (2021).

3. Song, H. et al. Single atom ruthenium-doped CoP/CdS nanosheets via splicing of carbon-dots for robust hydrogen production. *Angew. Chem. Int. Ed.* 60, 7234–7244 (2021).

4. Ding, H., Liu, H., Chu, W., Wu, C. & Xie, Y. Structural transformation of heterogeneous materials for electrocatalytic oxygen evolution reaction. *Chem. Rev.* 121, 13174–13212 (2021).

5. Ali, A., Long, F. & Shen, P. K. Innovative strategies for overall water splitting using nanostructured transition metal electrocatalysts. *Electrochem. Energy Rev.* 5, 1 (2022).

6. McCrory, C. C. L. et al. Benchmarking hydrogen evolving reaction and oxygen evolving reaction electrocatalysts for solar water splitting devices. *J. Am. Chem. Soc.* 137, 4347–4357 (2015).

7. McCrory, C. C. L., Jung, S., Peters, J. C. & Jaramillo, T. F. Benchmarking heterogeneous electrocatalysts for the oxygen evolution reaction. *J. Am. Chem. Soc.* 135, 16977–16987 (2013).

8. An, L. et al. Recent development of oxygen evolution electrocatalysts in acidic environment. *Adv. Mater.* 33, 2006328 (2021).

9. Lei, Z. et al. Coordination modulation of iridium single-atom catalyst maximizing water oxidation activity. *Nat. Commun.* 13, 24 (2022).

10. Chen, Z. et al. TM LDH meets birnessite: A 2D-2D hybrid catalyst with long-term stability for water oxidation at industrial operating conditions. *Angew. Chem. Int. Ed.* 60, 9699–9705 (2021).

11. Thangavel, P., Kim, G. & Kim, K. S. Electrochemical integration of amorphous NiFe (oxy)hydroxides on surface-activated carbon fibers for high-efficiency oxygen evolution in alkaline anion exchange membrane water electrolysis. *J. Mater. Chem. A* 9, 14043–14051 (2021).

12. He, Z. et al. Activating lattice oxygen in NiFe-based (oxy)hydroxide for water electrolysis. *Nat. Commun.* 13, 2191 (2022).

13. Carmo, M., Fritz, D. L., Mergel, J. & Stolten, D. A comprehensive review on PEM water electrolysis. *Int. J. Hydrogen Energ.* 38, 4901–4934 (2013).

14. Reier, T., Nong, H. N., Teschner, D., Schlögl, R. & Strasser, P. Electrocatalytic oxygen evolution reaction in acidic environments – reaction mechanisms and catalysts. *Adv. Energy Mater.* 7, 1601275 (2017).

15. Kibsgaard, J. & Chorkendorff, I. Considerations for the scaling-up of water splitting catalysts. *Nat. Energy* 4, 430–433 (2019).

16. Chen, Z. et al. Advances in oxygen evolution electrocatalysts for proton exchange membrane water electrolyzers. *Adv. Energy Mater.* 12, 2103670 (2022).

17. Li, L., Wang, P., Shao, Q. & Huang, X. Recent progress in advanced electrocatalyst design for acidic oxygen evolution reaction. *Adv. Mater.* 33, 2004243 (2021).

18. She, L. et al. On the durability of iridium-based electrocatalysts toward the oxygen evolution reaction under acid environment. *Adv. Funct. Mater.* 32, 2108465 (2022).”

Fig. R1 Physical characterizations of py-RuO₂:Zn and control catalysts. (a) XRD pattern and (b) TEM image, together with the FFT pattern (inset) and the simulated crystal structure, of py-RuO₂:Zn catalyst. (c) XANES and EXAFS at Ru K-edge of py-RuO₂:Zn and the control catalysts. (e) The edge energies for Ru K-edge as a function of the oxidation state of the Ru.

Fig. R2 (a) Raman spectra for py-RuO₂:Zn, py-RuO₂, and c-RuO₂ catalysts in O₂-saturated 0.5 M H₂SO₄ at given electrode potentials. (b) Normalized intensity of Raman band at 588 cm⁻¹ to that at 430 cm⁻¹ on the catalysts as a function of applied potential. The areas under the bands were used to calculate the I₅₈₈/I₄₃₀ ratio.

Our response to the comments on the structure of py-RuO₂:Zn catalyst:

According to the XRD and TEM results (Figs. R1a and b), py-RuO₂:Zn catalyst took a rutile phase as to the pristine RuO₂, which indicated that the Ru atoms are octahedral coordinated in py-RuO₂:Zn. Measurements of XANES and EXAFS at Ru K-edge further revealed a slightly lower oxidation state of Ru element in py-RuO₂:Zn than that in RuO₂, that is, +3.4 vs. +4, suggesting the presence of a coordinate unsaturation of the Ru centers in py-RuO₂:Zn (Figs. R1c and e). Moreover, the first Ru–O and Ru–Ru

coordination shells of Ru sites were observed at 1.47 and 3.17 in py-RuO₂:Zn, different from the 1.50 and 3.14 in RuO₂ (Fig. R1d). Thus, the symmetric octahedral coordination of Ru sites was slightly broken after the Zn doping. The results demonstrated that the doping of Zn element broke the octahedral coordination structure of the Ru cations in py-RuO₂:Zn with coordinately unsaturated feature.

In addition to the XAS and XPS characterizations, Raman measurement was also used to study the relative abundance of Ru³⁺ species on the surface. As shown in Fig. R2a, strong Raman bands at 430 and 588 cm⁻¹ were observed on both py-RuO₂:Zn and c-RuO₂ catalysts, associated with the vibration of Ru⁴⁺-O bonds and Ru³⁺-O bonds of hydrated RuO₂ on the surface (Electrochem. Solid-State Lett. 2005, 8, E39–E41). When further normalizing the intensity of the band at 588 cm⁻¹ to that at 430 cm⁻¹, represented by the area ratio under the bands (Fig. R2b), we found that the py-RuO₂:Zn catalyst showed a higher intensity than two pure RuO₂ samples, thereby possessing more Ru³⁺ species on the surface.

Figs. R1c and R2 have been added in the revised Supplementary Information (SI) file, shown as Supplementary Figs. 15 and 13, respectively. Corresponding discussions as below have been updated in the revised manuscript.

On page 10: “The higher content of low-valent Ru species on the surface of py-RuO₂:Zn catalyst remained under the OER conditions, as confirmed by the Raman measurements (Supplementary Fig. 13)”

On page 12: “Calculations on basis of adsorption edge energy revealed that an average oxidation state of Ru species in the catalyst was approximately +3.4 (Supplementary Fig. 15), which was considered as the combination of pristine Ru⁴⁺ and Ru³⁺ cations.”

Zn K-edge XANES results in Fig. R3 revealed that the Zn dopants in py-RuO₂:Zn catalyst took an oxidation state higher than Zn²⁺ in bulk ZnO, indicating an electron donation from Zn dopants to Ru sites. The “white line” feature was considerably broader than that of the bulk ZnO and did not show the characteristic ZnO shoulder at ca. 9663 eV. In addition, the Zn K-edge EXAFS spectrum of py-RuO₂:Zn closely resembled the Ru K-edge spectrum. The results demonstrated that the coordination of Zn²⁺ atoms in py-RuO₂:Zn was the adoption of an octahedral structure through substitutional doping of Zn at Ru sites in the RuO₂ lattice (J. Am. Chem. Soc. 2018, 140, 9383–9386; Prog. Nat. Sci.: Mater. Int. 2016, 26, 347–353).

Fig. R3 (a) XANES and (b) EXAFS at Ru K-edge of py-RuO₂:Zn and control catalysts. (c) A comparison of Ru K-edge and Zn K-edge FT-EXAFS R-space spectra for py-RuO₂:Zn.

The destruction in the octahedral coordination structure and the reduction in the associated oxidation state of the Ru cations indicated the presence of oxygen vacancy (V_O) defects in the py-RuO₂:Zn catalyst. XPS results of O 1s in Fig. R4 further revealed that the peak assigned to V_O defect appeared at ca. 530.5 eV (J. Am. Chem. Soc. 2020, 142, 12430–12439; J. Am. Chem. Soc. 2018, 140, 13644–13653). The concentration of V_O defect showed good linear relationships with the abundance of Ru³⁺ species and the

Zn dopants, again proving the change in the coordination of Ru sites in the catalyst. Accordingly, it can be confirmed that there were amount of V_O defects appearing in the vicinity of Zn–O–Ru moieties.

Fig. R4 (a, b) Core-level O 1s XPS spectra for as-prepared and post-OER py-RuO₂:Zn catalysts and two pure RuO₂ catalysts. (c) The relationship between concentrations of V_O defect (O_V/O_{L-Ru} ratio) and Ru³⁺ species (Ru³⁺/Ru⁴⁺ ratio). (d, e) The dependence of V_O defect (O_V/O_{L-Ru} ratio) on the concentration of Zn dopant, represented by Zn/Ru at.% and Zn/O_V at.%.

Figs. R4d and e have been added in the revised SI file, shown as Supplementary Fig. 16. Corresponding discussion as below has been updated in the revised manuscript.

On page 12–13: “We note that when the Zn dopants took an octahedral coordination structure through substitutional doping at Ru sites in the RuO₂ lattice, a fraction of the Ru will, in principle, be oxidized above 4+ to accommodate the divalent metal, associated with a generation of stoichiometric oxide.³⁰ However, when oxygen vacancies (V_O) present, the oxidation state of Ru^{*n*+} ($n > 4$) would be reduced. Recently, Liu and colleagues reported a Na-doped amorphous/crystalline RuO₂ catalyst containing more low-valent Ru^{*n*+} ($n < 4$) species with the presence of high abundant V_O defects.⁴⁰ To further understand the role of Zn doping on the generation of V_O defects, the relationship between Zn content and V_O concentration was analyzed on the basis of XPS results. A linear dependence of O_V/O_{L-Ru} on the Zn/Ru at.% and the Zn/O_V at.% was found (Supplementary Fig. 16), indicating that the doping of Zn element can induce the generation of V_O defects. In the meantime, the presence of Ru³⁺ and V_O defects was also found in the undoped py-RuO₂ catalyst, caused by the catalyst synthesis method used here.^{61,62} Thus, it can conclude that the Zn doping, in addition to the catalysis synthesis method, has induced the generation of V_O defects and the low-valent Ru sites.”

References cited in this section are listed as follows.

“30. Burnett, D. L. et al. (M,Ru)O₂ (M = Mg, Zn, Cu, Ni, Co) rutiles and their use as oxygen evolution

electrocatalysts in membrane electrode assemblies under acidic conditions. *Chem. Mater.* 32, 6150–6160 (2020).

40. Zhang, L. et al. Sodium-decorated amorphous/crystalline RuO₂ with rich oxygen vacancies: A robust pH-universal oxygen evolution electrocatalyst. *Angew. Chem. Int. Ed.* 60, 18821–18829 (2021).

61. Doubova, L. M., Daolio, S. & De Battisti, A. Examination of RuO₂ single-crystal surfaces: Charge storage mechanism in H₂SO₄ aqueous solution. *J. Electroanal. Chem.* 532, 25–33 (2002).

62. Arikawa, T., Takasu, Y., Murakami, Y., Asakura, K. & Iwasawa, Y. Characterization of the structure of RuO₂–IrO₂/Ti electrodes by EXAFS. *J. Phys. Chem. B* 102, 3736–3741 (1998).”

Comments: 1. The authors report Zn-doped RuO₂ lattice showing a low overpotential of 173 mV at 10 mA cm⁻² in 0.5 M H₂SO₄ solution. The authors stress the high OER performance in acidic solution. Previously, the RuO₂ based catalyst (a compressed metallic Ru-core and oxidized Ru-shell with Ni single atoms) which showed a low overpotential of 184 mV at 10 mA cm⁻² in 0.5 M H₂SO₄ solution was already reported (*Adv. Ener. Mater.* 2021, 11, 2003448). This needs to be addressed in text.

Response: Thanks for the nice suggestions. We have upgraded the relevant content in the Introduction section. The revision is as follows:

On page 4: “Guest elements are usually introduced to improve the OER performance of RuO₂ by modulating the chemical environment of Ru sites.²⁴ As reported recently, via constructing single atomic (e.g., Ni, Pt)^{25,26} and lattice doping (e.g., Mn, Cu, Na)²⁷⁻²⁹ sites, the overpotential of acidic OER on RuO₂ can be reduced to ~180 mV@10 mA cm⁻² with a durability over 200 h.²⁵”

References cited in this section are listed as follows:

“24. Sun, H. & Jung, W. Recent advances in doped ruthenium oxides as high-efficiency electrocatalysts for the oxygen evolution reaction. *J. Mater. Chem. A* 9, 15506–15521 (2021).

25. Harzandi, A. M. et al. Ruthenium core–shell engineering with nickel single atoms for selective oxygen evolution via nondestructive mechanism. *Adv. Energy Mater.* 11, 2003448 (2021).

26. Wang, J. et al. Single-site Pt-doped RuO₂ hollow nanospheres with interstitial C for high-performance acidic overall water splitting. *Sci. Adv.* 8, eabl9271 (2022).

27. Chen, S. et al. Mn-doped RuO₂ nanocrystals as highly active electrocatalysts for enhanced oxygen evolution in acidic media. *ACS Catal.* 10, 1152–1160 (2020).

28. Su, J. et al. Assembling ultrasmall copper-doped ruthenium oxide nanocrystals into hollow porous polyhedra: Highly robust electrocatalysts for oxygen evolution in acidic media. *Adv. Mater.* 30, 1801351 (2018).

29. Retuerto, M. et al. Na-doped ruthenium perovskite electrocatalysts with improved oxygen evolution activity and durability in acidic media. *Nat. Commun.* 10, 2041 (2019).”

Comments: 2. It seems that the authors chose different facet for DFT calculation of OER reaction on RuO₂ with O vacancies from the facet RuO₂/Zn-doped RuO₂. It is unclear why the authors chose different facet for RuO₂ with O vacancies.

Response: Thanks for the comments. We have redone all the calculations using a surface which is more stable and consistent with previous report (*J. Phys. Chem. C* 2015, 119, 4827–4833, *J. Phys. Chem. C* 2017, 121, 18516–18524, *Angew. Chem. Int. Ed.* 2021, 60, 2–11, *Nat. Commun.* 2022, 13, 3784). As shown in Fig. R5, the RuO₂ (110) surface was used. The Zn doped RuO₂ (RuO₂:Zn) and with O vacancies (RuO₂:Zn_V_O) were built on the optimized RuO₂ (110) surfaces. Zn was found to be more stable when doped at the

coordinatively unsaturated Ru (Ru_{cus}) position than the fully coordinated bridge Ru (Ru_{bri}) site, while the bridge row O could form stable vacancy site. All these optimized surfaces were further used for OER simulations.

Fig. R5 Optimized structures of (a) pristine RuO_2 (110) surface (side view, top view, and Ru sites with different coordination), (b) Zn doped RuO_2 ($\text{RuO}_2:\text{Zn}$) surface, and (c) V_O -containing $\text{RuO}_2:\text{Zn}$ ($\text{RuO}_2:\text{Zn}_\text{VO}$) surface.

Fig. R5 has been added in the revised SI file, shown as **Supplementary Figure 41**. Following discussion has been added in the revised manuscript.

On page 24: “To understand the Zn doping and vacancies effect, the Zn doped RuO_2 ($\text{RuO}_2:\text{Zn}$) and with O vacancies ($\text{RuO}_2:\text{Zn}_\text{VO}$) are built on the optimized RuO_2 (110) surfaces with DFT calculations (**Supplementary Figure 41**). Zn was found to be more stably doped at the coordinatively unsaturated Ru (Ru_{cus}) position than the fully coordinated bridge Ru (Ru_{bri}) site, while the bridge row O could form stable vacancy site.”

Comments: 3. In addition, the facet of RuO_2 and Zn-doped RuO_2 seems as (001) not (110). The RuO_2 (110) was found to be stable and widely studied for OER. As the rate determining step was found to be OOH^* formation rather than OOH^* deprotonation in case of Ru (001), the improvement in performance may come from different aspects. Besides, a large (110) surface exists in the Zn-doped RuO_2 . The activity on RuO_2 (110) and Zn-doped RuO_2 (110) surfaces need to be considered too.

Response: Thanks for the nice comments. From the TEM results and structure simulations of the py- $\text{RuO}_2:\text{Zn}$ catalyst in **Fig. R6a and b**, we found that the facet exposed on the surface was more consistent with the {110} planes than the {001}, which were more stable planes widely used for theoretical study on OER mechanism (ChemCatChem 2011, 3, 1159–1165; J. Phys. Chem. C 2017, 121, 18516–18524). We also noticed that an inappropriate crystal facet of RuO_2 and py- $\text{RuO}_2:\text{Zn}$ was used for DFT calculation. We have revised the DFT calculations based on (110) facet. As shown in **Fig. R6c–e**, the optimized RuO_2 (110) surface was used. The Zn doped RuO_2 ($\text{RuO}_2:\text{Zn}$) and with O vacancies ($\text{RuO}_2:\text{Zn}_\text{VO}$) were built on the optimized

RuO₂ (110) surfaces. Zn was found to be more stable when doped at the coordinatively unsaturated Ru (Ru_{CUS}) position than the fully coordinated bridge Ru (Ru_{BR}) site, while the bridge row O could form stable vacancy site. All these optimized surfaces were further used for OER simulations.

Fig. R6 (a) TEM results of the py-RuO₂:Zn catalyst. (b) Structure simulations of (-110), (110), and (001) facets shown from top and front views based on the crystal structure of rutile RuO₂. Optimized structures of (c) pristine RuO₂ (110) surface (side view, top view, and Ru sites with different coordination), (d) Zn doped RuO₂ (RuO₂:Zn) surface, and (e) V_O-containing RuO₂:Zn (RuO₂:Zn_VO) surface.

Fig. R6c–e has been added in the revised SI file, shown as Supplementary Figure 41. Following discussion

has been added in the revised manuscript.

On page 24: “To understand the Zn doping and vacancies effect, the Zn doped RuO₂ (RuO₂:Zn) and with O vacancies (RuO₂:Zn_V_O) are built on the optimized RuO₂ (110) surfaces with DFT calculations (Supplementary Figure 41). Zn was found to be more stably doped at the coordinatively unsaturated Ru (Ru_{cus}) position than the fully coordinated bridge Ru (Ru_{bri}) site, while the bridge row O could form stable vacancy site.”

Comments: 4. It seems that the intermediate adsorbed surfaces are not considered to construct Pourbaix diagram. The activity of each model could be affected by different stable surface state at a given condition. It is recommended to construct Pourbaix diagram by considering *O, *OH, *OOH, and H₂O adsorbed surfaces.

Response: Thanks for the comments. We built the Pourbaix diagram to study the stability of RuO₂ and Zn doped RuO₂ under different pH conditions. The surface coverage of (*O, *OH, *OOH, and H₂O) were in general applied to compare the same surfaces under different conditions (for example: Phys. Chem. Chem. Phys., 2008, 10, 3722–3730). It was built based on the absorption energies of different number of intermediates at different sites. Therefore, the coverage was not considered. In addition, de-metallization of Ru for RuO₂ was the main the reason for the poor stabilities (Nat. Commun.2020, 11, 5368). We thus also computed the energies required for the removal of Ru to study their stabilities.

Comments: 5. The optimized py-RuO₂: Zn nanoarrays showed OER overpotentials of 173 mV at 10 mA/cm² in 0.5 M H₂SO₄ solution. In Introduction, a current status of high-performing OER catalysts needs to be addressed in both acid and base conditions. For example, note that the catalysts of Ir/NFS and non-expensive 3D-a-NiFeOOH/N-CFP show a low overpotential of 170 mV at 10 mA/cm² in 1 M KOH (Nat. Commun. 2022, 13, 24; J. Mater. Chem. A 2021, 9, 14043), while the material in this manuscript uses an expensive novel metal of Ru. Nevertheless, this OER performance is not better than those of high-performing non-expensive transition metal materials in basic condition.

Response: Thanks for the valuable comments and suggestions. We noticed that the design of high performance OER catalysts has made great progress in the past decade. In particular, for basic OER process, attractive candidates based on earth-abundant transition metals have been widely reported. The most representatives are NiFe-based (oxy)hydroxides and layered double hydroxides, on which OER overpotential can be significantly reduced to ca. 170 mV at 10 mA cm⁻² and thus offered a chance to build efficient alkaline water electrolysis (AWE) assemblies without noble metals in application. However, the traditional AWE devices further suffer from intrinsic challenges, including the low operating pressure, inevitable gas crossover, delayed load response, and limited current density, mainly due to the utilization of a diaphragm and a liquid electrolyte.

Compared with AWE technology, water electrolysis based on proton exchange membrane (PEM) is more advanced, which can effectively address the above challenges with significantly improved performance, thus attracting special research interest recently. However, the PEM water electrolysis (PEMWE) is also lack of highly efficient OER catalysts. Most existing OER catalysts with excellent performance in basic condition generally show unsatisfied kinetics in acidic media, which, furthermore, suffer from severe degradation under the harshly corrosive conditions at highly positive potentials in acidic media. So far, only the catalysts based on Ru and Ir noble metals can meet the requirements of PEMWE systems in practical deployment.

The present work reports a skillful method to synthesize RuO₂ based nanowires with excellent

performance for acidic OER. The overpotential was reduced to 173 mV at 10 mA cm⁻² with a durability up to 1000 h, exhibiting a promising application in practice.

In the revised manuscript, we have upgraded the relevant content in the Introduction section. The revision is as follows:

On page 3: “Hydrogen (H₂) generation via electrochemical water splitting is a promising way to efficiently store intermittent renewable energy.¹⁻³ However, the sluggish oxygen evolution reaction (OER) on anode hinders the overall efficiency of water splitting and leads to large undesired energy consumption.^{4,5} Therefore, the design of high performance OER catalysts is regarded as a matter of urgency for the industrial application of water-to-H₂ conversion.⁶⁻⁹ To date, attractive candidates based on earth-abundant transition metals, especially the (oxy)hydroxides and layered double hydroxides of Ni-Fe,¹⁰⁻¹² have been widely reported under basic conditions, which offers a chance to build low-cost alkaline water electrolysis (AWE) assemblies without noble metals in application. However, the currently deployed AWE devices are still facing intrinsic challenges, including the low operating pressure, inevitable gas crossover, slow load response, and limited current density, mainly due to the utilization of a diaphragm and a liquid electrolyte.¹³

Compared with AWE, water electrolysis using proton exchange membrane (PEM) electrolyzers can effectively address the above challenges with significantly improved performance.¹⁴⁻¹⁶ But the highly corrosive conditions at high oxidation potentials under acidic environments make the development of efficient OER catalysts a great challenge. Most existing OER catalysts with excellent performance in basic condition generally show unsatisfied kinetics in acidic media, which, furthermore, suffer from severe degradation under the harsh conditions. So far, only the catalysts based on Ru and Ir noble metals can meet the requirements of PEM water electrolysis in practical deployment, though the scarcity of iridium and relatively low mass activity of Ir-based catalysts are serious obstacles to industrial scale H₂ production.^{8,17,18”}

The recommended references cited in this section are as follows:

“9. Lei, Z. *et al.* Coordination modulation of iridium single-atom catalyst maximizing water oxidation activity. *Nat. Commun.* 13, 24 (2022).”

“11. Thangavel, P., Kim, G. & Kim, K. S. Electrochemical integration of amorphous NiFe (oxy)hydroxides on surface-activated carbon fibers for high-efficiency oxygen evolution in alkaline anion exchange membrane water electrolysis. *J. Mater. Chem. A* 9, 14043–14051 (2021).”

Comments: 6. To provide an in-depth understanding of the OER mechanism on the py-RuO₂:Zn catalyst, the authors should pay more attention to clearly figuring out the oxygen vacancies, location of Zn +2 ions in the RuO₂ lattice, changes in the coordination environment of Ru-Ru/Ru-O-Ru after the Zn doping and catalyst structure after the stability test. The possible OER mechanism needs to be discussed among many mechanisms suggested in the literature.

Response: Thanks for the nice comments. We further compared the possible Zn doping locations and V_O in RuO₂ using DFT calculations. The Zn doping was found to be stable at the Ru_{cus} row, while the V_O preferred to form at the bridge row (Figs. R7a–c). In addition, apart from the conventional adsorbate evolving mechanism (AEM), we have added the discussed lattice oxygen mechanism (LOM, *ACS Catalysis* 2018, 8, 4628–4636; *Adv. Energy Mater.* 2021, 11, 2003448), and a dual-site oxide path mechanism (OPM, *Nat.*

Catal. 2021, 4, 1012–1023) (Figs. R7d–g). We found the AEM was energetically favorable for OER on pristine RuO₂ while OPM was the dominate reaction pathways for OER on Zn doped RuO₂ with/without V_O. We believe that the down shift of Fermi by O vacancy, the weaker absorption of *OH on Zn and the charge difference of Zn and Ru have synergistically lowered the OER overpotential ($\eta = \Delta G_{\max} - 2.13$) from 0.87 V for RuO₂ to 0.61 V for the O vacancy-containing Zn doped RuO₂, by converting the OER path from the single-site AEM to the dual-site OPM. The results are shown in Figs. R7h and i.

Fig. R7 (a) Optimized RuO₂ (110) surface, (b, c) Optimized locations of Zn dopants and V_O defects on the RuO₂ (110) surface. (d–g) OER mechanisms. (h) AEM and OPM paths of OER on V_O-containing Zn-doped RuO₂ catalyst. (i) Calculated free-energy diagrams for preferred OER paths on RuO₂, RuO₂:Zn, and RuO₂:Zn_V_O surfaces.

Fig. R7 and other DFT results have been added in both the revised manuscript and SI file. Discussion as below has been updated in the revised manuscript.

On page 24–26: “To understand the Zn doping and oxygen vacancies effect on the OER activity, density functional theory (DFT) calculations were performed. The Zn doped RuO₂ (RuO₂:Zn) and with O vacancies (RuO₂:Zn_V_O) were built on the optimized RuO₂ (110) surfaces (Supplementary Fig. 41). Zn was found to be more stably doped at the coordinatively unsaturated Ru (Ru_{CUS}) position than the fully coordinated bridge Ru (Ru_{BR}) site, while the bridge row O could form stable vacancy site. Then, different OER paths were investigated to determine the preferred reaction pathways, including the AEM and lattice oxygen mechanism (LOM), as well as the recently highlighted dual-site oxide path mechanism (OPM) (Supplementary Fig. 42).^{35,83} The adsorption energies of reaction intermediates were summarized in the Supplementary Table 6. For clean RuO₂, stronger binding of OH adsorbates ($\Delta G_{\text{OH}} = 0.82$ eV) resulted in the

OER proceeding favorably via a AEM path, following the four-proton-coupled electron transfer steps as $\text{H}_2\text{O} \rightarrow *OH \rightarrow *O \rightarrow *OOH \rightarrow \text{O}_2$.³⁶ The formation of $*OOH$ is the rate-determining step (RDS) with a large free energies barrier of 2.10 eV. By comparison, the LOM and dual site OPM paths are suppressed with much higher energy barriers of RDS (ΔG_{max} for LOM 3.79 eV and OPM 2.48 eV, where ΔG_{max} is the maximum free energy differences among the primary proton-coupled electron transfer steps) (Supplementary Fig. 43). For $\text{RuO}_2\text{-V}_\text{O}$, the presence of bridged O vacancies caused accumulated charge density at both the vicinal Ru_{bri} and Ru_{cus} sites (Supplementary Fig. 44), which then enhanced the binding of $*OH$ at Ru_{cus} centers ($\Delta G_{\text{OH}} = 0.70$ eV) and induced a larger free energies barrier of 2.28 eV for $*OOH$ formation (Supplementary Fig. 45). Therefore, the presence of V_O defects is harmful to the OER proceeding on RuO_2 .^{37,38} In contrast, on the surface of stoichiometric $\text{RuO}_2\text{:Zn}$ oxide, the doping of Zn at Ru_{cus} sites induced a reduction of the charge density at Ru centers, which agreed with the knowledge that a fraction of the Ru will be oxidized above +4 to accommodate the divalent Zn metal.³⁰ As a result, the $*OH$ binding is weakened ($\Delta G_{\text{OH}} = 1.01$ eV) and the OER activity is improved. More interestingly, a Ru–Zn dual-site OPM appeared to be more favorable with a lower ΔG_{max} of 1.91 eV for $*O_{\text{Ru}} \rightarrow *O_{\text{Ru}}\dots*OH_{\text{Zn}}$ of the third proton-coupled electron transfer step, caused by the different binding strength of intermediates on the two sites (Supplementary Fig. 46). The density of states (DOS) and charge density difference suggested that Zn donated some electron to the O and Zn had a lower d band center than Ru (Fig. 5c–e). Therefore, Zn showed weaker absorption of $*O$, $*OH$, and $*OOH$. For example, Zn sites had a ΔG_{OH} of 1.77 eV, while Ru site had had a ΔG_{OH} of 1.01 eV. This would ease the formation of second $*O$. In addition, the charge difference between Zn and Ru also played an important role in promoting the OER, which resulted in a $\sim 0.1 e$ charge difference for the two absorbed $*O$ on Zn and Ru and thus promoted the formation of O–O coupling, and eventually the formation of O_2 (Fig. 5d). With the presence of V_O defects, the charge density at both the Ru_{cus} and Zn_{cus} sites on $\text{RuO}_2\text{:Zn-V}_\text{O}$ surface is slightly increased (Supplementary Fig. 44), associated with a shift of Ru d band center away from Fermi, which further optimized the absorption of intermediates (Fig. 5e). Consequently, the ΔG_{max} ($*O_{\text{Ru}} \rightarrow *O_{\text{Ru}}\dots*OH_{\text{Zn}}$) of OPM is further decreased to 1.84 eV for $\text{RuO}_2\text{:Zn}$ with V_O defects (Fig. 5b). Therefore, we believed that the down shift of Fermi by O vacancy, the weaker absorption of $*OH$ on Zn and the charge difference of Zn and Ru synergistically lowered the OER overpotential ($\eta = \Delta G_{\text{max}} - 2.13$) from 0.87 V for RuO_2 to 0.61 V for the O vacancy-containing Zn doped RuO_2 , by converting the OER path from the single-site AEM to the dual-site OPM (Fig. 5a, b).”

References cited in this section are as follows:

30. Burnett, D. L. et al. (M,Ru)O₂ (M = Mg, Zn, Cu, Ni, Co) rutiles and their use as oxygen evolution electrocatalysts in membrane electrode assemblies under acidic conditions. *Chem. Mater.* 32, 6150–6160 (2020).

35. Lin, C. et al. In-situ reconstructed Ru atom array on $\alpha\text{-MnO}_2$ with enhanced performance for acidic water oxidation. *Nat. Catal.* 4, 1012–1023 (2021).

36. Man, I. C. et al. Universality in oxygen evolution electrocatalysis on oxide surfaces. *ChemCatChem* 3, 1159–1165 (2011).

37. Gayen, P., Saha, S., Bhattacharyya, K. & Ramani, V. K. Oxidation state and oxygen-vacancy-induced work function controls bifunctional oxygen electrocatalytic activity. *ACS Catal.* 10, 7734–7746 (2020).

38. Hubert, M. A. et al. Acidic oxygen evolution reaction activity–stability relationships in Ru-based pyrochlores. *ACS Catal.* 10, 12182–12196 (2020).

83. Vonrüti, N., Rao, R., Giordano, L., Shao-Horn, Y. & Aschauer, U. Implications of nonelectrochemical

reaction steps on the oxygen evolution reaction: Oxygen dimer formation on perovskite oxide and oxynitride surfaces. ACS Catal. 12, 1433–1442 (2022).”

Besides insights from the DFT calculations, the morphologies, charge states of these sample were also characterized using SEM, XPS analysis. The morphology and crystal structure of py-RuO₂:Zn catalyst after the stability test are shown in Fig. R8. An obvious degradation of the catalyst was found after a 1000 h test at 10 mA cm⁻² (Fig. R8a), making it not an appropriate sample for further physical characterizations (Fig. R8b). Another sample undergone a stability test for 350 h was thus employed (Figs. R8a and d). The wire-like morphology and rutile crystal phase of py-RuO₂:Zn did not show obvious changes after the test (Figs. R8c and e), as well as the features of Ru 3p_{3/2}, O 1s, and Zn 2p core level XPS spectra compared with those before the test. Low-valent Ru³⁺ species and V_O defects were also observed on the surface but with a slight decrease in the concentration. The results indicate the coordination environment of Ru maintained well after the stability test.

Fig. R8 (a) Optical photograph of py-RuO₂:Zn electrode before and after the stability tests. (b, c) SEM images of the py-RuO₂:Zn catalyst after an OER stability test at 10 mA cm⁻² for 1000 and 350 h, respectively. (c) CP curve for OER stability test on the py-RuO₂:Zn catalyst at 10 mA cm⁻² for 350 h in 0.5 M H₂SO₄ solution. (e) XRD pattern, (f) Ru 3p_{3/2}, (g) O 1s, and (h) Zn 2p XPS spectra of the py-RuO₂:Zn catalyst after a OER stability test at 10 mA cm⁻² for 350 h.

Fig. R8 has been updated in the revised SI file (Supplementary Figs. 31, 32, and 34) and the revised manuscript (Fig. 4). Following discussion has been also added in the revised manuscript.

On page 21: “No obvious change was found in the Zn 2p XPS spectra (Supplementary Fig. 34).”

Comments: 7. The catalyst can be stable for 500 h with ~15% Ru loss. However, this loss seems to be serious for practical applications because Ru is highly expensive.

Response: Thanks for this nice comment. A mass loss of Ru up to ~15% from the py-RuO₂:Zn catalyst was detected under the durability test for 500 h, which seems to be serious for industrial applications. However, the corresponding dissolution rate of Ru, 0.156 μg cm_{geo}⁻² h⁻¹, is much lower than that of the commercial RuO₂ (~40 μg cm_{geo}⁻² h⁻¹). Further compared with the sputtering-prepared crystalline RuO₂ (~0.36 μg

$\text{cm}_{\text{geo}}^{-2} \text{h}^{-1}$, Catal. Today 2016, 262, 170–180), Na-a/c-RuO₂ ($\sim 0.383 \mu\text{g cm}_{\text{geo}}^{-2} \text{h}^{-1}$, Angew. Chem. Int. Ed, 2021, 60, 18821–18829), Y₂Ru₂O₇ and Gd₂Ru₂O₇ (~ 0.102 and $\sim 0.156 \mu\text{g cm}_{\text{geo}}^{-2} \text{h}^{-1}$, respectively, ACS Catal. 2020, 10, 12182–12196), RuO₂-WC ($0.516 \mu\text{g cm}_{\text{geo}}^{-2} \text{h}^{-1}$, Angew. Chem. Int. Ed, 2022, 61, e202202519), Li_xRuO₂ ($\sim 0.229 \mu\text{g cm}_{\text{geo}}^{-2} \text{h}^{-1}$, Nat. Commun. 2022, 13, 3784), and RuNi₂@G-250 ($0.326 \mu\text{g cm}_{\text{geo}}^{-2} \text{h}^{-1}$, Adv. Mater. 2020, 32, 1908126), the py-RuO₂:Zn catalyst also ranks the top-level of stability in terms of the Ru dissolution rate. When normalized by the ECSA, a value of $37.6 \text{ pg cm}_{\text{ECSA}}^{-2} \text{h}^{-1}$ for Ru dissolution rate was obtained on the py-RuO₂:Zn catalyst, significantly lower than the $\sim 1.05 \mu\text{g cm}_{\text{ECSA}}^{-2} \text{h}^{-1}$ on commercial RuO₂, indicating an intrinsically improved stability of the catalyst. The presence of low-valent Ru species was suggested to account for the stability enhancement. The observation will benefit the design of more stable OER catalysts under acidic environments.

We have added the discussion as follows in the revised manuscript.

On page 20: “We also note that the mass loss of Ru up to $\sim 15\%$ within 500 h from the py-RuO₂:Zn catalyst seems to be serious for industrial applications. However, the corresponding dissolution rate of Ru, $0.156 \mu\text{g cm}_{\text{geo}}^{-2} \text{h}^{-1}$, is much lower than that of the commercial RuO₂ ($\sim 40 \mu\text{g cm}_{\text{geo}}^{-2} \text{h}^{-1}$). Further compared with the high active RuO₂-based acidic OER catalysts recently reported (Supplementary Table 5), the py-RuO₂:Zn catalyst also ranks the top-level of stability in terms of the Ru dissolution rate. When normalized by the ECSA, a value of $37.6 \text{ pg cm}_{\text{ECSA}}^{-2} \text{h}^{-1}$ for Ru dissolution rate was obtained on the py-RuO₂:Zn catalyst, significantly lower than the $\sim 1.05 \mu\text{g cm}_{\text{ECSA}}^{-2} \text{h}^{-1}$ on commercial RuO₂, indicating an intrinsically improved stability of the catalyst.”

Comments: 8. The catalytic stability of py-RuO₂:Zn for OER needs to be checked at a high current density.

Response: Thanks for this nice comment. The stability test of py-RuO₂:Zn for OER at 100 mA cm^{-2} was performed for 24 h in $0.5 \text{ M H}_2\text{SO}_4$. As shown in Fig. R9, a relatively larger increase in the overpotential, $\sim 70 \text{ mV}$, was observed after the test at 100 mA cm^{-2} , compared with the 15 and 22 mV at 10 and 50 mA cm^{-2} , respectively, indicating an accelerated degradation of the catalyst. The result is consistent with the observation that the dissolution of Ru became faster at potentials above 1.46 V.

Fig. R9 Chronopotentiometric stability tests of py-RuO₂:Zn for OER at 10, 50, and 100 mA cm^{-2} for 24 h in O₂-saturated $0.5 \text{ M H}_2\text{SO}_4$. The dashed line was assigned to a potential of 1.46 V.

Following discussion and Fig. R9 have been added in the revised manuscript and SI file (Supplementary Fig. 29), respectively.

On page 19: “At a higher current density of 50 mA cm^{-2} , py-RuO₂:Zn showed excellent stability over 100 h with an overpotential increase of only 90 mV (Fig. 3e), while potential increase was 70 mV after a test at the current density of 100 mA cm^{-2} for 24 h (Supplementary Fig. 29).”

On page 22: “The accelerated degradation of py-RuO₂:Zn at potentials above 1.46 V was further observed under a CP test at 100 mA cm^{-2} , which exhibited a faster increase of overpotential by 70 mV within 24 h (Supplementary Fig. 29).”

Comments: 9. The Tafel plots should be derived from the steady-state polarization curves (ACS Energy Lett. 2021, 6, 1607).

Response: Thanks for this great suggestion. According to the recommended reference (ACS Energy Lett. 2021, 6, 1607–1611), we reanalyzed the Tafel slopes of OER based on the steady-state polarization curves. As shown in Fig. R10, the derived Tafel slopes were 36.1, 60.8, and 55.7 mV dec⁻¹ on the py-RuO₂:Zn, py-RuO₂, and c-RuO₂ catalysts, respectively. The values are generally consistent with those obtained from the 100%-iR corrected LSV curves, although there are small variations. The py-RuO₂:Zn catalyst still showed the better OER kinetics than the pure RuO₂ catalysts.

Fig. R10 (a) Chronoamperometric responses of the py-RuO₂:Zn, py-RuO₂, and c-RuO₂ catalysts for OER at different potentials. (b) Tafel plots derived from the LSV curves (solid line) and the steady-state polarization curves (scatters). Values in parentheses were derived from steady-state polarization curves.

We have updated the Tafel plots in Fig. 3d in the revised manuscript based on the results in Fig. R10. The steady-state chronoamperometric (CA) responses in Fig. R10a have been added as Supplementary Fig. 26 in the revised SI file. Following discussion has also been added in the revised manuscript.

On page 18: “Fig. 3d shows the Tafel slope analyses for the different catalysts. The plots were derived from the *iR*-corrected LSV curves and the steady-state polarization curves (Supplementary Fig. 26).⁷⁶ Clearly, py-RuO₂:Zn offered the lowest Tafel slope of 38.9 (36.1) mV dec⁻¹, suggesting faster OER kinetics compared to the py-RuO₂ and c-RuO₂ catalysts.^{25,73}”

References cited in this section are as follows:

“25. Harzandi, A. M. et al. Ruthenium core-shell engineering with nickel single atoms for selective oxygen evolution via nondestructive mechanism. *Adv. Energy Mater.* 11, 2003448 (2021).

73. Wen, Y. et al. Stabilizing highly active Ru sites by suppressing lattice oxygen participation in acidic water oxidation. *J. Am. Chem. Soc.* 143, 6482–6490 (2021).

76. Anantharaj, S., Noda, S., Driess, M. & Menezes, P. W. The pitfalls of using potentiodynamic polarization curves for tafel analysis in electrocatalytic water splitting. *ACS Energy Lett.* 6, 1607–1611 (2021).”

Comments: 10. Faradaic efficiency analysis should be performed.

Response: Thanks for this suggestion. The faradaic efficiency (FE) of OER on py-RuO₂:Zn catalyst was measured by the water displacement method under the chronopotentiometric condition at current densities of 25 and 40 mA cm⁻². As shown in Fig. R11, the measured oxygen amount fits well with the theoretical values calculated from Faraday’s law of electrolysis, approaching ~99% and ~100% FE at 25 and 40 mA cm⁻², respectively.

Fig. R11 The FE of OER on py-RuO₂:Zn catalyst as a function of reaction time determined by the water displacement method at current densities of 25 and 40 mA cm⁻². The geometric area of electrode was 1.0 cm².

Discussion as below has been added in the revised manuscript.

On page 17: “The faradaic efficiency (FE) of OER on py-RuO₂:Zn catalyst was measured by the water displacement method under the chronopotentiometric condition at current densities of 25 and 40 mA cm⁻². As shown in Supplementary Fig. 24, the measured oxygen amount fits well with the theoretical values calculated from Faraday’s law of electrolysis, approaching ~99% and ~100% FE at 25 and 40 mA cm⁻², respectively.”

Experimental details of the FE measurement and Fig. R11 shown as Supplementary Fig. 24 have been added in the revised SI file.

In the experimental section of SI file: “Faradaic efficiency (FE) of OER was determined by water displacement method. The FE was calculated by the equation (1):

$$FE(\%) = \frac{n_{\text{exp}}(\text{O}_2)}{n_{\text{theor}}(\text{O}_2)} \times 100 \quad (1)$$

where $n_{\text{exp}}(\text{O}_2)$ and $n_{\text{theor}}(\text{O}_2)$ are experimental and theoretical amount of O₂ produced during the OER

process.

According to the Faraday's law, the $n_{\text{theo}}(\text{O}_2)$ was calculated by the equation (2):

$$n_{\text{theo}}(\text{O}_2) = \frac{I \cdot t}{z \cdot F} \text{ (mol)} \quad (2)$$

where I (A) is the current, t (s) is the OER reaction time, $z = 4$ is the electron transfer number of OER, $F = 96485 \text{ C mol}^{-1}$ is the Faraday constant.

The was determined by a water displacement method and calculated by the ideal gas law:

$$n_{\text{exp}}(\text{O}_2) = \frac{p \cdot V}{R \cdot T} \text{ (mol)} \quad (3)$$

where $p = (101325 - 2813) \text{ Pa}$ is the partial pressure of O_2 produced, V (m^3) is the volume of O_2 produced, $R = 8.314 \text{ J mol}^{-1} \text{ K}^{-1}$ is the ideal gas constant, $T = 293.15 \text{ K}$ is the reaction temperature.”

Comments: 11. The authors need to perform all the OER experiments in oxygen-saturated environments.

Response: Thanks for the comment. All the OER experiments were confirmed to be performed in O_2 -saturated H_2SO_4 , except otherwise stated. This detail was lost in the draft of an earlier version and have been updated in the main text and captions of Figs. 3 and 4 in the revised manuscript.

Comments: 12. The authors in the current manuscript show the fabrication of py-RuO₂:Zn nanowire arrays on carbon fiber paper and fluorine-doped tin oxide glass. The authors also need to show the catalytic performance of py-RuO₂:Zn nanowire arrays on these substrates also. What are the advantages of using a Ti plate over the other substrate?

Response: Thanks for the nice suggestions and comments.

Fig. R12 LSV curves of py-RuO₂:Zn on Ti, CFP, and FTO supports for OER in 0.5 M H_2SO_4 solution with O_2 saturation.

We found that the py-RuO₂:Zn nanowire arrays can be well fabricated on carbon fiber paper (CFP) and fluorine-doped tin oxide glass (FTO) supports in addition to the metallic Ti plate. The OER activity of py-RuO₂:Zn catalyst on different supports is displayed in Fig. R12, which shows a successive increase with the support changing from FTO, CFP, to Ti plate. This support-dependent activity may result from the difference in the electron transfer efficiency at the catalyst-support interface. In addition, as the operation conditions of acidic OER is more corrosive, the CFP will undergo serious degradation at high anodic

potentials in acidic media (Adv. Energy Mater. 2020, 10, 1902494). FTO glass is more stable under the corrosive reaction conditions, but the relatively lower conductivity would hinder the electron transfer efficiency at the interface (Angew. Chem. Int. Ed. 2017, 56, 5994–6021). Therefore, the Ti plate was selected as the support for py-RuO₂:Zn catalyst in this work, which is a widely used dimensionally stable anodes material in chlorine evolution process (Chem. Rev. 2016, 116, 2982–3028).

In the revised manuscript the following discussion has been updated. Fig. R12 and related discussion have been also updated in Supplementary Fig. 8 in the revised SI file.

On page 7: “In addition, the py-RuO₂:Zn nanowire arrays could readily be fabricated on other substrates, such as carbon fiber paper (CFP) and fluorine-doped tin oxide (FTO) glass (Supplementary Fig. 8), highlighting the versatility of one-step pyrolysis catalyst fabrication strategy developed herein.⁵⁰⁻⁵⁴ But both CFP and FTO supported py-RuO₂:Zn catalyst shows a relatively lower OER activity for acidic OER (Supplementary Fig. 8). Therefore, the Ti plate was selected as the support for py-RuO₂:Zn catalyst in this work, which is a widely used DSA material in chlorine evolution process.⁴⁷”

References cited in this section are as follows.

“47. Karlsson, R. K. B. & Cornell, A. Selectivity between oxygen and chlorine evolution in the chlor-alkali and chlorate processes. Chem. Rev. 116, 2982–3028 (2016).

50. Chueh, Y. L. et al. RuO₂ nanowires and RuO₂/TiO₂ core/shell nanowires: From synthesis to mechanical, optical, electrical, and photoconductive properties. Adv. Mater. 19, 143–149 (2007).

51. Kang, M. et al. Single carbon fiber decorated with RuO₂ nanorods as a highly electrocatalytic sensing element. Anal. Chem. 84, 9485–9491 (2012).

52. Chen, Z. G., Pei, F., Pei, Y. T. & De Hosson, J. T. M. A versatile route for the synthesis of single crystalline oxide nanorods: Growth behavior and field emission characteristics. Cryst. Growth. Des. 10, 2585–2590 (2010).

53. Lee, Y. et al. Facile synthesis of single crystalline metallic RuO₂ nanowires and electromigration-induced transport properties. J. Phys. Chem. C 115, 4611–4615 (2011).

54. Kuete Saa, D. et al. Synthesis of RuO₂ nanowires from Ru thin films by atmospheric pressure micro-post-discharge. Surf. Coat. Technol. 295, 13–19 (2016).”

Comments: 13. Schematic illustration of the catalyst fabrication method (Figure 1. (a)) is not up to the standard level for publication.

Response: Thanks for the nice comment. We have revised the schematic illustration of the catalyst fabrication procedure, shown as Fig. R13 that has been updated in Fig. 1a in the revised manuscript.

Fig. R13 Schematic illustration of the catalyst fabrication.

Reviewer #2 (Remarks to the Author):

Comments: I recommend the paper “Construction of Zn-doped RuO₂ nanowires for exceptional efficient and stable water oxidation in acidic media” to be published in Nature Communications after Major Revision. The catalytic performance of the catalyst studied for the OER is very good as it shows acceptable durability for a Ru catalyst. The characterization is also well discussed. However, the authors would need to perform some more electrochemical measurements and correct some details.

Response: We appreciate the reviewer for the positive comments about the article. We have addressed the comments point-by-point as follows.

Comments: 1. Line 101, page 5; Results and Discussion Section: Which are the “unstable species formed” during the synthesis process?

Response: Thanks for the comment. In this work, the Zn-doped RuO₂ nanowires (py-RuO₂:Zn) were prepared by a pyrolysis method at 350 °C in the air. The catalyst precursors were an aqueous solution of RuCl₃ and Zn(NO₃)₂, which was then transformed into a mixed py-RuO₂:Zn and ZnO product under the pyrolysis procedure. Prior to further investigations the mixed product was treated by acidic etching to remove the undesired ZnO species from the py-RuO₂:Zn. Accordingly, the unstable species formed are referred to the ZnO component.

We have revised the sentence in the revised manuscript. The revision is as follows:

On page 6: “Finally, the undesired ZnO component in the product was removed by an acid etching treatment.”

Comments: 2. Line 99-106, page 5; Results and Discussion Section: The mass loading of Ru and Zn in the plates was calculated by comparing the initial RuCl₃ and Zn(NO₃)₂ pipetted on the Ti plate and then subtracted the part (calculated from ICP) lost after the acid leaching, right? There was no loss of the initial precursors-dissolution when it was pipetted onto the freshly etched Ti plate? How did the authors control it?

Response: Thanks for the comments. The mass loadings of Ru and Zn in py-RuO₂:Zn catalysts before and after the acid etching treatment were separately measured by the ICP-MS method. To prepare the analytical solution, 5 mg of the py-RuO₂:Zn powder scraped off the Ti substrate was dispersed in 20 mL solution containing HNO₃, HCl, and HClO₄ with the ratio of 4:12:3, then transferred into a hydrothermal 50 mL Teflon-lined stainless-steel autoclave. Finally, the sample was sealed and treated at 180 °C for 72 h to fully digest all solid parts.

We have updated the corresponding Experimental details in the revised Supplementary Information (SI) file. The revision is as follows:

On page S3 in SI file: “The mass loadings of Ru and Zn in py-RuO₂:Zn catalysts before and after the acid etching treatment were separately measured by the ICP-MS method (Supplementary Table 1). To prepare the analytical solution, 5 mg of the py-RuO₂:Zn powder scraped off the Ti substrate was dispersed in 20 mL solution containing HNO₃, HCl, and HClO₄ with the ratio of 4:12:3, then transferred into a hydrothermal 50 mL Teflon-lined stainless-steel autoclave. Finally, the sample was sealed and treated at 180 °C for 72 h to fully digest all solid parts.”

Comments: 3. Line 168, page 9; I do not understand why the electrochemical characterization proves the existence of Ru^{3+} in the catalyst. The redox peaks on the cyclic voltammetry (CV) curves between 0.5 ~ 0.8 V associated to $\text{Ru}^{3+}/\text{Ru}^{4+}$ couple (Figure S12) appear below the open circuit potential, so it only proves that there is Ru^{4+} in the initial catalyst.

Response: We thank the reviewer very much for the nice comments. We re-examined the CV results in Supplementary Figure 12 (Fig. R14) and confirmed that the explanation on a_1/c_1 and a_2/c_2 redox peaks for the existence of Ru^{3+} is not appropriate, when the disappear of a_3/c_3 redox peaks indicated the suppressed oxidation of Ru^{4+} to higher valent Ru species.

Accordingly, the corresponding sentence: *“The existence of Ru^{3+} species on the surface of catalysts were then examined by the electrochemical studies. The current peaks on cyclic voltammetry (CV) curves between 0.5 ~ 0.8 V were associated with the redox of $\text{Ru}^{3+}/\text{Ru}^{4+}$ couple (Figure S12).”* has been removed from the revised manuscript.

Fig. R14 CV curves of py- $\text{RuO}_2\text{:Zn}$ and py- RuO_2 catalysts in 0.5 M H_2SO_4 solution with a scan rate of 25 mV s^{-1} .

We then analyzed the valence of Ru species according to the XANES at Ru K-edge (Fig. R15a). An average value of +3.4 was found for Ru species in the py- $\text{RuO}_2\text{:Zn}$ catalyst (Fig. R15b), agreeing with the observations of Ru^{3+} and oxygen vacancies in XPS measurements. Furthermore, the Raman spectra was also used to study the relative abundance of Ru^{3+} species on the surface. As shown in Fig. R15c, strong Raman bands at 430 and 588 cm^{-1} were observed on both py- $\text{RuO}_2\text{:Zn}$ and c- RuO_2 catalysts, attributed to the vibration of $\text{Ru}^{4+}\text{-O}$ bonds and $\text{Ru}^{3+}\text{-O}$ bonds of hydrated RuO_2 on the surface (Electrochem. Solid-State Lett. 2005, 8, E39–E41). By normalizing the area intensity of the band at 588 cm^{-1} by that at 430 cm^{-1} , denoted as I_{588}/I_{430} ratio, (Fig. R15d), we found that the py- $\text{RuO}_2\text{:Zn}$ catalyst showed a higher intensity than two others, thereby possessing more Ru^{3+} species on the surface.

Fig. R15 has been added in the revised SI file, shown in Supplementary Figs. 13 and 15. Following discussion as below have been updated in the revised manuscript.

On page 10: *“The higher content of low-valent Ru species on the surface of py- $\text{RuO}_2\text{:Zn}$ catalyst remained under the OER conditions, as confirmed by the Raman measurements (Supplementary Fig. 13).”*

On page 12: *“Calculations on basis of adsorption edge energy revealed that an average oxidation state of Ru species in the catalyst was approximately +3.4 (Supplementary Fig. 15), which was considered as the*

combination of pristine Ru⁴⁺ and Ru³⁺ cations.”

Fig. R15 (a) XANES spectra at Ru K-edge for py-RuO₂:Zn, Ru foil, and c-RuO₂. (b) The edge energies for Ru K-edge as a function of the oxidation state of the Ru. (c) In situ Raman spectra for py-RuO₂:Zn, py-RuO₂, and c-RuO₂ catalysts in O₂-saturated 0.5 M H₂SO₄ at given electrode potentials. (d) Normalized intensity of Raman band at 588 cm⁻¹ to that at 430 cm⁻¹ on the catalysts as a function of applied potential. The areas under the bands were used to calculate the I_{588}/I_{430} ratio.

Comments: 4. The XAS study is interesting, with the observation of Zn-Ru distances similar to Ru-Ru on the RuO₂, and therefore verifying the incorporation of Zn²⁺ in the structure. Could the authors try to explain why even with the introduction of Zn²⁺ in the RuO₂ structure (which is already more reduced than Ru⁴⁺ and therefore has to produce oxygen vacancies) do they think that Ru⁴⁺ is in part also reduced into Ru³⁺? I would expect the opposite effect on Ru⁴⁺ to balance the introduction of Zn²⁺ and keep the electroneutrality of RuO₂ in a more stable structure, even with some oxygen vacancies.

Response: We thank the reviewer very much for the insightful comments. The XAS results indicated that the Zn dopants took an octahedral coordination structure through substitutional doping at Ru sites in the RuO₂ lattice. In principle, to accommodate the divalent metal a fraction of the Ru will be oxidized above +4 (Chem. Mater. 2020, 32, 6150–6160). When oxygen vacancies (V_O) present, the oxidation state of Ruⁿ⁺ ($n > 4$) would be reduced. Recently, Liu and colleagues reported a Na-doped amorphous/crystalline RuO₂ catalyst containing more low-valent Ruⁿ⁺ ($n < 4$) species with the presence of high abundant V_O defects (Angew. Chem. Int. Ed, 2021, 60, 18821–18829).

In this work, more low-valent Ru³⁺ species were found in the py-RuO₂:Zn catalyst, mainly due to the presence of abundant V_O defects. First, according to the XANES result and corresponding oxidation state analysis, Ru element in the py-RuO₂:Zn catalyst possessed an average valence of +3.4, lower than the +4 in pristine RuO₂ (Fig. R15a and b). XRD results revealed that the py-RuO₂:Zn catalyst owned a rutile phase as to the RuO₂. Therefore, the Ru cations were still located at the center of octahedral coordination

structure. In this regard, the reduction in the oxidation state of Ru element should associate with the appearance of V_O defects. On the surface of the py-RuO₂:Zn, more Ru³⁺ species were also observed according to the XPS and Raman measurements (Figs. R16a, b, and R15c, d). The concentrations of Ru³⁺ and V_O defects further showed a good linear relationship (Fig. R16c), suggesting that the presence of V_O defect should account for the reduction in the oxidation state of Ru element. The higher abundance of Ru³⁺ and V_O defects on py-RuO₂:Zn than on py-RuO₂ and c-RuO₂ was probably caused by the Zn doping. To understand the role of Zn doping on the generation of V_O defects, the relationship between Zn content and V_O concentration was analyzed. Figs. R16d, e shows that there was a linear dependence of O_V/O_{L-Ru} on the Zn content (shown as Zn/Ru at.% and the Zn/ O_V at.%), indicating that the doping of Zn element can induce the generation of V_O defects. However, we noticed that the undoped py-RuO₂ catalyst also contained more Ru³⁺ and V_O defects on the surface than the c-RuO₂. Therefore, in addition to the Zn doping, the catalyst synthesis method used here appears to also lead to the generation of V_O defects and the low-valent Ru species. In fact, previous studies proved that the RuO₂ prepared by similar pyrolysis methods contains more coordinately unsaturated Ru sites in lattice, due to the incomplete crystallization of the grains at the relatively lower calcination temperature (J. Phys. Chem. B 1998, 102, 3736–3741; J. Electroanal. Chem. 2002, 532, 25–33). Thus, it can conclude that the Zn doping and the catalysis synthesis method have induced the generation of V_O defects and the associated reduction of Ru oxidation state.

Fig. R16 (a, b) Core-level Ru 3p XPS spectra for as-prepared and post-OER py-RuO₂:Zn catalysts and two pure RuO₂ catalysts. (c) The relationship between concentrations of V_O defect (O_V/O_{L-Ru} ratio) and Ru³⁺ species (Ru^{3+}/Ru^{4+} ratio). (d, e) The dependence of V_O defect (O_V/O_{L-Ru} ratio) on the content of Zn dopant, represented by Zn/Ru at.% and Zn/ O_V at.% values, respectively.

Figs. R16d and e have been added in the revised SI file, shown as Supplementary Fig. 16. Corresponding discussion as below has been updated in the revised manuscript.

On page 12–13: “We note that when the Zn dopants took an octahedral coordination structure through substitutional doping at Ru sites in the RuO₂ lattice, a fraction of the Ru will, in principle, be oxidized above +4 to accommodate the divalent metal, associated with a generation of stoichiometric oxide.³⁰ However, when oxygen vacancies (V_O) present, the oxidation state of Ruⁿ⁺ ($n > 4$) would be reduced. Recently, Liu and colleagues reported a Na-doped amorphous/crystalline RuO₂ catalyst containing more low-valent Ruⁿ⁺ ($n < 4$) species with the presence of high abundant V_O defects.⁴⁰ To further understand the role of Zn doping on the generation of V_O defects, the relationship between Zn content and V_O concentration was analyzed on basis of the XPS results. A linear dependence of O_V/O_{L-Ru} on the Zn content was found (Supplementary Fig. 16), indicating that the doping of Zn element can induce the generation of V_O defects. In the meantime, the presence of Ru³⁺ and V_O defects was also found in the undoped py-RuO₂ catalyst, caused by the catalyst synthesis method used here.^{61,62} Thus, it can conclude that the Zn doping, in addition to the catalysis synthesis method, has induced the generation of V_O defects and the low-valent Ru sites.”

References cited in this section are listed as follows.

“30. Burnett, D. L. et al. (M,Ru)O₂ (M = Mg, Zn, Cu, Ni, Co) rutiles and their use as oxygen evolution electrocatalysts in membrane electrode assemblies under acidic conditions. *Chem. Mater.* 32, 6150–6160 (2020).

40. Zhang, L. et al. Sodium-decorated amorphous/crystalline RuO₂ with rich oxygen vacancies: A robust pH-universal oxygen evolution electrocatalyst. *Angew. Chem. Int. Ed.* 60, 18821–18829 (2021).

61. Doubova, L. M., Daolio, S. & De Battisti, A. Examination of RuO₂ single-crystal surfaces: Charge storage mechanism in H₂SO₄ aqueous solution. *J. Electroanal. Chem.* 532, 25–33 (2002).

62. Arikawa, T., Takasu, Y., Murakami, Y., Asakura, K. & Iwasawa, Y. Characterization of the structure of RuO₂–IrO₂/Ti electrodes by EXAFS. *The Journal of Physical Chemistry B* 102, 3736–3741 (1998).”

Comments: 5. In relation with the previous question, in Figure S14. Why did the authors not include the effect of the introduction of Zn²⁺? Zn²⁺ should introduce more oxygen defects than the presence of Ru³⁺, right?

Response: Thanks for the valuable comments. The relationship between Zn content and V_O concentration was analyzed and displayed in Figs. R16d and e. The relevant data were collected from XPS tests. Clearly, there was a linear dependence of O_V/O_{L-Ru} on the Zn content, indicating that the doping of Zn element can induce the generation of V_O defects. However, a longitudinal interception of the fitted curve in Fig. R16d was found to be 0.96. This means that the undoped RuO₂ would also contain high concentration of V_O defects. The value, interestingly, well matched the measured 0.97 for O_V/O_{L-Ru} on the undoped py-RuO₂ catalyst prepared here. The presence of V_O defect in pure RuO₂ was caused by the relatively lower calcination temperature of catalyst synthesis procedure (*J. Phys. Chem. B* 1998, 102, 3736–3741). In the meantime, the doping of Zn induced a further linear increase of V_O defect, from 0.97 to 1.63 of O_V/O_{L-Ru} as the Zn/Ru at.% increased from 0 to 10.4% on the surface of the py-RuO₂:Zn catalysts. Accordingly, two factors, Zn doping and catalyst synthesis method, have resulted in the generation of V_O defect.

Comments: 6. Figure 3a inset should be plotted bigger in the main text, maybe instead of Figure 3c. That the catalyst can achieve such large current densities at low potentials is a relevant result, so it should be plotted independently and larger and without the capacitance correction, so the readers can see the hysteresis between the anodic and cathodic curves. Also, is it possible to measure (and plot) more than one cycle up to such current densities?

Response: Thanks for the great suggestions and nice comments. Figs. 3a and b have been revised as bellow (Fig. R17), and further updated in the Fig. 3 in the revised manuscript.

Fig. R17 (a) Geometric area and Ru mass normalized LSV curves of py-RuO₂:Zn, py-RuO₂, and c-RuO₂ for OER in 0.5 M H₂SO₄ solution with O₂ saturation. (b) Geometric area and Ru mass normalized LSV curve of py-RuO₂:Zn for OER under high current density.

Fig. R18 (a) Six continuous CV cycling curves of OER on the py-RuO₂:Zn catalyst under the condition when the current density reached higher than 1 A cm⁻². (b) A capacitance correction for the as-measured CV curve of OER.

Six continuous CV cycles of OER on the py-RuO₂:Zn catalyst are shown in Fig. R18a under the condition when the current density reached higher than 1 A cm⁻². No capacitance correction was performed on the CV curves. Clearly, the current density decayed significantly as the test progressed, most probably due to the degradation of the catalyst under the high current density. An obvious hysteresis between the anodic and cathodic curves was further observed. Compared with that found on the CV curves measured under relatively lower current density condition (Fig. R18b), this obvious hysteresis may be also caused by the

degradation of the catalyst under the high current density.

Fig. R18 has been added in the revised SI file, shown as Supplementary Fig. 23. Following discussion has been updated in the revised manuscript.

On page 16: “Such a large current density can be reached more than five continuous CV cycles, but accompanied by a gradually degradation in the OER activity (Supplementary Fig. 23).”

Comments: 7. Are Figure 3a, Figure 3b and Figure S19 already iR corrected? Please, check it and if they are iR corrected change the axis to $E-iR$. The results change a lot if the graphs are corrected or not.

Response: Thanks for the comments. The potential reported in this work has been corrected with iR compensation unless specific statement. We have updated this detail in the experimental section in the revised SI file. The title of the axis in all relevant figures has been changed to “ $E - iR$ ”.

Comments: 8. Figure S19: Could the authors measure more than 30 OER cycles (at least up to 150 mA cm^{-2}) to determine the durability of the catalyst over cycling?

Response: Thanks for the comments. The durability of the $\text{py-RuO}_2\text{:Zn}$ catalyst for OER under the cycling condition has been measured by the CV method up to 170 mA cm^{-2} . The result is displayed in the Fig. R19. A gradually degradation in the OER current density was observed during the 2000 CV cycling measurement, accompanied with an increase in the potential at 100 mA cm^{-2} by about 28 mV. The result indicates a good durability of the $\text{py-RuO}_2\text{:Zn}$ catalyst for the acidic OER process.

Fig. R19 Durability test of the $\text{py-RuO}_2\text{:Zn}$ catalyst for acidic OER under the CV condition up to 2000 cycles. A doubled catalyst loading, about 520 mg per 0.5 cm^2 , was adopted for this measurement. Potential scan rate was 10 mV s^{-1} .

Following discussion has been added in the revised manuscript. Fig. R19 has been added in the revised SI file, shown as Supplementary Fig. 30.

On page 19: “The good stability of the was further investigated under the CV cycling condition. The potential at 100 mA cm^{-2} was increased by about 28 mV after a 2000-cycles test (Supplementary Fig. 30).”

Comments: 9. In Figure S5f, which is the meaning of x ? Is it the value of Zn/Ru? The values seems to be too high compared to the value of Zn/Ru in py-RuO₂:Zn, right?

Response: Thanks for the comments. The “ x ” in Supplementary Fig. 5f (Fig. R20) is referred to the Zn/Ru molar ratio in the precursor solution, equally to the dosage ratio of Zn(NO₃)₂ to RuCl₃ used in the catalyst preparation. The value is much higher than the final content of Zn dopants in py-RuO₂:Zn. This is due to the fact that most of the Zn elements from Zn(NO₃)₂ precursor were converted to the ZnO product under the pyrolytic treatment, rather than being doped into RuO₂ lattice. The unwanted ZnO product was then removed using an acid etching procedure, and the desired py-RuO₂:Zn catalyst was obtained.

Fig. R20 LSV curves for OER in O₂-saturated 0.5 M H₂SO₄ on the catalysts prepared with different Zn/Ru molar ratio in precursor solution. The amount of Ru precursor was fixed at 1 μmol. Electrode area: 0.5 cm².

Following discussion attaching to the Supplementary Fig. 5 has been added in the revised SI file.

On page S11 in SI file: “The “ x ” in Supplementary Fig. 5f is referred to the Zn/Ru molar ratio in the precursor solution, equally to the dosage ratio of Zn(NO₃)₂ to RuCl₃ used in the catalyst preparation. The value is obviously higher than the final content of Zn dopants in py-RuO₂:Zn, due to the fact that most of the Zn elements from Zn(NO₃)₂ precursor were converted to the ZnO product under the pyrolytic treatment, rather than being doped into RuO₂ lattice. The unwanted ZnO composition was removed using an acid etching procedure before the subsequent experiments.”

Comments: 10. Figure S21a: Is the morphology of py-RuO₂ also nanowires. How do the authors explain the huge difference between the ECSA of py-RuO₂:Zn and py-RuO₂?

Response: Thanks for the comments.

Fig. R21 SEM images of (a) py-RuO₂ and (b) py-RuO₂:Zn catalysts. (c) C_{dl} values for OER on the py-RuO₂:Zn, py-RuO₂, and c-RuO₂ catalysts in O₂-saturated 0.5 M H₂SO₄ solution.

The morphology of py-RuO₂ catalyst that is without the doping of Zn element was characterized by SEM method and displayed in Supplementary Fig. 5a (Fig. R21a). Clearly, the py-RuO₂ showed a morphology of aggregation of small nanoparticles, different with the wire-like morphology of Zn-doped counterpart (py-RuO₂:Zn). The difference of morphology has greatly resulted in the huge difference between the ECSA of py-RuO₂:Zn and py-RuO₂, as shown in Supplementary Fig. 25a (Fig. R21c).

In the revised manuscript we have updated this information as bellow.

On page 18: “As shown in Supplementary Fig. 25, py-RuO₂:Zn possessed a much larger ECSA and a higher specific OER activity compared to the pure RuO₂ catalysts studied in this work, largely due to the significant difference in the morphology of them (Supplementary Fig. 5)”

Comments: 11. The authors did not find differences in the Raman bands at 430 and 588 cm⁻¹ between py-RuO₂:Zn and c-RuO₂ catalysts. In principle, they claim that those peaks are associated with the vibration of Ru⁴⁺-O bonds and Ru³⁺-O bonds. So, that means that both catalysts have the same number of reduced Ru³⁺? Then the large number of defects can be related to Zn²⁺ or not?

Response: Thanks for the insightful comments. The in situ Raman results in Supplementary Fig. 13 showed that the bands at 430 and 588 cm⁻¹, assigned to the vibration of Ru⁴⁺-O and Ru³⁺-O bonds, respectively, seemed to maintain generally constant vibration frequency for py-RuO₂:Zn and two pure RuO₂ catalysts (py-RuO₂ and c-RuO₂), indicating that all of the catalysts owned similar low-valent Ru³⁺ species on the surface. But when further comparing the area intensity of band at 588 cm⁻¹ to that at 430 cm⁻¹, a higher value of I₅₈₈/I₄₃₀ has been found on the py-RuO₂:Zn than on two others, suggesting that there were more amount of low-valent Ru species on the surface of py-RuO₂:Zn catalyst (Figs. R22a, b). We have found that the presence of low-valent Ru³⁺ species on py-RuO₂:Zn and py-RuO₂ catalysts, associated with the presence of oxygen vacancy (V_O) defect in the rutile phase oxides, was partly caused by the catalyst preparation method used in this work, while the more higher concentration of Ru³⁺ species on the py-RuO₂:Zn than on the py-RuO₂ should be caused by the Zn doping that has induced more V_O defects in the catalyst (Figs. R22c, d).

Fig. R22 (a) In situ Raman spectra for py-RuO₂:Zn, py-RuO₂, and c-RuO₂ catalysts in 0.5 M H₂SO₄ at given electrode potentials. (b) Normalized intensity of Raman band at 588 cm⁻¹ to that at 430 cm⁻¹ on the catalysts as a function of applied potential. The areas under the bands were used to calculate the I₅₈₈/I₄₃₀ ratio. (c, d) The dependence of V_O defect (O_V/O_{L-Ru} ratio) on the concentration of Zn dopants, represented by Zn/Ru at.% and Zn/O_V at.% values

Reviewer #3 (Remarks to the Author):

Comments: This is my review of the MS titled "Construction of Zn-doped RuO₂ nanowires for exceptionally efficient and stable water oxidation in acidic media" by Baozhong Liu, Siyu Lu, and collaborators. This work describes a Zn-Doped RuO₂ catalyst working in an acidic electrolyte with an ultra-low overpotential of 173 meV and stable for 1000h. The works argue that the origin of this performance comes from Ru +3 and oxygen vacancies (V_O) and that the OOH_{ads} to the O₂ step/barrier is lowered. The formation of Zn-O-Ru is argued to prevent Ru dissolution.

Response: We appreciate the Reviewer for the comments.

Comments: 1. Reading this manuscript, which on the surface is very promising leaves me with many unanswered questions and lots of potential issues.

Response: We appreciate the Reviewer for the positive comments on this work. We have addressed the comments point-by-point as follows.

Comments: 2. The noteworthy results are the synthesis of the nice RuO₂ nanowires and low overpotentials obtained.

Response: Thanks for the positive comments on this work.

Comments: 3. The significance of this work to the field and related fields is that low onset overpotential is observed.

Response: Thanks for the positive comments on this work.

Comments: 4. How does it compare to the established literature? If the work is not original, please provide relevant references.

Many theoretical references and stability studies are omitted. The originality of the work is somehow limited as previous Zn@RuO₂ has been made.

Response: Thanks for the comments.

Our response to the Reviewer's concern on the theoretical analysis of OER performance according to the reported theoretical references.

In this work, an overpotential of 173 mV at 10 mA cm⁻² was observed on the LSV curve of OER for the Zn-doped RuO₂ (py-RuO₂:Zn) catalyst, which was about 200 mV lower than that for the commercial RuO₂ catalyst. Such a low overpotential is impressive because it well exceeds the theoretical limit (~250 mV for the OER overpotential) on the optimal catalyst, following the linear scaling relationships between the adsorption energies of *O, *OH, and *OOH intermediates ($\Delta E_{\text{OOH}} = \Delta E_{\text{OH}} + 3.2 \text{ eV} \pm 0.2 \text{ eV}$) (ChemCatChem 2011, 3, 1159–1165; Phys. Chem. Chem. Phys. 2018, 20, 3813–3818). The result suggested that there may be other pathways of OER on py-RuO₂:Zn in addition to the adsorbate evolving mechanism (ChemCatChem 2011, 3, 1159–1165), at least at the low overpotentials. Recently, Scott and colleagues performed a trace detection of O₂ at low overpotentials of OER using a chip-electrochemistry-mass spectrometry (EC-MS) setup (Energy Environ. Sci., 2022, 15, 1977–1987). They observed an electrochemical generation of O₂ from OER on the RuO_x catalyst at the potential as low as 1.30 V, only 70 mV above the standard thermodynamic potential for water oxidation to oxygen. By further comparing the trends in Ru dissolution and oxygen evolution, they suggested a negligible contribution of lattice oxygen evolution to the overall OER activity for RuO_x in acidic media (Energy Environ. Sci., 2022, 15, 1988–2001).

A comprehensive theoretical study on the recently reported mechanisms of OER revealed that the presence of nonelectrochemical steps (e.g., *OO dimer formation/desorption) tends to increase rather than to reduce the thermodynamic overpotential of OER, while the presence of surface defects (e.g., V_O defects) probably alter the configuration of adsorbed intermediates to improve the OER activity (ACS Catal. 2022, 12, 1433–1442). In this work, a high concentration of V_O defects and low-valent Ru species existed in the py-RuO₂:Zn catalyst, which may play important roles in improving the OER property. When plotting specific current densities against the V_O concentrations, a good linear relationship was established, revealing a clear impact of V_O defects on the OER activity. However, according to the theoretical understanding, the lower oxidation state of Ru and the stronger *OH adsorption, in principle, lead to worse, not better OER activity of RuO₂-based catalysts, on basis of the conventional AEM path of OER proceeding on single Ru center (ChemCatChem 2011, 3, 1159–1165; J. Electroanal. Chem. 2007, 607, 83–89). Given the formation of *OOH species as the rate-determining step of OER on RuO₂, increasing *OH binding will increase the energy barrier of *OOH formation and thus reduce the OER activity, due to the existence of the linear scaling relationships between the adsorption energies of oxygenated intermediates (ChemCatChem 2011, 3, 1159–1165; Phys. Chem. Chem. Phys. 2018, 20, 3813–3818). In accordance with this, recent studies on the OER activity of Ru-based pyrochlores found that the increase in the Ru oxidation state and the reduction of V_O defects led to an enhanced activity, attributed to the weakened binding of reaction intermediates at Ru centers (ACS Catal. 2020, 10, 7734–7746; ACS Catal. 2020, 10, 12182–12196). By comparison with these observations, Kuznetsov and co-authors reported a role of V_O defects in improving the OER activity of Ru-based pyrochlores by evoking the participation of lattice oxygen in the process (J. Am. Chem. Soc. 2020, 142, 7883–7888). We note that, in addition to the py-RuO₂:Zn catalyst reported herein, some other RuO₂-based catalysts containing low-valent Ru species and V_O defects were found to exhibit similar improvement on the OER activity (Angew. Chem. Int. Ed. 2021, 60, 18821–18829; Nat. Commun. 2022, 13, 3784; Energy Environ. Sci. 2022, 15, 1119–1130). The positive role of V_O defects is not limited to RuO₂-based materials, but was also observed on the Ir- and Co₃O₄-based catalysts (ACS Catal. 2019, 9, 6653–6663; J. Am. Chem. Soc. 2020, 142, 12087–12095; J. Am. Chem. Soc. 2020, 142, 18378–18386), although all of them are located on the left branch of the volcano plot for OER between activity and free energy descriptor ($\Delta G_{O} - \Delta G_{OH}$) (ChemCatChem 2011, 3, 1159–1165). Probably, the presence of V_O defects impairs the coordinated symmetry of metal ion centers, which tends to alter the configuration of adsorbed intermediates (ACS Catal. 2022, 12, 1433–1442). Recently, a dual-site oxide path mechanism (OPM) was highlighted, which involves only *O and *OH species as intermediates and allows direct O–O radical coupling for O₂ (Nat. Catal. 2021, 4, 1012–1023).

We have updated the theoretical discussions as bellow in the revised manuscript.

On page 23: “On the LSV curve for OER (Fig. 3a and Supplementary Fig. 22), low onset potential (~1.33 V) and overpotential (173 mV at 10 mA cm⁻²) were observed and have been assigned to an anodic OER process on the py-RuO₂:Zn catalyst. Such low threshold potentials are impressive because they well exceeded the theoretical limit of OER onset overpotential (~250 mV) on the optimal catalyst, based on the adsorbate evolution mechanism (AEM) involving single active metal site and the linear scaling relationships between the adsorption energies of *O, *OH, and *OOH intermediates ($\Delta E_{OOH} = \Delta E_{OH} + 3.2 \text{ eV} \pm 0.2 \text{ eV}$).^{36,81} We then performed experiments using a rotating ring-disk electrode (RRDE) setup and confirmed the explicit contribution of OER process to the observed anodic current at potentials around 1.40 V (Supplementary Fig. 39). Thus, the low threshold potentials of OER suggested that there may be other paths of OER on the py-RuO₂:Zn catalyst in addition to the AEM, especially at low overpotentials. Recently,

Scott and colleagues performed a trace detection of O₂ and found an electrochemical generation of O₂ from OER on the RuO_x catalyst at the potential as low as 1.30 V.⁸² But by comparing the trends in Ru dissolution and oxygen evolution, they suggested a negligible contribution of lattice oxygen evolution to the overall OER activity for RuO_x in acidic media.²² A comprehensive theoretical study on the recently reported mechanisms of OER revealed that the presence of nonelectrochemical steps (e.g., *OO dimer formation/desorption) tends to increase rather than to reduce the thermodynamic overpotential of OER, while the presence of surface defects (e.g., V_O defects) probably alter the configuration of adsorbed intermediates to improve the OER activity.⁸³

On page 24: “However, the lower oxidation state of Ru sites and higher concentration of V_O defects were expected to result in much stronger *OH adsorption and be detrimental to the OER activity of RuO₂-based catalysts, based on the linear scaling relationships between the adsorbates binding energies following conventional AEM path.^{36,81} Accordingly, enhancement on OER activity was achieved when there were high-valent Ru sites and less V_O defects.^{37,38} This seems conflict with our result that an enhanced OER activity was obtained on V_O defects containing Zn-doped RuO₂ catalyst. We speculated that the positive effect of V_O defects on OER activity was realized with the assistance of the Zn dopants. V_O defect and Zn dopants can synergistically regulate the coordinative environment and electronic structure of vicinal Ru centers and thus optimize the binding configurations of OER intermediates.^{40,41,85} Consequently, the OER activity may be improved.”

The references cited in this section are as follows:

22. Scott, S. B. et al. The low overpotential regime of acidic water oxidation part II: Trends in metal and oxygen stability numbers. *Energy Environ. Sci.* 15, 1988–2001 (2022).

36. Man, I. C. et al. Universality in oxygen evolution electrocatalysis on oxide surfaces. *ChemCatChem* 3, 1159–1165 (2011).

37. Gayen, P., Saha, S., Bhattacharyya, K. & Ramani, V. K. Oxidation state and oxygen-vacancy-induced work function controls bifunctional oxygen electrocatalytic activity. *ACS Catal.* 10, 7734–7746 (2020).

38. Hubert, M. A. et al. Acidic oxygen evolution reaction activity–stability relationships in Ru-based pyrochlores. *ACS Catal.* 10, 12182–12196 (2020).

40. Zhang, L. et al. Sodium-decorated amorphous/crystalline RuO₂ with rich oxygen vacancies: A robust pH-universal oxygen evolution electrocatalyst. *Angew. Chem. Int. Ed.* 60, 18821–18829 (2021).

41. Qin, Y. et al. RuO₂ electronic structure and lattice strain dual engineering for enhanced acidic oxygen evolution reaction performance. *Nat. Commun.* 13, 3784 (2022).

81. Rossmeis, J., Qu, Z. W., Zhu, H., Kroes, G. J. & Nørskov, J. K. Electrolysis of water on oxide surfaces. *J. Electroanal. Chem.* 607, 83–89 (2007).

82. Scott, S. B. et al. The low overpotential regime of acidic water oxidation part I: The importance of O₂ detection. *Energy Environ. Sci.* 15, 1977–1987 (2022).

83. Vonrüti, N., Rao, R., Giordano, L., Shao-Horn, Y. & Aschauer, U. Implications of nonelectrochemical reaction steps on the oxygen evolution reaction: Oxygen dimer formation on perovskite oxide and oxynitride surfaces. *ACS Catal.* 12, 1433–1442 (2022).

85. Jin, H. et al. Safeguarding the RuO₂ phase against lattice oxygen oxidation during acidic water electrooxidation. *Energy Environ. Sci.* 15, 1119–1130 (2022). ”

Our response to the Reviewer's concern on the stability analysis of OER on the py-RuO₂:Zn catalyst.

To comprehensively understand the durability of py-RuO₂:Zn catalyst, more stability studies were performed. The cycling durability was measured by the CV method up to 170 mA cm⁻². As shown in Fig. R23a, a gradually degradation in the OER current density was observed during the 2000 CV cycling measurement, accompanied with an increase in the potential at 100 mA cm⁻² by about 28 mV. The result indicates a good durability of the py-RuO₂:Zn catalyst for the acidic OER process. Fig. R23b shows the result of six continuous CV cycles of OER on the py-RuO₂:Zn catalyst under the condition when the current density reached higher than 1 A cm⁻². Clearly, the current density decayed significantly as the test progressed, most probably due to the degradation of the catalyst under the high current density. An obvious hysteresis between the anodic and cathodic curves was further observed, mainly caused by the degradation of the catalyst under the high current density. The stability test of py-RuO₂:Zn for acidic OER at 100 mA cm⁻² was also performed for 24 h. As shown in Fig. R23c, a relatively larger increase in the overpotential, ~70 mV, was observed after the test at 100 mA cm⁻², compared with the 15 and 22 mV at 10 and 50 mA cm⁻², respectively, indicating an accelerated degradation of the catalyst. The result is consistent with the observation that the dissolution of Ru became faster at potentials above 1.46 V. Fig. R23d shows the mass loss analysis of Ru on py-RuO₂:Zn during the stability test at 10 mA cm⁻² for 1000 h in 0.5 M H₂SO₄, as well as the related stability number (S-number). S-number is a recommended metric to quantify the catalyst stability during the reaction (Nat. Catal. 2018, 1, 508–515), defined as the moles of O₂ evolved normalized by the moles of Ru dissolved, i.e., the O₂ evolved/Ru_{dissolved} in molar ratio. The calculated S-number exhibited an increase in the initial 500 h and then a decrease in the following 500 h. A top value of ~6 × 10⁴ was obtained, which is comparable to those observed on Ru-based pyrochlores (ACS Catal. 2020, 10, 12182–12196).

Fig. R23 (a) Durability test of the py-RuO₂:Zn catalyst for acidic OER under the CV condition up to 2000 cycles in 0.5 M H₂SO₄. A doubled catalyst loading, about 520 mg per 0.5 cm², was adopted for this measurement. Potential scan rate was 10 mV s⁻¹. (b) Six continuous CV cycling curves of OER on the py-RuO₂:Zn catalyst under the condition when the current density reached higher than 1 A cm⁻². (c) Chronopotentiometric stability tests of py-RuO₂:Zn for OER at 10, 50, and 100 mA cm⁻² for 24 h in 0.5 M H₂SO₄. The dashed line was assigned to a potential of 1.46 V. (d) mass loss analysis of Ru and corresponding stability number (S-number) on py-RuO₂:Zn during the stability test at 10 mA cm⁻² for 1000 h in 0.5 M H₂SO₄.

Figs. R23a–c have been added in the revised SI file, shown as Supplementary Figs. 23, 30, and 31. Fig. R22d has been updated in Fig. 3 in the revised manuscript. Following discussions have been added in the

revised manuscript.

On page 19: “At a higher current density of 50 mA cm^{-2} , py-RuO₂:Zn showed excellent stability over 100 h with an overpotential increase of only 90 mV (Fig. 3e), while potential increase was 70 mV after a test at the current density of 100 mA cm^{-2} for 24 h (Supplementary Fig. 29). The good stability of py-RuO₂:Zn was further investigated under the CV cycling condition. The potential at 100 mA cm^{-2} was increased by about 28 mV after a 2000-cycles test (Supplementary Fig. 30).”

On page 16: “Such a large current density can be reached more than five continuous CV cycles, but accompanied by a gradually degradation in the OER activity (Supplementary Fig. 23).”

On page 20: “In addition, stability number (S-number), a recommended metric to quantify the catalyst stability during the reaction,⁷⁸ was calculated by normalizing the moles of O₂ evolved ($n_{\text{O}_2 \text{ evolved}}$) with the moles of Ru dissolved ($n_{\text{Ru dissolved}}$), *i.e.*, $S\text{-number} = n_{\text{O}_2 \text{ evolved}}/n_{\text{Ru dissolved}}$.³⁸ As shown in Fig. 3e lower plot, the S-number exhibited an increase in the initial 500 h and then a decrease in the following 500 h. A top value of $\sim 6 \times 10^4$ was obtained, which is comparable to those observed on Ru-based pyrochlores.³⁸”

The references cited in this section are as follows:

“38. Hubert, M. A. et al. Acidic oxygen evolution reaction activity–stability relationships in Ru-based pyrochlores. ACS Catal. 10, 12182–12196 (2020).

78. Geiger, S. et al. The stability number as a metric for electrocatalyst stability benchmarking. Nat. Catal. 1, 508–515 (2018).”

Our response to the Reviewer’s concern on the novelty of the py-RuO₂:Zn catalyst for OER compared with the previously reported Zn-doped RuO₂ counterparts.

We also noticed that Zn-doped RuO₂ oxides have been previously reported as promising OER catalysts in acidic media (Chem. Mater. 2020, 32, 6150–6160; J. Mater. Chem. A 2022, 10, 16193–16203; ChemNanoMat 2021, 7, 117-121). Burnett and co-authors constructed a stoichiometric Zn-doped RuO₂ oxide that contained high-valent Ru^{*n*+} (*n* > 4) sites to balance charge (Chem. Mater. 2020, 32, 6150–6160). The oxides showed a clearly enhanced activity for OER compared with the crystalline RuO₂, probably due to the increase in the Ru oxidation state. Surface evolution of Zn-doped RuO₂ under the reaction enabled a construction of surface defects (e.g., V_O defects) and active Ru sites (J. Mater. Chem. A 2022, 10, 16193–16203), consistent with the theoretically predicted results on RuO₂ catalyst (J. Phys. Chem. C 2017, 121, 18516–18524). A low overpotential of 190 mV and a good stability up to 60 h were observed at the current density of 10 mA cm^{-2} on this surface etched catalyst. By comparison, the proposed Zn-doped RuO₂ (py-RuO₂:Zn) catalyst in this work exhibited much better OER performance, an overpotential of 173 mV at 10 mA cm^{-2} and a stability reaching 1000 h, than those reported analogues. The regular wire-like morphology was found to induce a greatly increased abundance of the Ru active sites. Theoretical calculations further revealed that V_O defects and Zn dopants caused an alleviated binding of oxygen adsorbates at active Ru centers and, more interestingly, enabled a moderate adsorption of *OH species on Zn sites. Consequently, a Ru–Zn dual-site oxide path of OER was favored and significantly enhanced the OER activity. Present work provides a case study on the effects of dopants and surface defects in regulating the OER property of Ru-based catalysts.

The corresponding discussion as bellow has been added in the revised manuscript.

On page 18: “Recently, a surface evolution of Zn-doped RuO₂ under the reaction was found to enable a construction of surface defects (e.g., V_O defects) and active Ru sites,⁷⁵ consistent with the theoretically predicted results on RuO₂ catalyst.³⁹ A low overpotential of 190 mV and a good stability up to 60 h were

observed at the current density of 10 mA cm^{-2} on this surface etched catalyst.”

The references cited in this section are as follows:

“39. Dickens, C. F. & Nørskov, J. K. A theoretical investigation into the role of surface defects for oxygen evolution on RuO_2 . *J. Phys. Chem. C* 121, 18516–18524 (2017).

75. Zhou, Y.-N. et al. Surface evolution of Zn doped- RuO_2 under different etching methods towards acidic oxygen evolution. *J. Mater. Chem. A* 10, 16193–16203 (2022).”

Comments: 5. Does the work support the conclusions and claims, or is additional evidence needed?

The characterization is somewhat supporting the claim of Ru+3, but again the XPS signal is quite small and XAS Ru+3 signal doesn't exist. The whole DFT part and claims there are simply unsupported in the data. The stronger OH^* adsorption will lead to worse, not better OER as concluded by the authors, as binding OH^* too strongly will prohibit the $\text{OOH} \rightarrow \text{O}_2$ step, which is in direct conflict with the author's claims. Where is the detailed view of why the $\text{OOH} \rightarrow \text{O}_2$ step is suddenly lowered, while OH^* and therefore OOH^* bind stronger?

Response: Thanks for the insightful comments.

Fig. R24 (a) XANES spectra at Ru K-edge for py-RuO₂:Zn, Ru foil, and c-RuO₂. (b) The edge energies for Ru K-edge as a function of the oxidation state of the Ru. (c) In situ Raman spectra for py-RuO₂:Zn, py-RuO₂, and c-RuO₂ catalysts in 0.5 M H₂SO₄ at given electrode potentials. (d) Normalized intensity of Raman band at 588 cm⁻¹ by that at 430 cm⁻¹ on the catalysts as a function of applied potential. The areas under the bands were used to calculate the I_{588}/I_{430} ratio.

In addition to the XPS evidence of low-valent Ru ions, the XANES at Ru K-edge then revealed an average oxidation state of +3.4 for Ru species in the py-RuO₂:Zn catalyst, by fitting the related adsorption edge energies (Fig. R24a, b). Furthermore, we performed a Raman test to study the relative abundance of Ru³⁺ species on the surface. Strong Raman bands at 430 and 588 cm⁻¹ were observed on both py-RuO₂:Zn and

c-RuO₂ catalysts (Fig. R24c), attributed to the vibrations of Ru⁴⁺–O bonds and Ru³⁺–O bonds of hydrated RuO₂ on the surface (Electrochem. Solid-State Lett. 2005, 8, E39–E41). By normalizing the area intensity of the band at 588 cm⁻¹ with that at 430 cm⁻¹, denoted as the I_{588}/I_{430} ratio (Fig. R24d), we found a relatively higher I_{588}/I_{430} ratio on the py-RuO₂:Zn catalyst than on two others, indicating more Ru³⁺ species on the surface. Thus, these results support the presence of low-valent Ru sites in a relatively higher content on the py-RuO₂:Zn catalyst.

Fig. R24 has been added in the revised SI file, shown in Supplementary Figs. 13 and 15. Following discussion has been updated in the revised manuscript.

On page 10: “The higher content of low-valent Ru species on the surface of py-RuO₂:Zn catalyst remained under the OER conditions, as confirmed by the Raman measurements (Supplementary Fig. 13).”

On page 12: “Calculations on basis of adsorption edge energy revealed that an average oxidation state of Ru species in the catalyst was approximately +3.4 (Supplementary Fig. 15), which was considered as the combination of pristine Ru⁴⁺ and Ru³⁺ cations.”

The experimental results revealed that the py-RuO₂:Zn catalyst contained a higher content of oxygen vacancy (V_O) defects and low-valent Ru species, which were regarded as the factors in enhancing the OER activity. However, according to the theoretical understanding, the lower oxidation state of Ru and the stronger *OH adsorption will, in principle, lead to worse, not better OER activity of RuO₂-based catalysts, on the basis of conventional OER mechanism involving four proton-coupled electron transfer steps on metal cation centers (J. Electroanal. Chem. 2007, 607, 83–89; ChemCatChem 2011, 3, 1159–1165). Given the formation of *OOH species as the rate-determining step of OER on RuO₂, increasing *OH binding will increase the energy barrier of *OOH formation and thus reduce the OER activity, due to the existence of the linear scaling relationships between the adsorption energies of oxygenated intermediates ($\Delta E_{\text{OOH}} = \Delta E_{\text{OH}} + 3.2 \text{ eV} \pm 0.2 \text{ eV}$). In accordance with this, recent studies on the OER activity of Ru-based pyrochlores found that the increase in the Ru oxidation state and the reduction of V_O defects led to an enhanced activity, attributed to the weakened binding of reaction intermediates at Ru centers (ACS Catal. 2020, 10, 12182–12196; ACS Catal. 2020, 10, 7734–7746). By comparison, Kuznetsov and co-authors reported a role of V_O defects in improving the OER activity of Ru-based pyrochlores by evoking the participation of lattice oxygen in the process (J. Am. Chem. Soc. 2020, 142, 7883–7888). We note that, in addition to the py-RuO₂:Zn catalyst reported herein, some other RuO₂-based catalysts containing low-valent Ru species and V_O defects were found to exhibit similar improvement on the OER activity (Angew. Chem. Int. Ed. 2021, 60, 18821–18829; Nat. Commun. 2022, 13, 3784; Energy Environ. Sci. 2022, 15, 1119–1130).

To understand the Zn doping and oxygen vacancies effect, we redid all the DFT calculations using a surface which is more stable and consistent with previous reports (J. Phys. Chem. C 2015, 119, 4827–4833, J. Phys. Chem. C 2017, 121, 18516–18524, Angew. Chem. Int. Ed. 2021, 60, 2–11, Nat. Commun. 2022, 13, 3784). Fig. R25 shows the optimized clean RuO₂ (110) surface and three others with the presence of V_O defects and/or Zn dopants. Zn was found to be more stably doped at the coordinatively unsaturated Ru (Ru_{cus}) position than the fully coordinated bridge Ru (Ru_{bri}) site, while the bridge row O could form stable vacancy sites. Fig. R26 shows the corresponding Bader charge analysis of relevant Ru, Zn, O sites and the free energy of *OH formation at Ru_{cus} sites on different surfaces. Clearly, by comparing RuO₂_V_O with RuO₂, the presence of bridged V_O defects causes a charge density increase at both the vicinal Ru_{bri} and Ru_{cus} sites, associated with a reduction of Ru valence, which then enhances the binding of OH adsorbates at Ru_{cus} centers ($\Delta G_{\text{OH}} = 0.70 \text{ eV}$). Therefore, the presence of V_O defects is harmful to the OER proceeding on RuO₂,

agreeing with the reports (ACS Catal. 2020, 10, 12182–12196; ACS Catal. 2020, 10, 7734–7746). By comparison, on the surface of stoichiometric RuO₂:Zn oxide, the doping of Zn at Ru_{CUS} sites induces a reduction of the charge density at Ru sites, consistent with the knowledge that a fraction of the Ru will be oxidized above +4 to accommodate the divalent Zn metal (Chem. Mater. 2020, 32, 6150–6160). As a result, the OH binding is weakened ($\Delta G_{OH} = 1.01$ eV), resulting an improvement in OER activity. For the RuO₂:Zn_V_O, the presence of both the Zn dopants and V_O defects can synergistically regulate the electronic structure of Ru centers and the associated OH binding strength ($\Delta G_{OH} = 0.88$ eV), thereby enhancing the OER activity.

Fig. R25 Optimized structures of (a) pristine RuO₂ (110) surface (side view, top view, and Ru sites with different coordination), (b) Zn doped RuO₂ (RuO₂:Zn) surface, and V_O-containing RuO₂:Zn (RuO₂:Zn_V_O) surface.

Fig. R26 (a–d) Bader charge analysis at Ru (brown), Zn (dark cyan), and oxygen (red) sites on different sample surface. (e) Calculated free energy diagram of the *OH formation at Ru_{CUS} sites on different sample surface.

Fig. R25 and R26 have been added in the revised SI file, shown as Supplementary Figs. 42 and 45. *DFT discussion and Fig. 5 have been updated in the revised manuscript.*

On page 24–26: “To understand the Zn doping and oxygen vacancies effect on the OER activity, density

functional theory (DFT) calculations were performed. The Zn doped RuO₂ (RuO₂:Zn) and with O vacancies (RuO₂:Zn_V_O) were built on the optimized RuO₂ (110) surfaces (Supplementary Fig. 41). Zn was found to be more stably doped at the coordinatively unsaturated Ru (Ru_{cus}) position than the fully coordinated bridge Ru (Ru_{bri}) site, while the bridge row O could form stable vacancy site. Then, different OER paths were investigated to determine the preferred reaction pathways, including the AEM and lattice oxygen mechanism (LOM), as well as the recently highlighted dual-site oxide path mechanism (OPM) (Supplementary Fig. 42).^{35,83} The adsorption energies of reaction intermediates were summarized in the Supplementary Table 6. For clean RuO₂, stronger binding of OH adsorbates ($\Delta G_{\text{OH}} = 0.82$ eV) resulted in the OER proceeding favorably via a AEM path, following the four-proton-coupled electron transfer steps as H₂O → *OH → *O → *OOH → O₂.³⁶ The formation of *OOH is the rate-determining step (RDS) with a large free energies barrier of 2.10 eV. By comparison, the LOM and dual site OPM paths are suppressed with much higher energy barriers of RDS (ΔG_{max} for LOM 3.79 eV and OPM 2.48 eV, where ΔG_{max} is the maximum free energy differences among the primary proton-coupled electron transfer steps) (Supplementary Fig. 43). For RuO₂_V_O, the presence of bridged O vacancies caused accumulated charge density at both the vicinal Ru_{bri} and Ru_{cus} sites (Supplementary Fig. 44), which then enhanced the binding of *OH at Ru_{cus} centers ($\Delta G_{\text{OH}} = 0.70$ eV) and induced a larger free energies barrier of 2.28 eV for *OOH formation (Supplementary Fig. 45). Therefore, the presence of V_O defects is harmful to the OER proceeding on RuO₂.^{37,38} In contrast, on the surface of stoichiometric RuO₂:Zn oxide, the doping of Zn at Ru_{cus} sites induced a reduction of the charge density at Ru centers, which agreed with the knowledge that a fraction of the Ru will be oxidized above +4 to accommodate the divalent Zn metal.³⁰ As a result, the *OH binding is weakened ($\Delta G_{\text{OH}} = 1.01$ eV) and the OER activity is improved. More interestingly, a Ru–Zn dual-site OPM appeared to be more favorable with a lower ΔG_{max} of 1.91 eV for *O_{Ru} → *O_{Ru...*}OH_{Zn} of the third proton-coupled electron transfer step, caused by the different binding strength of intermediates on the two sites (Supplementary Fig. 46). The density of states (DOS) and charge density difference suggested that Zn donated some electron to the O and Zn had a lower d band center than Ru (Fig. 5c–e). Therefore, Zn showed weaker absorption of *O, *OH, and *OOH. For example, Zn sites had a ΔG_{OH} of 1.77 eV, while Ru site had had a ΔG_{OH} of 1.01 eV. This would ease the formation of second *O. In addition, the charge difference between Zn and Ru also played an important role in promoting the OER, which resulted in a ~0.1 e charge difference for the two absorbed *O on Zn and Ru and thus promoted the formation of O–O coupling, and eventually the formation of O₂ (Fig. 5d). With the presence of V_O defects, the charge density at both the Ru_{cus} and Zn_{cus} sites on RuO₂:Zn_V_O surface is slightly increased (Supplementary Fig. 44), associated with a shift of Ru d band center away from Fermi, which further optimized the absorption of intermediates (Fig. 5e). Consequently, the ΔG_{max} (*O_{Ru} → *O_{Ru...*}OH_{Zn}) of OPM is further decreased to 1.84 eV for RuO₂:Zn with V_O defects (Fig. 5b). Therefore, we believed that the down shift of Fermi by O vacancy, the weaker absorption of *OH on Zn and the charge difference of Zn and Ru synergistically lowered the OER overpotential ($\eta = \Delta G_{\text{max}} - 2.13$) from 0.87 V for RuO₂ to 0.61 V for the O vacancy-containing Zn doped RuO₂, by converting the OER path from the single-site AEM to the dual-site OPM (Fig. 5a, b)."

References cited in this section are listed as follows.

30. Burnett, D. L. *et al.* (M,Ru)O₂ (M = Mg, Zn, Cu, Ni, Co) rutiles and their use as oxygen evolution electrocatalysts in membrane electrode assemblies under acidic conditions. *Chem. Mater.* **32**, 6150–6160 (2020).

35. Lin, C. *et al.* In-situ reconstructed Ru atom array on α -MnO₂ with enhanced performance for acidic water oxidation. *Nat. Catal.* **4**, 1012–1023 (2021).

36. Man, I. C. *et al.* Universality in oxygen evolution electrocatalysis on oxide surfaces. *ChemCatChem* **3**, 1159–1165 (2011).
37. Gayen, P., Saha, S., Bhattacharyya, K. & Ramani, V. K. Oxidation state and oxygen-vacancy-induced work function controls bifunctional oxygen electrocatalytic activity. *ACS Catal.* **10**, 7734–7746 (2020).
38. Hubert, M. A. *et al.* Acidic oxygen evolution reaction activity–stability relationships in Ru-based pyrochlores. *ACS Catal.* **10**, 12182–12196 (2020).
83. Vonrüti, N., Rao, R., Giordano, L., Shao-Horn, Y. & Aschauer, U. Implications of nonelectrochemical reaction steps on the oxygen evolution reaction: Oxygen dimer formation on perovskite oxide and oxynitride surfaces. *ACS Catal.* **12**, 1433–1442 (2022).”

Comments: 6. The lower the oxidation state of Ru, or when binding at the vacancy sites will lead to much stronger OH* binding and higher overpotentials. See 10.1021/acs.jpcc.7b03481. In fact, higher oxidation state of Ru leads to better activity. (10.1021/acs.jpcc.7b03481, 10.1021/acscatal.0c02252). This is in stark contrast to what the authors find. Also, authors need to provide detailed dGs for each OER step in a table format. Why the calculated OER overpotentials are so high? They should be in the 0.5 to 0.3 V range as in all these other studies. Lastly, authors need to show how/where the Zn dopant is more stable with vacancy present as opposed to without vacancy. If authors cannot calculate lower overpotentials and show calculated dGs w. structures, or to show a more stable Zn in presence of V_O, I request to remove the whole DFT part.

Response: Thanks for the insightful comments. We have redone all the calculations using a surface which is more stable and consistent with previous report (J. Phys. Chem. C 2015, 119, 4827–4833, J. Phys. Chem. C 2017, 121, 18516–18524, Angew. Chem. Int. Ed. 2021, 60, 2–11, Nat. Commun. 2022, 13, 3784). As shown in Fig. R27a, the RuO₂ (110) surface was used. The Zn doped RuO₂ (RuO₂:Zn) and with O vacancies (RuO₂:Zn_V_O) were built on the optimized RuO₂ (110) surfaces. Zn was found to be more stably when doped at the coordinatively unsaturated Ru (Ru_{CUS}) position than the fully coordinated bridge Ru (Ru_{BR}) site, while the bridge row O could form stable vacancy site. The total energies of Zn doping and V_O at different location was compared in the Figs. R27b and c. All these optimized surfaces were further used for OER simulations. The bridged Ru_{BR} with V_O was not the active reaction center. The Ru and Zn atoms on the CUS rows were the active center. Figs. R27d–g show the corresponding Bader charge analysis of relevant Ru, Zn, O sites and the free energy of *OH formation at Ru_{CUS} sites on different surfaces. Clearly, by comparing RuO₂_V_O with RuO₂, the presence of bridged V_O defects causes a charge density increase at both the vicinal Ru_{BR} and Ru_{CUS} sites, associated with a reduction of Ru valence, which then enhances the binding of OH adsorbates at Ru_{CUS} centers ($\Delta G_{OH} = 0.70$ eV). Therefore, the presence of V_O defects is harmful to the OER proceeding on RuO₂, agreeing with the reports (ACS Catal. 2020, 10, 12182–12196; ACS Catal. 2020, 10, 7734–7746). By comparison, on the surface of stoichiometric RuO₂:Zn oxide, the doping of Zn at Ru_{CUS} sites induces a reduction of the charge density at Ru sites, consistent with the knowledge that a fraction of the Ru will be oxidized above +4 to accommodate the divalent Zn metal (Chem. Mater. 2020, 32, 6150–6160). As a result, the OH binding is weakened ($\Delta G_{OH} = 1.01$ eV), resulting an improvement in OER activity. For the RuO₂:Zn_V_O, the presence of both the Zn dopants and V_O defects can synergistically regulate the electronic structure of Ru centers and the associated OH binding strength ($\Delta G_{OH} = 0.88$ eV), thereby enhancing the OER activity.

In addition, to unveil the OER mechanism on the py-RuO₂:Zn catalyst, we have compared the AEM and LOM, as well as the recently highlighted dual site oxide path mechanism (OPM). The detailed reaction steps

and related free energy changes are summarized in Table R1 and R2, respectively. We found that the Zn doping induced a preferred dual-site OPM path of OER. A relatively lower overpotential was obtained on the V_O-containing Zn-doped RuO₂ surface.

Fig. R27 Optimized structures of (a) pristine RuO₂ (110) surface (side view, top view, and Ru sites with different coordination), (b) Zn doped RuO₂ (RuO₂:Zn) surface, and V_O-containing RuO₂:Zn (RuO₂:Zn_V_O) surface. (d–g) Bader charge analysis at Ru (brown), Zn (dark cyan), and oxygen (red) sites on different sample surface. (h) Calculated free energy diagram of the *OH formation at Ru_{CUS} sites on different sample surface.

Table R1 Mechanisms of OER process.

AEM	OPM	LOM
$H_2O + * \rightarrow *OH + H^+ + e^-$	$H_2O + * \rightarrow *OH + H^+ + e^-$	$H_2O + * \rightarrow *OH + H^+ + e^-$
$*OH \rightarrow *O + H^+ + e^-$	$*OH \rightarrow *O + H^+ + e^-$	$*OH \rightarrow *O + H^+ + e^-$
$*O + H_2O \rightarrow *OOH + H^+ + e^-$	$*O + H_2O \rightarrow *O...*OH + H^+ + e^- \dots\dots\dots OPM_1$	$*O + M-O \rightarrow O_2 + *$
$*OOH \rightarrow O_2 + H^+ + e^-$	$*OH + H_2O \rightarrow *OH...*OH + H^+ + e^-$	$M-Vac + H_2O \rightarrow *OH + H^+ + e^-$
	$*OH...*OH \rightarrow *O...*OH + H^+ + e^- \dots\dots\dots OPM_2$	$*OH \rightarrow M-O + H^+ + e^-$
	$*O...*OH \rightarrow *O...*O + H^+ + e^-$	
	$*O...*O \rightarrow O_2 + *$	

Table R2 Adsorption energies of reaction intermediates.

	ΔOH (eV)	ΔO (eV)	$\Delta\text{OH}\dots\text{OH}$ for OPM_2 (eV)	ΔOOH (eV)	$\Delta\text{O}\dots\text{OH}$ for OPM_1 (eV)	ΔH for LOM (eV)	ΔG_{max} (eV)	OP (V)
RuO ₂	0.82	1.79	1.41	3.89	2.44	1.13	2.10	0.87
ZnRuO ₂	1.01	1.89	1.58	3.99	3.80	0.53	1.91	0.68
ZnRuO ₂ _Vo	0.88	1.73	1.55	3.60	3.57	1.54	1.84	0.61
RuO ₂ _Vo	0.70	1.64		3.92			2.28	1.05

Fig. R27 has been added in the revised SI file, shown as Supplementary Figs. 42 and 45. Tables R1 and R2 have also been added in the revised SI file, shown as Supplementary Tables 6 and 7. *Whole DFT discussion has been updated and highlighted in red on page 24–26 in the revised manuscript.*

The recommended references cited in this section are listed as follows.

“38. Hubert, M. A. et al. Acidic oxygen evolution reaction activity–stability relationships in Ru-based pyrochlores. *ACS Catal.* 10, 12182–12196 (2020).

39. Dickens, C. F. & Nørskov, J. K. A theoretical investigation into the role of surface defects for oxygen evolution on RuO₂. *J. Phys. Chem. C* 121, 18516–18524 (2017).”

Comments: 7. Are there any flaws in the data analysis, interpretation and conclusions? Do these prohibit the publication or require revision?

The stability window is limited to 1.46 eV above which the catalyst dissolves! This is well known for all RuO₂-containing compounds. So far none of the works was able to fix this problem. (pls cite these works such as <https://doi.org/10.1021/ja510442p> or)

Response: Thanks for the comments. In this work, the stability study revealed an accelerated degradation of the py-RuO₂:Zn catalyst above the potential of 1.46 V. The result agrees with the previous researches on the stability window of RuO₂-based catalysts (*J. Am. Chem. Soc.* 2015, 137, 4347–4357; *Catal. Today* 2016, 262, 170–180; *npj Comput. Mater.* 2020, 6, 160). We note that the stability of py-RuO₂:Zn did not obviously break the reported potential limit, but the onset overpotential of OER was significantly reduced, providing a widened stability window to the application of py-RuO₂:Zn.

Following discussion has been added in the revised manuscript.

On page 22: “The result agrees with the previous reports on the stability window of RuO₂-based catalysts.^{6,79,80} Although the stability of py-RuO₂:Zn did not obviously break the reported potential limit, the onset overpotential of OER was significantly reduced, providing a widened stability window to the application of py-RuO₂:Zn.”

The recommended references cited in this section are listed as follows.

“6. McCrory, C. C. L. et al. Benchmarking hydrogen evolving reaction and oxygen evolving reaction electrocatalysts for solar water splitting devices. *J. Am. Chem. Soc.* 137, 4347–4357 (2015).

79. Cherevko, S. et al. Oxygen and hydrogen evolution reactions on Ru, RuO₂, Ir, and IrO₂ thin film electrodes in acidic and alkaline electrolytes: A comparative study on activity and stability. *Catalysis Today* 262, 170–180 (2016).

80. Wang, Z., Guo, X., Montoya, J. & Nørskov, J. K. Predicting aqueous stability of solid with computed pourbaix diagram using scan functional. *npj Comput. Mater.* 6, 160 (2020).”

Fig. R28 (a) CV curves of py-RuO₂:Zn, py-RuO₂, and c-RuO₂ catalysts in 0.5 M H₂SO₄ solution with a potential scan rate of 50 mV s⁻¹. (b) ICOHP analysis of Ru–O, Ru···Ru, Ru···Zn, and Zn–O on the surfaces of RuO₂, RuO₂:Zn, and RuO₂:Zn_V_O. (c) De-metallization energies of Ru from RuO₂, and Ru and Zn from RuO₂:Zn_V_O.

Further, we performed more experimental and theoretical investigations to deeply understand the enhanced stability of VO-containing Zn-doped RuO₂. Fig. R28a shows the electrochemical redox features of the Ru species on py-RuO₂:Zn, and pure py-RuO₂ and c-RuO₂ catalysts in potential regions preceding OER process (Supplementary Fig. 47). Compared with those on py-RuO₂ and c-RuO₂, the redox peaks of Ru⁴⁺/Ruⁿ⁺ ($n > 4$) above 1.2 V were significantly suppressed on py-RuO₂:Zn, indicating an efficient protection on Ru cations from over oxidation to soluble species. Consequently, the catalytic stability of py-RuO₂:Zn for OER would be enhanced. Fig. R28b shows the crystal orbital Hamilton population (COHP) analysis of Ru–O and Zn–O bonds, as well as Ru···Ru and Ru···Zn metal couplings on the optimized RuO₂, RuO₂:Zn, and RuO₂:Zn_V_O surfaces. The integrated COHP (ICOHP) values of Ru_{CUS}–O for RuO₂:Zn, and RuO₂:Zn_V_O are –3.40 eV and –2.44 eV, respectively, which have been negatively shifted from that for pristine RuO₂ (–2.25 eV), thereby revealing a strengthened Ru_{CUS}–O bond on those Zn-doped samples. In addition, small negative ICOHP values of Ru_{CUS}···Zn were found on both the RuO₂:Zn (–0.04 eV) and RuO₂:Zn_V_O (–0.03 eV) with the Zn doping, indicating a weak long range orbital coupling between Zn dopants and the vicinal Ru_{CUS} sites. In contrast, there is no clear interaction of Ru_{CUS}···Ru_{CUS} (0.08 eV for ICOHP) on the pristine RuO₂. Accordingly, the Ru_{CUS} sites would be further stabilized by the Zn dopants. When bridged V_O defects present, the ICOHP of Ru_{brl}···Ru_{brl} for RuO₂:Zn_V_O also acquired a small negative value of –0.01 eV, while it was a positive value of 0.12 eV on both the RuO₂ and RuO₂:Zn to –0.01 eV on RuO₂:Zn_V_O. This indicated an enhanced interaction between two adjacent Ru_{brl} sites in the vicinity of V_O defect. The enhanced stability of Zn doped RuO₂ with Vo is also demonstrated by the de-metallization energies of Ru and Zn (Fig. R28c). The doping of Zn induced an increased de-metallization energy of Ru by around 0.5 eV and thus stabilized the RuO₂. The

Zn dopants themselves possessed relatively higher de-metallization energies by around 0.2 eV than the Ru in RuO₂:Zn_V_O. The overall results suggested that the RuO₂ structure become more stable after the introduction of Zn dopants and V_O defects.

Fig. R28a has been added in the revised SI file, shown as the Supplementary Fig. 47. Figs R28b and c have been updated in Fig. 5 in the revised manuscript, shown as Figs. 5f and g. Discussion as bellow has also added in the revised manuscript.

On page 26-27: “In terms of the stability enhancement, the present dual-site OPM path of OER avoids the step of *O → *OOH, which generally proceeds above 1.3 V on single Ru site.⁸⁶ Thus, it was possible to stabilize the OER active sites against the excessive oxidation under the OPM path. We then studied the electrochemical redox features of the Ru species on py-RuO₂:Zn, and pure py-RuO₂ and c-RuO₂ catalysts in potential regions preceding OER process (Supplementary Fig. 47). Compared with those on py-RuO₂ and c-RuO₂, the redox peaks of Ru⁴⁺/ Ruⁿ⁺ (n > 4) above 1.2 V were significantly suppressed on py-RuO₂:Zn, indicating an efficient protection on Ru cations from over oxidation to soluble species.^{21,87,88} Consequently, the catalytic stability of py-RuO₂:Zn for OER would be enhanced. To gain more insights into the effect of Zn doping and V_O defects on the structure stabilization of RuO₂, the crystal orbital Hamilton population (COHP) of Ru–O and Zn–O bonds, as well as Ru···Ru and Ru···Zn metal couplings, were analyzed on the optimized RuO₂, RuO₂:Zn, and RuO₂:Zn_V_O surfaces. As shown in Fig. 5f, the integrated COHP (ICOHP) values of Ru_{cus}–O for RuO₂:Zn, and RuO₂:Zn_V_O are –3.40 eV and –2.44 eV, respectively, which have been negatively shifted from that for pristine RuO₂ (–2.25 eV), thereby revealing a strengthened Ru_{cus}–O bond on those Zn-doped samples. In addition, small negative ICOHP values of Ru_{cus}···Zn were found on both the RuO₂:Zn (–0.04 eV) and RuO₂:Zn_V_O (–0.03 eV) with the Zn doping, indicating a weak long range orbital coupling between Zn dopants and the vicinal Ru_{cus} sites. In contrast, there is no clear interaction of Ru_{cus}···Ru_{cus} (0.08 eV for ICOHP) on the pristine RuO₂. Accordingly, the Ru_{cus} sites would be further stabilized by the Zn dopants. When bridged V_O defects present, the ICOHP of Ru_{bri}···Ru_{bri} for RuO₂:Zn_V_O also acquired a small negative value of –0.01 eV, while it was a positive value of 0.12 eV on both the RuO₂ and RuO₂:Zn to –0.01 eV on RuO₂:Zn_V_O. This indicated an enhanced interaction between two adjacent Ru_{bri} sites in the vicinity of V_O defect. The enhanced stability of Zn doped RuO₂ with Vo is also demonstrated by the de-metallization energies of Ru and Zn (Fig. 5g). The doping of Zn induced an increased de-metallization energy of Ru by around 0.5 eV and thus stabilized the RuO₂. The Zn dopants themselves possessed relatively higher de-metallization energies by around 0.2 eV than the Ru in RuO₂:Zn_V_O. The overall results suggested that the RuO₂ structure become more stable after the introduction of Zn dopants and V_O defects.”

References cited in this section are as follows.

- “21. Rao, R. R. *et al.* Operando identification of site-dependent water oxidation activity on ruthenium dioxide single-crystal surfaces. *Nat. Catal.* **3**, 516–525 (2020).
86. Rao, R. R. *et al.* Towards identifying the active sites on RuO₂(110) in catalyzing oxygen evolution. *Energy Environ. Sci.* **10**, 2626–2637 (2017).
87. Stoerzinger, K. A. *et al.* Orientation-dependent oxygen evolution on RuO₂ without lattice exchange. *ACS Energy Lett.* **2**, 876–881 (2017).
88. Guerrini, E., Consonni, V. & Trasatti, S. Surface and electrocatalytic properties of well-defined and vicinal RuO₂ single crystal faces. *J. Solid State Electrochem.* **9**, 320–329 (2005).”

Comments: 8. Is the methodology sound? Does the work meet the expected standards in your field?

The whole DFT part and claims there are simply unsupported in the data.

Response: Thanks for the comment. The whole DFT part has been redone. *The main changes include: RPBE functional, and a more stable RuO₂ (110) surface is used which is consistent with previous report (J. Phys. Chem. C 2015, 119, 4827–4833, J. Phys. Chem. C 2017, 121, 18516–18524, Angew. Chem. Int. Ed. 2021, 60, 2–11, Nat. Commun. 2022, 13, 3784).*

Fig. R29a shows the optimized RuO₂ (110) surface used for DFT calculations. The Zn doped RuO₂ (RuO₂:Zn) and with O vacancies (RuO₂:Zn_V_O) were built on the optimized RuO₂ (110) surfaces. Zn was found to be more stably when doped at the coordinatively unsaturated Ru (Ru_{CUS}) position than the fully coordinated bridge Ru (Ru_{brl}) site, while the bridge row O could form stable vacancy site. The total energies of Zn doping and V_O at different location was compared in the Figs. R29b and c. All these optimized surfaces were further used for OER simulations.

Fig. R29 Optimized structures of (a) pristine RuO₂ (110) surface (side view, top view, and Ru sites with different coordination), (b) Zn doped RuO₂ (RuO₂:Zn) surface, and V_O-containing RuO₂:Zn (RuO₂:Zn_V_O) surface.

Fig. R30 shows the OER mechanism studies on the optimized RuO₂, RuO₂:Zn, RuO₂:Zn_V_O surfaces. The AEM and LOM path, and the recently highlighted dual-site OPM path were investigated. For clean RuO₂, the OER preferred to proceed via a AEM path, following four-proton-coupled electron transfer steps as H₂O → *OH → *O → *OOH → O₂. The formation of *OOH is the rate-limiting step with a large free energies barrier of 2.10 eV. The LOM and dual site OPM paths are suppressed due to higher ΔG_{max} (LOM 3.79 eV and OPM 2.48 eV). By comparison, the presence of V_O defects leads to an increased ΔG_{max} of 2.28 eV (AEM) for RuO₂_V_O, thus harmful to the activity. For Zn doped RuO₂, the Zn dopants appear as another active sites to bind OH adsorbates, enabling the Ru–Zn dual-site OPM path to become more favorable with a lower ΔG_{max} of 1.91 eV. With the O vacancy presence, the OPM ΔG_{max} further decreased to 1.84 eV on the surface of RuO₂:Zn_V_O, which is the lowest value obtained on the four kinds of concerned surfaces. Consequently, the best OER activity is achieved on RuO₂:Zn_V_O.

Then, the density of states (DOS) and charge density analysis were performed to further understand the generation of OPM path. As shown in Fig. R30d and e, Zn ions has donated some electrons to the O and Zn had a lower d band center than Ru. Therefore, Zn showed weaker absorption of *O, *OH, and *OOH. For example, Zn sites had a ΔG_{OH} of 1.77 eV, while Ru site had had a ΔG_{OH} of 1.01 eV. This would ease the formation of second *O. In addition, the charge difference between Zn and Ru also played an important role in promoting the OER, which has resulted in a $\sim 0.1 e$ charge difference for the two absorbed *O on Zn and Ru and thus promoted the O–O coupling. With the presence of V_{O} defects, the charge density at both the Ru_{CUS} and Zn_{CUS} sites on $\text{RuO}_2:\text{Zn}_V\text{O}$ surface is slightly increased, associated with a shift of Ru d band center away from Fermi, which further optimized the absorption of intermediates. Under the conditions, the $\Delta G_{\text{max}} (*\text{O}_{\text{Ru}} \rightarrow *\text{O}_{\text{Ru}} \dots *\text{OH}_{\text{Zn}})$ of OPM is decreased to 1.84 eV. Therefore, we believe that the down shift of Fermi by O vacancy, the weaker absorption of *OH on Zn and the charge difference of Zn and Ru have synergistically altered the preferred OER path from the single-site AEM on pristine RuO_2 to the dual-site OPM on O vacancy-containing Zn doped RuO_2 and thus induced a significantly enhanced OER activity.

Fig. R30 (a) OER mechanisms. (b) AEM and OPM paths of OER on V_{O} -containing Zn-doped RuO_2 catalyst. (c) Calculated free-energy diagrams for preferred OER path on RuO_2 , $\text{RuO}_2:\text{Zn}$, and $\text{RuO}_2:\text{Zn}_V\text{O}$ surfaces. (d) Charge density analysis of clean $\text{RuO}_2:\text{Zn}$, and *O bonded RuO_2 and $\text{RuO}_2:\text{Zn}$. Brown, red, and dark cyan spheres represent Ru, O, and Zn atoms, respectively. (e) PDOS of Ru 4d, O 2p, and Zn 4d-bands for RuO_2 , $\text{RuO}_2:\text{Zn}$, and $\text{RuO}_2:\text{Zn}_V\text{O}$; corresponding d-band centers are denoted by dashed lines.

Fig. R29 has been added in the revised SI file, shown as Supplementary Fig. 41. Other DFT results have been added in the revised SI file, shown as Supplementary Figures 42–46 and Tables 6–7. Fig. 30 has been updated in the Fig. 5 of the revised manuscript. Following discussion has been updated in the revised manuscript.

On page 24–26: “To understand the Zn doping and oxygen vacancies effect on the OER activity, density functional theory (DFT) calculations were performed. The Zn doped RuO₂ (RuO₂:Zn) and with O vacancies (RuO₂:Zn_V_O) were built on the optimized RuO₂ (110) surfaces (Supplementary Fig. 41). Zn was found to be more stably doped at the coordinatively unsaturated Ru (Ru_{cus}) position than the fully coordinated bridge Ru (Ru_{bri}) site, while the bridge row O could form stable vacancy site. Then, different OER paths were investigated to determine the preferred reaction pathways, including the AEM and lattice oxygen mechanism (LOM), as well as the recently highlighted dual-site oxide path mechanism (OPM) (Supplementary Fig. 42).^{35,83} The adsorption energies of reaction intermediates were summarized in the Supplementary Table 6. For clean RuO₂, stronger binding of OH adsorbates ($\Delta G_{\text{OH}} = 0.82$ eV) resulted in the OER proceeding favorably via a AEM path, following the four-proton-coupled electron transfer steps as H₂O → *OH → *O → *OOH → O₂.³⁶ The formation of *OOH is the rate-determining step (RDS) with a large free energies barrier of 2.10 eV. By comparison, the LOM and dual site OPM paths are suppressed with much higher energy barriers of RDS (ΔG_{max} for LOM 3.79 eV and OPM 2.48 eV, where ΔG_{max} is the maximum free energy differences among the primary proton-coupled electron transfer steps) (Supplementary Fig. 43). For RuO₂_V_O, the presence of bridged O vacancies caused accumulated charge density at both the vicinal Ru_{bri} and Ru_{cus} sites (Supplementary Fig. 44), which then enhanced the binding of *OH at Ru_{cus} centers ($\Delta G_{\text{OH}} = 0.70$ eV) and induced a larger free energies barrier of 2.28 eV for *OOH formation (Supplementary Fig. 45). Therefore, the presence of V_O defects is harmful to the OER proceeding on RuO₂.^{37,38} In contrast, on the surface of stoichiometric RuO₂:Zn oxide, the doping of Zn at Ru_{cus} sites induced a reduction of the charge density at Ru centers, which agreed with the knowledge that a fraction of the Ru will be oxidized above +4 to accommodate the divalent Zn metal.³⁰ As a result, the *OH binding is weakened ($\Delta G_{\text{OH}} = 1.01$ eV) and the OER activity is improved. More interestingly, a Ru–Zn dual-site OPM appeared to be more favorable with a lower ΔG_{max} of 1.91 eV for *O_{Ru} → *O_{Ru...*}OH_{Zn} of the third proton-coupled electron transfer step, caused by the different binding strength of intermediates on the two sites (Supplementary Fig. 46). The density of states (DOS) and charge density difference suggested that Zn donated some electron to the O and Zn had a lower d band center than Ru (Fig. 5c–e). Therefore, Zn showed weaker absorption of *O, *OH, and *OOH. For example, Zn sites had a ΔG_{OH} of 1.77 eV, while Ru site had had a ΔG_{OH} of 1.01 eV. This would ease the formation of second *O. In addition, the charge difference between Zn and Ru also played an important role in promoting the OER, which resulted in a ~0.1 e charge difference for the two absorbed *O on Zn and Ru and thus promoted the formation of O–O coupling, and eventually the formation of O₂ (Fig. 5d). With the presence of V_O defects, the charge density at both the Ru_{cus} and Zn_{cus} sites on RuO₂:Zn_V_O surface is slightly increased (Supplementary Fig. 44), associated with a shift of Ru d band center away from Fermi, which further optimized the absorption of intermediates (Fig. 5e). Consequently, the ΔG_{max} (*O_{Ru} → *O_{Ru...*}OH_{Zn}) of OPM is further decreased to 1.84 eV for RuO₂:Zn with V_O defects (Fig. 5b). Therefore, we believed that the down shift of Fermi by O vacancy, the weaker absorption of *OH on Zn and the charge difference of Zn and Ru synergistically lowered the OER overpotential ($\eta = \Delta G_{\text{max}} - 2.13$) from 0.87 V for RuO₂ to 0.61 V for the O vacancy-containing Zn doped RuO₂, by converting the OER path from the single-site AEM to the dual-site OPM (Fig. 5a, b).”

References cited in this section are listed as follows.

“30. Burnett, D. L. *et al.* (M,Ru)O₂ (M = Mg, Zn, Cu, Ni, Co) rutiles and their use as oxygen evolution electrocatalysts in membrane electrode assemblies under acidic conditions. *Chem. Mater.* **32**, 6150–6160 (2020).

35. Lin, C. *et al.* In-situ reconstructed Ru atom array on α -MnO₂ with enhanced performance for acidic

water oxidation. *Nat. Catal.* **4**, 1012–1023 (2021).

36. Man, I. C. *et al.* Universality in oxygen evolution electrocatalysis on oxide surfaces. *ChemCatChem* **3**, 1159–1165 (2011).

37. Gayen, P., Saha, S., Bhattacharyya, K. & Ramani, V. K. Oxidation state and oxygen-vacancy-induced work function controls bifunctional oxygen electrocatalytic activity. *ACS Catal.* **10**, 7734–7746 (2020).

38. Hubert, M. A. *et al.* Acidic oxygen evolution reaction activity–stability relationships in Ru-based pyrochlores. *ACS Catal.* **10**, 12182–12196 (2020).

83. Vonrüti, N., Rao, R., Giordano, L., Shao-Horn, Y. & Aschauer, U. Implications of nonelectrochemical reaction steps on the oxygen evolution reaction: Oxygen dimer formation on perovskite oxide and oxynitride surfaces. *ACS Catal.* **12**, 1433–1442 (2022).”

Comments: 9. Is there enough detail provided in the methods for the work to be reproduced?

To a degree.

Response: Thanks for this comment. More experimental and computational details have been added in the revised SI file.

Comments: 10. In summary, the low overpotential is likely due to oxidation of water (not O₂ evolution) or dual site OER mechanism, but the authors failed to prove convincingly that is caused by Ru +3.

Response: Thanks for the insightful comments. Our additional work indeed found the dual site OPM OER mechanism is the underlie reason for the low overpotential.

Our response to the reviewer’s concern on the OER mechanism on py-RuO₂:Zn catalyst:

In this work, an overpotential of 173 mV at 10 mA cm⁻² was observed on the LSV curve of OER for the Zn-doped RuO₂ (py-RuO₂:Zn) catalyst, which was about 200 mV lower than that for the commercial RuO₂ catalyst. Such a low overpotential is impressive because it well exceeds the theoretical limit (~250 mV for the OER overpotential) on the optimal catalyst, following the linear scaling relationships between the adsorption energies of *O, *OH, and *OOH intermediates ($\Delta E_{OOH} = \Delta E_{OH} + 3.2 \text{ eV} \pm 0.2 \text{ eV}$) (ChemCatChem 2011, 3, 1159–1165; Phys. Chem. Chem. Phys. 2018, 20, 3813–3818). The result suggested that there may be other pathways of OER on py-RuO₂:Zn in addition to the adsorbate evolving mechanism (ChemCatChem 2011, 3, 1159–1165), at least at the low overpotentials. To deeply understand the OER enhancement on py-RuO₂:Zn catalyst, DFT calculations were performed following mechanisms including the frequently discussed AEM and LOM, as well as the recently highlighted dual-site OPM. The results are shown in Fig. R31a–c. For clean RuO₂, the OER preferred to proceeds via a AEM path, following four-proton-coupled electron transfer steps as H₂O → *OH → *O → *OOH → O₂. The formation of *OOH is the rate-limiting step with a large free energies barrier of 2.10 eV. The LOM and dual site OPM paths are suppressed due to higher ΔG_{max} (LOM 3.79 eV and OPM 2.48 eV). By comparison, the presence of V_O defects leads to an increased ΔG_{max} of 2.28 eV (AEM) for RuO₂_V_O, thus harmful to the activity. For Zn doped RuO₂, the Zn dopants appear as another active sites to bind OH adsorbates, enabling the Ru–Zn dual-site OPM path to become more favorable with a lower ΔG_{max} of 1.91 eV. With the O vacancy presence, the OPM ΔG_{max} further decreased to 1.84 eV on the surface of RuO₂:Zn_V_O, which is the lowest value obtained on the four kinds of concerned surfaces. Consequently, the best OER activity is achieved on RuO₂:Zn_V_O.

Then, the density of states (DOS) and charge density analysis were performed to further understand the generation of OPM path. As shown in Fig. R31d and e, Zn ions has donated some electrons to the O and Zn had a lower d band center than Ru. Therefore, Zn showed weaker absorption of *O, *OH, and *OOH. For

example, Zn sites had a ΔG_{OH} of 1.77 eV, while Ru site had had a ΔG_{OH} of 1.01 eV. This would ease the formation of second $\ast\text{O}$. In addition, the charge difference between Zn and Ru also played an important role in promoting the OER, which has resulted in a $\sim 0.1 e$ charge difference for the two adsorbed $\ast\text{O}$ on Zn and Ru and thus promoted the O–O coupling. With the presence of V_O defects, the charge density at both the Ru_{CUS} and Zn_{CUS} sites on $\text{RuO}_2\text{:Zn}_\text{V}_\text{O}$ surface is slightly increased, associated with a shift of Ru d band center away from Fermi, which further optimized the absorption of intermediates. Under the conditions, the $\Delta G_{\text{max}} (\ast\text{O}_{\text{Ru}} \rightarrow \ast\text{O}_{\text{Ru}} \dots \ast\text{OH}_{\text{Zn}})$ of OPM is decreased to 1.84 eV. Therefore, we believe that the down shift of Fermi by O vacancy, the weaker absorption of $\ast\text{OH}$ on Zn and the charge difference of Zn and Ru have synergistically altered the preferred OER path from the single-site AEM on pristine RuO_2 to the dual-site OPM on O vacancy-containing Zn doped RuO_2 and thus induced a significantly enhanced OER activity.

The whole DFT part has been updated and highlighted in red on page 24–26 in the revised manuscript. Supplementary results have been added in the revised SI file, shown as Supplementary Figures 41–46 and Tables 6–7.

Fig. R31 (a) OER mechanisms. (b) AEM and OPM paths of OER on V_O -containing Zn-doped RuO_2 catalyst. (c) Calculated free-energy diagrams for preferred OER path on RuO_2 , $\text{RuO}_2\text{:Zn}$, and $\text{RuO}_2\text{:Zn}_{\text{V}_\text{O}}$ surfaces. (d) Charge density analysis of clean $\text{RuO}_2\text{:Zn}$, and $\ast\text{O}$ bonded RuO_2 and $\text{RuO}_2\text{:Zn}$. Brown, red, and dark cyan spheres represent Ru, O, and Zn atoms, respectively. (e) PDOS of Ru 4d, O 2p, and Zn 4d-bands for RuO_2 , $\text{RuO}_2\text{:Zn}$, and $\text{RuO}_2\text{:Zn}_{\text{V}_\text{O}}$; corresponding d-band centers are denoted by dashed lines.

Our response to the reviewer’s concern on the origin of low overpotential observed on py-RuO₂:Zn catalyst:

The nature of the electrode process observed at the low overpotentials on the LSV curve of OER is important with respect to the understanding of OER kinetics and mechanism. Recently, Scott and

colleagues performed a trace detection of O₂ at low overpotentials of OER using a chip-electrochemistry-mass spectrometry (EC-MS) setup (Energy Environ. Sci., 2022, 15, 1977–1987). They observed an electrochemical generation of O₂ from OER on the RuO_x catalyst at the potential as low as 1.30 V, only 70 mV above the standard thermodynamic potential for water oxidation to oxygen. Such a low overpotential is impressive because it well exceeds the theoretical limit (~250 mV for the OER overpotential) on the optimal catalyst, following the linear scaling relationships between the adsorption energies of *O, *OH, and *OOH intermediates ($\Delta E_{\text{OOH}} = \Delta E_{\text{OH}} + 3.2 \text{ eV} \pm 0.2 \text{ eV}$) (ChemCatChem 2011, 3, 1159–1165; Phys. Chem. Chem. Phys. 2018, 20, 3813–3818). The results suggested that there may be other pathways of OER on RuO_x in addition to the adsorbate evolving mechanism (ChemCatChem 2011, 3, 1159–1165), at least at the low overpotentials. They further found that the related faradaic efficiency (FE) of OER was ~20%, while it was ~100% at potentials higher than 1.40 V. The decrease of FE at low overpotentials was probably caused by the increase of residual capacitance current ratio. By comparing the trends in Ru dissolution and oxygen evolution, they suggested a negligible contribution of lattice oxygen evolution to the overall OER activity for RuO_x in acidic media (Energy Environ. Sci., 2022, 15, 1988–2001).

In this work, an overpotential of 173 mV at 10 mA cm⁻² was observed on the LSV curve of OER for the Zn-doped RuO₂ (py-RuO₂:Zn) catalyst, which was about 200 mV lower than that for the commercial RuO₂ catalyst. We then studied the electrode process at low potentials around 1.40 V using a rotating ring-disk (Pt-GC) electrode (RRDE, E7R9PTGC from PINE Research Instrum.) setup to understand whether it was the 4e⁻ oxidation of water to O₂. Collection efficiency of the RRDE (*N*) was first calibrated with the Fe(CN)₆³⁻/Fe(CN)₆⁴⁻ redox couple (Fig. R32a). The measured value was *N* = 36.4%, well agreeing with the parameter, 37%, provided by the manufacturer. RRDE measurement of OER was then performed on the py-RuO₂:Zn catalyst with a reduced loading of 15 μg cm⁻² to minimize the disturbance from O₂ bubble accumulation (Nat. Catal. 2021, 4, 1012–1023). As shown in Fig. R32b, anodic current appeared on the catalyst loaded GC disk electrode when the potential swept to about 1.35 V. Meanwhile, clear cathodic current was observed on the Pt ring, indicating an explicit reduction of the products released from the disk electrode. The products could be the O₂, dissolved Ru cations, and/or H₂O₂. However, in the stability test at 10 mA cm⁻² we have found a good stability number (molar ratio of the evolved O₂ to the dissolved Ru element) of the py-RuO₂:Zn catalyst (Fig. R23d). Thus, the contribution of Ru dissolution/reduction to the anodic/cathodic current on the disk/ring electrodes would be negligible. For the possible production of H₂O₂, Nørskov and colleagues recently reported a 2e⁻ oxidation path of water to H₂O₂ in addition to the 4e⁻ path to O₂ (i.e., OER). However, they also found that the OER path was thermodynamically more favorable on RuO₂ (J. Phys. Chem. Lett. 2015, 6, 4224–4228), consistent with the previous research from Dousikou and co-authors who found an electrooxidation of H₂O₂ on RuO₂ beginning at potentials around 1.0 V, negative to the standard thermodynamic potential of OER (Phytochem. Anal. 2006, 17: 255–261). Therefore, it seems that the 2e⁻ oxidation of water to H₂O₂ has not contributed to the observed current on the disk.

We further calculated the corresponding faradaic efficiency of OER ($FE_{\text{OER}} = i_{\text{ring}} / (i_{\text{disk}} \times N)$, assuming only a 4e⁻ ORR process existing on the Pt ring electrode). As shown in Fig. R32b, a volcanic trend in the FE_{OER} was found with a maximum value of about 65%. Obviously, the measured FE_{OER} were generally lower than the expected FE_{OER} = 100%. The deviation probably stems from two sources: (i) the low efficient mass transfer of the produced O₂ from the ring to the disk, attributed to the slow kinetics of O₂ gas dissolution; (ii) the reduced Pt ring activity for ORR, due to the deposition of dissolved Ru on the Pt surface. During the RRDE test, a part of the O₂ product may accumulate on the surface of catalyst due to the slow kinetics of

O₂ gas dissolution, thereby reducing the amount of dissolved O₂ that can transfer to the ring electrode and the related FE_{OER}. But with the potential increasing more O₂ was produced, alleviating the impact of O₂ accumulation. Thus, there was a gradual increase of FE_{OER} (the left branch of the FE_{OER} plot). However, at more positive potentials the formation and release of O₂ in the form of bubbles was accelerated. As a competitive process of oxygen dissolution, it would inevitably impair the measurement of FE_{OER} (the right branch of the FE_{OER} plot) (Nat. Catal. 2021, 4, 1012–1023). In addition, the dissolved Ru from the catalyst (disk) can be also electrochemically reduced on the Pt ring electrode. It was reported that the deposition of Ru has detrimental impact on the ORR activity of the Pt ring electrode, although the amount of Ru was small. (ECS Trans. 2006, 3, 607). In short, the results of RRDE more support the observed anodic process as the 4e⁻ oxidation of water to oxygen, i.e., OER, rather than other water oxidation processes.

Fig. R32 (a) Calibration of the collection efficiency of RRDE using the $\text{Fe}(\text{CN})_6^{3-}/\text{Fe}(\text{CN})_6^{4-}$ redox couple in Ar-saturated solution of 0.1 M KNO_3 and 10 mM $\text{K}_3\text{Fe}(\text{CN})_6$. Pt ring potential was held at 0.8 V vs Ag/AgCl. (b) Disk and ring currents collected in the RRDE measurement for OER on the py-RuO₂:Zn catalyst in Ar-saturated solution of 0.5 M H_2SO_4 , together with the calculated faradaic efficiency (FE) of OER. The catalyst with a loading of 15 $\mu\text{g cm}^{-2}$ was dispersed on the GC disk electrode. Pt ring potential was held at 0.1 V vs RHE, while a potential scan rate of 2 mV s^{-1} was performed on the disk electrode. The RRDE was operated at a constant rotation rate of 1600 rpm.

Following discussion has been added in the revised manuscript.

On page 23: “We then performed experiments using a rotating ring-disk electrode (RRDE) setup and confirmed the explicit contribution of OER process to the observed anodic current at potentials around 1.40 V (Supplementary Fig. 39).”

Fig. R32 (shown as Supplementary Fig. 39) and the discussion as bellow have been added in the revised SI file.

On page S36–S37 in SI file: “In this work, an overpotential of 173 mV at 10 mA cm^{-2} was observed on the LSV curve of OER for the Zn-doped RuO₂ (py-RuO₂:Zn) catalyst, which was about 200 mV lower than that for the commercial RuO₂ catalyst. We then studied the electrode process at low potentials around 1.40 V_{RHE} using a rotating ring-disk (Pt-GC) electrode (RRDE, E7R9PTGC from PINE Research Instrum.) setup to understand whether it was the 4e⁻ oxidation of water to O₂. Collection efficiency of the RRDE (*M*) was first calibrated with the $\text{Fe}(\text{CN})_6^{3-}/\text{Fe}(\text{CN})_6^{4-}$ redox couple (Supplementary Figure 39a). The measured value

was $N = 36.4\%$, well agreeing with the parameter, 37% , provided by the manufacturer. RRDE measurement of OER was then performed on the py-RuO₂:Zn catalyst with a reduced loading of $15 \mu\text{g cm}^{-2}$ to minimize the disturbance from O₂ bubble accumulation (Nat. Catal. 2021, 4, 1012-1023). As shown in Supplementary Figure 39b, anodic current appeared on the catalyst loaded GC disk electrode when the potential swept to about 1.35 V. Meanwhile, clear cathodic current was observed on the Pt ring, indicating an explicit reduction of the products released from the disk electrode. The products could be the O₂, dissolved Ru cations, and/or H₂O₂. However, in the stability test at 10 mA cm^{-2} we have found a good stability number (molar ratio of the evolved O₂ to the dissolved Ru element) of the py-RuO₂:Zn catalyst (Fig. 3e, lower plot in the main text). Thus, the contribution of Ru dissolution/reduction to the anodic/cathodic current on the disk/ring electrodes would be negligible. For the possible production of H₂O₂, Nørskov and colleagues recently reported a $2e^-$ oxidation path of water to H₂O₂ in addition to the $4e^-$ path to O₂ (i.e., OER). However, they also found that the OER path was thermodynamically more favorable on RuO₂ (J. Phys. Chem. Lett. 2015, 6, 4224-4228), consistent with the previous research from Dousikou and co-authors who found an electrooxidation of H₂O₂ on RuO₂ beginning at potentials around 1.0 V, negative to the standard thermodynamic potential of OER (Phytochem. Anal. 2006, 17: 255-261). Therefore, it seems unlikely that the $2e^-$ oxidation of water to H₂O₂ has contributed to the observed current on the disk.

We further calculated the corresponding faradaic efficiency of OER ($\text{FE}_{\text{OER}} = i_{\text{ring}}/(i_{\text{disk}} \times N)$, assuming only a $4e^-$ ORR process existing on the Pt ring electrode). As shown in Supplementary Figure 40b, a volcanic trend in the FE_{OER} was found with a maximum value of about 65%. Obviously, the measured FE_{OER} were generally lower than the expected $\text{FE}_{\text{OER}} = 100\%$. The deviation probably stems from two sources: (i) the low efficient mass transfer of the produced O₂ from the ring to the disk, attributed to the slow kinetics of O₂ gas dissolution; (ii) the reduced Pt ring activity for ORR, due to the deposition of dissolved Ru on the Pt surface. During the RRDE test, a part of the O₂ product may accumulate on the surface of catalyst due to the slow kinetics of O₂ gas dissolution, thereby reducing the amount of dissolved O₂ that can transfer to the ring electrode and the related FE_{OER} . But with the potential increasing more O₂ was produced, alleviating the impact of O₂ accumulation. Thus, there was a gradual increase of FE_{OER} (the left branch of the FE_{OER} plot). However, at more positive potentials the formation and release of O₂ in the form of bubbles was accelerated. As a competitive process of oxygen dissolution, it would inevitably impair the measurement of FE_{OER} (the right branch of the FE_{OER} plot) (Nat. Catal. 2021, 4, 1012-1023). In addition, the dissolved Ru from the catalyst (disk) can be also reduced on the Pt ring electrode. It was reported that the deposition of Ru has detrimental impact on the ORR activity of the Pt ring electrode, although the amount of Ru was small. (ECS Trans. 2006, 3, 607). In short, the results of RRDE more support the observed anodic process as the $4e^-$ oxidation of water to oxygen, i.e., OER, rather than other water oxidation processes."

REVIEWERS' COMMENTS

Reviewer #1 (Remarks to the Author):

The revised manuscript is now acceptable for publication.

Reviewer #2 (Remarks to the Author):

The authors have addressed all the changes needed for publication in Nature Communications

Reviewer #3 (Remarks to the Author):

I am now happy with the revised manuscript.

I recommend accepting in the Nature Comm.